

# The "Golden Points" and nonequilibrium correction of high-accuracy frost point hygrometers

Yann Poltera[1,2], Beiping Luo[2], Frank G. Wienhold[2], and Thomas Peter[2]

[1]Institute of Applied Physics, University of Bern, CH-3012 Bern, Switzerland
[2]Institute for Atmospheric and Climate Science, ETH Zurich, CH-8032 Zurich, Switzerland

*Correspondence to*: Beiping Luo (beiping.luo@env.ethz.ch)

**Abstract.** We introduce a new retrieval protocol for chilled mirror hygrometer measurements under rapidly changing humidity conditions that enables balloon-borne frost point measurements in the upper troposphere/lower stratosphere of unprecedented accuracy. Chilled mirror hygrometers measure the frost point (or dew point) of air by quantifying the degree of saturation of the air with respect to the condensed phases of water (ice or liquid water). To this end, they attempt to determine the thermodynamic equilibrium of the mirror condensate with the vapor phase by measuring the mirror reflectance, which changes with the amount of condensed material. In the rapidly changing environment along the balloon trajectory, however, the adjustment of the mirror temperature to the new equilibrium point leads to frequent, damped overshoots or nonequilibrium errors. For the Cryogenic Frost Point Hygrometer (CFH), a balloon-borne chilled mirror instrument of reference quality, we (i) identify points in time along the sounding profile when the mirror is in true equilibrium with the gas phase, which we term 'Golden Points', and (ii) correct the measurements under nonequilibrium conditions between these Golden Points. For (i), we identify the points where the mirror reflectance assumes an extreme value, i.e. a maximum or a minimum. At these extreme points, the CFH mirror temperature represents the frost point with an accuracy better than 0.2 K (resulting from the uncertainties of the mirror temperature sensor and of the precise timing of the Golden Points along the sounding profile). These accurately determined frost points can be used to detect and correct offsets, biases and time-lag errors in other humidity sensors flown together with CFH on the same balloon payload, such as the FLASH-B fluorescence hygrometer or the thin-film capacitive hygrometer of the Vaisala RS41 radiosonde. In the middle stratosphere (~ 28 km), a frost point uncertainty of 0.2 K corresponds to < 4 % uncertainty in $H_2O$ mixing ratio (including the 0.3 hPa uncertainty of the RS41 radiosonde GPS-based pressure measurement), assuming the absence of degassing from the balloon or from instrument components. At lower altitudes, the uncertainty is even less. For (ii), we compute the time-derivative of the mirror reflectance, which is proportional to the nonequilibrium error. The proportionality factor is related to a property of the mirror condensate, which we term 'morphological sensitivity', and allows correction of the CFH nonequilibrium data. The sensitivity constant is determined using an a-priori reference, such as the RS41 radiosonde humidity measurements after they have been time-lag and bias-corrected by means of (i). Using 70 nighttime CFH-RS41 tandem flights, we find that the deviations from equilibrium of CFH are typically less than 0.5 K, which corresponds to less than 10 % error in $H_2O$ mixing ratio in the tropopause region. While this is consistent with the reported



accuracy of the CFH instrument, there are situations when the mirror temperature deviates significantly from the true atmospheric frost point, exceeding 3 K (or > 40 % error in $H_2O$ mixing ratio) in the tropopause region. Such large errors of CFH are due to suboptimal control of the mirror temperature in certain measurement scenarios (such as large mixing ratio changes in the atmosphere or the presence of a coarse ice film on the mirror). We estimate that the nonequilibrium correction removes over 80 % of the nonequilibrium error, which is superior to the low-pass filtering and time-lag correction techniques found in the literature. This procedure paves the way for frost point measurements that meet the requirements for $H_2O$ mixing ratios better than 4 % (at $2\sigma$) set by the World Meteorological Organization in 2023 as target for reference instrumentation measuring water vapor in the atmosphere.

# 1   Introduction

## 1.1   Role of water vapor in the atmosphere

Gaseous $H_2O$ is a major component of the Earth's atmosphere and a driving force of weather and climate. It is the most important greenhouse gas, accounting for about half of the natural greenhouse effect, a major driver of weather in the troposphere and of great importance for the chemical self-cleansing of the atmosphere. Its volumetric mixing ratio varies between 4-5 % under warm and moist tropical conditions near the surface, where evapotranspiration supplies the air with water molecules, and only a few ppmv (parts per million by volume = $10^{-6}$) in the stratosphere and mesosphere (Dobson et al., 1946; Gille and Russell, 1984). The decrease in mixing ratio with altitude in the troposphere by four orders of magnitude is due to the temperature dependence of the saturation vapor pressure, which controls the maximum amount of water vapor an air parcel can hold before condensation and precipitation occur, and drives the strong tropospheric $H_2O$ climate feedback (Soden and Held, 2006; Dessler and Sherwood, 2009).

In the troposphere, water vapor is essential for the formation of clouds and regulates the atmospheric oxidation capacity via the formation of the hydroxyl radical, which oxidizes pollutants and enables their washout, thus continuously purifying the global atmosphere (Seinfeld and Pandis, 2006). Latent heat release is a major driver of convection and of global circulation (Manabe, 1956; Schneider et al., 2010). Water vapor also controls the formation of cirrus clouds in the upper troposphere (Peter et al., 2006), which cover about 30 % of the Earth (Wylie & Menzel, 1999), modulate the Earth's radiative budget as they interact with both long- and short-wave radiation (Chen et al., 2000) and limit the stratospheric entry of water vapor (Corti et al., 2006; Fueglistaler et al., 2009).

The dryness of the stratosphere can be explained by the Brewer-Dobson circulation (Brewer, 1949; Dobson, 1956; Holton et al., 1995): air assumes the lowest $H_2O$ mixing ratios at the cold tropical tropopause, from where it enters the stratosphere, and is transported towards mid- and high-latitudes over the course of months to years, finally re-entering the lowermost stratosphere and troposphere, mainly in the winter hemisphere. Through this freeze-drying process (Jensen and Pfister, 2004), $H_2O$ mixing





ratios as low as 1-2 ppmv can be reached above the tropical tropopause (Vömel et al., 1995), giving rise to a seasonal signal in the $H_2O$ mixing ratio carried upward by the circulation, the so-called tropical "tape-recorder" (Mote et al., 1996). In the stratosphere, photochemical oxidation of methane and hydrogen (Jones et al., 1986; Le Texier et al., 1988) slowly increases

water vapor mixing ratios again to about 7 ppmv (Chiou et al., 1997). In volcanically or pyro-Cumulonimbus perturbed periods even much higher mixing ratios have been observed (Millán et al., 2022).

Changes in stratospheric $H_2O$ abundance influence decadal trends in global surface temperature (Forster and Shine, 2002; Solomon et al., 2010). The associated surface climate warming is caused primarily by $H_2O$ reaching the lowermost stratosphere via the extratropical tropopause in a warmer climate (Dessler et al., 2013; Banerjee et al., 2019). In addition, stratospheric

water vapor enhancements lead to ozone losses (Dvortsov and Solomon, 2001; Vogel et al., 2011), as stratospheric $H_2O$ is a source of reactive radicals and promotes the formation of polar stratospheric clouds (PSCs), which affect chlorine activation and cause the formation of the springtime ozone hole (Solomon et al., 1986; Solomon, 1999).

## 1.2 Observational challenges for water vapor instrumentation

Accurate measurements and long-term monitoring of atmospheric $H_2O$ in a changing climate are of paramount importance but

nontrivial, especially in the upper troposphere/lower stratosphere (UT/LS) (Kley et al., 2000; Fahey et al., 2014). Measurements in the UT/LS can be made in-situ using balloons, aircraft and rocket-based platforms, or remotely using ground-based and satellite-based platforms (Kley et al., 2000; Kämpfer et al., 2013). Observations in the UT/LS have improved our understanding of how ice clouds form via homogeneous or heterogeneous nucleation (e.g. Krämer et al., 2020), how they dehydrate/hydrate and supersaturate the air (e.g. Vömel et al., 1995; Khaykin et al., 2009; Reinares Martínez et al., 2019), and how

water vapor enters the tropical lower stratosphere (e.g. Corti et al., 2006). They have further shown how sensitive the Earth's radiation balance is to the distribution of water vapor in the UT/LS (Held and Soden, 2000; Dessler and Sherwood, 2009), where uncertainties of 10 % lead to large discrepancies in climate modeling, comparable to the effects of a doubling of $CO_2$ concentration (Harries, 1997). At the same time, however, observations have also revealed large discrepancies between reference instruments (e.g. Peter et al., 2006; Krämer et al., 2009; Fahey et al., 2014).

Long-term monitoring for climate research requires not only high accuracy, but also long-term stability of the instrumentation. Hurst et al. (2016) reported that Aura/MLS (launched in 2004) and balloon-borne frost point hygrometers (FPH, CFH) started diverging in the early 2010s, with differences exceeding the combined uncertainty of the instruments in some locations. The divergences, albeit reduced, still persist in the most recent MLS data product (MLS v5), despite correcting for a drift in the 190 GHz sideband fraction (i.e. the relative sensitivity to the two different parts of the 190 GHz spectrum) of the MLS receiver

(Livesey et al., 2021). The challenge of capturing UT/LS water vapor on the global scale is also reflected in the data products from the European Centre for Medium-range Weather Forecasts (ECMWF), which have their highest uncertainty (biases > 10 %) in the tropopause region (Kaufmann et al., 2018; Brunamonti et al., 2019).



As part of the SPARC "Assessment of Upper Tropospheric and Stratospheric Water Vapour", Kley et al. (2000) compared over 25 scientific instruments utilizing different techniques to measure water vapor in the UT/LS. While the majority of the instruments clustered within 10 %, some exhibited discrepancies of 30 % or more, well above the stated nominal uncertainties.
This behavior was confirmed during the 2006 CR-AVE flight campaign near the tropical cold-point tropopause over Costa Rica, with discrepancies of more than ±40 % between a Lyman-α and a chilled-mirror hygrometer (Rollins et al., 2014), leading to speculations about large-scale regions of persistent supersaturation of more than 100 % in cloud-free regions, which were questioned in view of the large uncertainties in the aircraft- and balloon-borne hygrometer data (Peter et al., 2006).

This motivated organizing the AquaVIT-1 laboratory intercomparison campaign in 2007 (Fahey et al., 2014) and the MACPEX in-flight intercomparison campaign in 2011 (Rollins et al., 2014). AquaVIT-1 took place in the AIDA climate chamber at the Karlsruhe Institute of Technology (Fahey et al., 2014) and included more than 20 instruments, of which six were termed "core instruments": Tunable Diode Lasers (APic-T, JLH), Lyman-α's (FISH, FLASH-B, HWV), and a chilled mirror (CFH). For these six instruments agreement was considered "good", namely within ±10 % (±1σ) of their mean value for humidity between 10 and 150 ppmv under UT conditions, and was considered "fair", namely within ±20 % (±1σ) for humidity between 1 and 10 ppmv under LS conditions. The mean value of the core instruments was used as the reference value, because no SI-traceable, metrological primary standard was available for UT/LS conditions (low frost points at low temperatures and pressures) (Buchholz and Ebert, 2018). The better agreement in the AquaVIT-1 laboratory intercomparison compared to earlier flight campaigns can be partly attributed to the laboratory-controlled conditions compared to moving platforms, which likely caused instrument artifacts, such as instrument outgassing and cloud artifacts (Fahey et al., 2014). It is encouraging that the 2011 MACPEX in-flight comparisons yielded values within ±10 % (±1σ), comparable to the AquaVIT-1 laboratory campaign, which can be seen as progress towards better $H_2O$ measurements under the harsh campaign conditions. This improvement suggests that the causes of some of the in-flight problems that have led to large discrepancies in the past may have been mitigated, although the improvement cannot be clearly attributed to specific changes to one or more instruments or operating procedures (Rollins et al., 2014).

Similarly, Meyer et al. (2015) reported in a review of the Lyman-α hygrometer FISH (Fast In-situ Stratospheric Hygrometer) that the agreement between different types of hygrometers improved from > 30 % in the late 1990s to about 5-20 % for mixing ratios below and 0-15 % above 10 ppmv. Based on results from the 2014 ML-CIRRUS campaign, which included an instrument comparison of aircraft hygrometers (AIMS-H2O: Ionization Mass Spectroscopy; FISH: Lyman-α total water; WARAN, SHARC and HAI TDLs: tunable diode absorption spectrometers), Kaufmann et al. (2018) reported systematic differences in the order of 10 %, with maxima at 15 %, at mixing ratios < 10 ppm. Singer et al. (2022) found standard deviations of differences < 8 % between FISH (Lyman-α total water), FLASH (Lyman-α gas phase water) and the ChiWIS (TDL absorption-spectrometer) at < 10 ppmv in the clear-sky UT/LS during the 2017 Asian summer monsoon.





Potential drifts and long-term stability issues, offsets and temperature/pressure dependencies, are common hygrometry chal-
125 lenges (Buchholz et al., 2014; Sonntag et al., 2021). For example, discrepancies between FISH and FLASH (or, more generally,
offsets in Lyman-α hygrometers) is a long-standing issue (e.g. Krämer et al., 2009; Weinstock et al., 2009; Meyer et al., 2015).
These instruments allow for very fast response, but require recalibration between flights (Kaufmann et al., 2018; Singer et al.,
2022). The main challenges are the long-term stability of the UV-light source, the aging of the $MgF_2$ window (Meyer et al.,
2015), stray-light interference and water vapor contamination in the measurement cells (e.g. Sitnikov et al., 2007). It is only
130 recently that developments are underway towards SI-traceable calibration under laboratory conditions representative of the
UT/LS (e.g. Sairanen et al., 2018; Buchholz and Ebert, 2018; Lee et al., 2019; Brunamonti et al., 2023).

### 1.3 Observational and monitoring requirements for water vapor

The WMO Commission for Instruments and Methods of Observation (CIMO) and the Global Climate Observing System
(GCOS) have set *threshold* (i.e. 'useful enough'), *goal* (i.e. 'ideal') and *breakthrough* (between *threshold* and *goal*, interpret-
135 able as optimal cost-benefit) requirements for various essential variables. CIMO states, for $H_2O$ measurements in the UT/LS
with in-situ (radiosonde) sensors, the following requirements (WMO, 2023, their Annex 12.A):

- *threshold*: not specified;
- *breakthrough*: < 30 %RH 2σ uncertainty in the UT and < 20 % 2σ uncertainty in mixing ratio in the LS;
- *goal/optimum*: < 10 %RH 2σ uncertainty in the UT and < 4 % 2σ uncertainty in mixing ratio in the LS.

GCOS has set significantly more stringent requirements for $H_2O$ in the UT/LS (GCOS, 2025, their Section 2.3):

- *threshold*: < 0.5 ppmv or < 2 %RH 2σ uncertainty (depending on the quantity of interest) at 250 m vertical resolution;
- *breakthrough*: < 0.25 ppmv or < 1 %RH 2σ uncertainty at 100 m vertical resolution (based on the magnitude of the
  radiative forcing of stratospheric water vapor (about 0.24 $Wm^{-2}$ for 1 ppmv decadal change) reported by Solomon et al.
  (2010));
- *goal*: < 0.1 ppmv or < 0.5 %RH 2σ uncertainty at 10 m vertical resolution (based on the magnitude of the stratospheric
  water vapor feedback (about 0.3 $Wm^{-2}K^{-1}$) reported by Dessler et al. (2013), on interannual variability, on the data quality
  required to study supersaturation and dehydration processes, and on the vertical resolution required to study fine cirrus
  layers and complex tropopause profiles).

The WMO OSCAR (Observing Systems Capability Analysis and Review tool) rolling requirements for atmospheric climate
monitoring are also specified in terms of RH and are currently (March 2025) the same as the GCOS 2025 RH requirements
(OSCAR, 2025, Variable ID 790). We note that currently available balloon and aircraft reference instruments just meet the
GCOS 2025 breakthrough requirements, but are unable to reach the GCOS 2025 goal for $H_2O$ in the UT/LS. However, as there
is small-scale variability in atmospheric water vapor (Calbet et al., 2022) and in temperature (Podglajen et al., 2016), it remains
an open question if comparing and validating instruments in-situ at the GCOS 2025 'goal' level of accuracy and vertical
resolution is at all feasible.



## 2 Measurement methods and instruments

### 2.1 Balloon-borne hygrometry

Balloon-borne measurements are easy to deploy and provide in-situ measurements from the ground to the middle stratosphere with high vertical resolution, including regions that are difficult to access, such as the tropical UT/LS or volcanic aerosol plumes (Kley et al., 2000). Müller et al. (2016) advocate a long-term balloon-borne measurement program for atmospheric and climate research, which might be straight-forward to implement.

For example, balloon-borne measurements have provided unprecedented insights into the moist stratospheric plume following the eruption of the submarine Hunga Tonga-Hunga Ha'apai volcano on 15 January 2022 (Khaykin et al., 2022; Evan et al., 2023; Asher et al., 2023) and revealed the existence of thin moist layers with 1300 ppmv $H_2O$ (Vömel et al., 2022) that could not be resolved by the Aura/MLS satellite measurements (Millán et al., 2022). However, they are limited to an altitude of ~35 km and are only sparsely distributed in space and time. In the case of the Hunga Tonga eruption, the radiosondes could neither reach the ~53 km high ceiling of the plume in the mesosphere, nor accurately display the full horizontal extent of the plume (Millán et al., 2022; Vömel et al., 2022).

Currently there are four types of hygrometers with state-of-the-art technology that can be flown on small meteorological rubber balloons for in-situ measurements in the UT/LS: (i) chilled mirror hygrometers such as CFH or FPH (Vömel et al., 2016; Hall et al., 2016); (ii) capacitive thin-film polymer sensors such as in the Vaisala RS41 radiosonde (Survo et al., 2015; Vömel et al., 2022); (iii) Lyman-α fluorescence hygrometers such as FLASH-B (Lykov et al., 2011; Lykov and Khaykin, 2017); (iv) new developments of light-weight laser absorption spectrometers such as Pico-Light $H_2O$ (Ghysels et al., 2024) and ALBA-TROSS (Graf et al., 2021; Brunamonti et al., 2023). The chilled mirror instruments are considered "best-in-class" and are used as a reference in several research and instrument intercomparison projects (e.g., Fahey et al., 2014). In the following, we compare the performance of the CFH with the RS41 capacitive polymer sensor and the FLASH-B Lyman-α hygrometer, while the newly developed spectrometers are not yet taken into account here.

### 2.1.1 Chilled mirror instruments FPH and CFH

Balloon-borne chilled mirror instruments are suitable for multi-decadal measurements with low drift at relatively low cost (Müller et al., 2016; Hall et al., 2016). The "NOAA Frost Point Hygrometer" (FPH) measurements, routinely performed in Boulder, Colorado (40°N) since 1980, is the longest uninterrupted time series of stratospheric water vapor (e.g. Oltmans, 1985; Hurst et al., 2011a, 2016, 2023; Kiefer et al., 2023). A close relative of FPH is the "Cryogenic Frostpoint Hygrometer" (CFH), manufactured by EN-SCI, with similar measurement capabilities (Vömel et al., 2007a, 2016; Hall et al., 2016; EN-SCI, 2025).

Balloon-borne chilled mirror hygrometry is the only known technique that can measure from the ground to the middle strato-sphere with high long-term stability. The accuracy of the $H_2O$ mixing ratios derived from the dew/frostpoints measured by





CFH and FPH from the ground to an altitude of ~ 28 km is specified as better than 10-12 % after downsampling (Vömel et al., 2007a; 2016; Hall et al., 2016). Under conditions of good mirror frost control and a cloud-free troposphere these instruments offer the potential to achieve the CIMO measurement goal of less than 4 % uncertainty in $H_2O$ mixing ratio. For example, Hall et al. (2016) report that, because of good frost control at those altitudes with generally only slowly varying frost point, the

NOAA frost point hygrometer (FPH) achieves a mean uncertainty < 4 % at 250 m resolution in the stratosphere between 19 km and 25 km altitude over Boulder, Colorado (see their Fig. 6b), reaching the CIMO but not the GCOS goal. The uncertainty increases to about 5 % at ~ 28 km due to the increasing effect of the radiosonde pressure uncertainty (0.5 hPa for the iMet-1-RSB radiosonde) at higher altitudes (whereas FPH could satisfy < 4% uncertainty up to ~ 28 km if it were flown with RS41 with 0.3 hPa pressure uncertainty). Above ~ 28 km, measurements during balloon ascent often show an unrealistic increase in

the $H_2O$ mixing ratio, which is a measurement artifact due to outgassing from the balloon skin and the payload train contaminating the air at these low air densities (Brunamonti et al., 2018; Jorge et al., 2021). The onset of contamination may occur at altitudes lower than 28 km in case of strong solar radiation, moist tropospheric conditions, or short balloon to payload distance. Measurements above ~ 28 km can still be performed on balloon-borne platforms, but only in case of slow descent, which requires a dedicated descent control apparatus (e.g. Kräuchi et al., 2016).

The main source of uncertainty for FPH and CFH in the stratosphere up to ~ 28 km under clear sky conditions arises from oscillations in the mirror temperature induced by the frost control scheme, while balloon outgassing and pressure measurement uncertainties dominate the uncertainty of mixing ratios above ~ 28 km. The FPH and CFH controllers attempt to keep the voltage, which is a measure of the light reflected by the mirror with the ice layer, constant. This voltage is a measure of the reflectance of the ice-covered mirror, which is the reflected light power relative to the incident power (and small changes in

the incident power are compensated by debiasing). The controllers attempt to keep the reflectance and, thus the amount of condensed material (ice layer "thickness"), constant even in situations in which the water vapor concentration changes rapidly. This results in oscillations of the mirror temperature around the ambient frost point temperature (for CFH with ice on the mirror this is nominally to within less than 0.3 % at 1σ of the reflectance at 1 s resolution, similarly for FPH even with lower noise (Vömel et al., 2007a; Hall et al., 2016)). Such oscillations of the frost layer are unavoidable with rapidly changing

atmospheric conditions along the balloon trajectory, but are generally well controllable. However, instabilities in the electronic control system or, in rare cases, the loss of condensate, may occur, leading to measurement artifacts that are difficult to distinguish from true atmospheric features (Vömel and Jeannet, 2013). The uncertainty associated with controller oscillations is not fully understood and has been treated as an auto-correlated error component in the case of CFH (Vömel et al., 2016) and as a purely random error component (i.e. normally distributed, uncorrelated random 'noise') in the case of FPH (Hall et al., 2016),

although both instruments rely on the same measuring technique with similar hardware and operating scheme.

While the reflectance and mirror temperature oscillations caused by the frost control scheme at varying relative humidity at first appear to be a major disadvantage of FPH and CFH, we will show below that the resulting reflectance values can be used as advantage in determining the "Golden Points" of chilled mirror instruments, i.e. in obtaining frost points of unprecedented



accuracy at the reflectance extreme values, and furthermore in correcting the measurements for nonequilibrium conditions
between the Golden Points (Sections 3 and 4).

### 2.1.2    Vaisala RS41 capacitive thin-film polymer sensor

In the early 1990s, Elliott and Gaffen (1991) questioned the usefulness of radiosondes for climate studies, but since then their
performance has greatly improved. However, there are differences in hardware and processing between different radiosonde
models (Nash et al., 2011; Dirksen et al., 2024). For example, Khaykin et al. (2022) reported that the Vaisala RS92 (Vaisala,
2013) and RS41 (Vaisala, 2018a; 2018b) models were able to detect the Hunga Tonga-Hunga Ha'apai stratospheric $H_2O$ plume
in early 2022, while other operational radiosonde models could not. The capacitive thin-film humidity sensor of the Vaisala
RS41 radiosonde (successor to the Vaisala RS92 radiosonde) remains sensitive to humidity as long as the relative humidity is
greater than about 1-2 % RH (Survo et al., 2015; Brunamonti et al., 2019; Vömel et al., 2022). This suggests that it can be
utilized within certain UT/LS research topics, such as the study of deep overshooting convection, polar stratospheric clouds or
moist volcanic plumes. However, the RS41 (with the Vaisala MW41 sounding software) suffers from residual time-lag and
bias errors (leading to errors of 20 % and more (e.g. Brunamonti et al., 2019; Khordakova et al., 2022), for which a universal
correction may be applied (Poltera, 2022). In addition, there is sonde-to-sonde variability, which introduces additional uncer-
tainty in the low humidity regime of the tropopause region and the stratosphere (Brunamonti et al., 2019; Vömel et al., 2022).
It therefore remains unclear whether the RS41, even when a large number of measurements are aggregated in one location to
reduce measurement uncertainty, is suitable for climate research in the tropopause region and in the stratosphere.

### 2.1.3    The Lyman-α hygrometer FLASH-B

FLASH-B (Fluorescent Lyman-α Stratospheric Hygrometer for Balloon) is a compact light-weight balloon-borne hygrometer
for the upper troposphere and the stratosphere manufactured by CAO (Central Aerological Observatory, Russia; Yushkov et
al., 1998; 2001; Lykov et al., 2011). It is based on the Lyman-α fluorescence technique for fast measurements of low concen-
trations of water vapor, a technique first described in the late 1970's (Kley and Stone, 1978; Bertaux and Delannoy, 1978).
FLASH-B measures in open configuration with downward-facing optics. Measurements are only possible at night to protect
the instrument's photocathode. The measuring range is 0.5-1000 ppmv at pressures between 5 hPa and 300 hPa. At higher
atmospheric pressure, Lyman-α absorption by $H_2O$ vapor becomes significant and in the lower troposphere fluorescence count-
ing is strongly attenuated, which is a general limitation of the Lyman-α fluorescence method and not specific to FLASH-B
(Kley and Stone, 1978; Bertaux and Delannoy, 1978). The overall calibration error is estimated to be 4 % (1σ), and the meas-
urement accuracy for a fluorescence signal integrated over 4 s is typically 5.5 % under stratospheric conditions. The overall
uncertainty (1σ) is < 10 % for $H_2O$ mixing ratios of more than 3 ppmv and < 20 % below 3 ppmv (e.g. Vömel et al., 2007b;
Lykov et al., 2011; Lykov and Khaykin, 2017);



The performance of FLASH-B has been assessed in the laboratory during the AquaVIT-1 hygrometer intercomparison cam-
paign (Fahey et al., 2014) in the stratospheric chamber AIDA in Karlsruhe (Germany), and in the atmosphere by comparisons
with the MIAWARA microwave radiometer (Deuber et al., 2005), the FPH (Vömel et al., 2007b) and CFH (Khaykin et al.,
2013) chilled mirror hygrometers, and against the tunable diode laser IR-absorption spectrometer Pico-SDLA H$_2$O (Ghysels
et al., 2016). These comparisons showed average relative deviations less than 2.4 % in the lower stratosphere (Kalnajs et al.,
2021), which is well within the instrumental uncertainties of the instruments involved. Similar to FLASH-A, the aircraft ver-
sion of the instrument, also the balloon version requires to be recalibrated between flights, with the challenge being the long-
term stability of the UV-light source and the aging of the MgF$_2$ window (Meyer et al., 2015), as well as possible interferences,
such as stray-light interference (e.g. Sitnikov et al., 2007) or an offset in fluorescence counts (Poltera, 2022), see Section 4.3.1.

## 2.2 Operational principle of CFH

The schematics of the CFH instrument and its various elements are shown in Fig. 1. The performance of a chilled mirror
hygrometer is constrained by its sensitivity to the condensate (in mV/(μg/cm$^2$), see Eq. 2) and by the electronic noise of its
detection unit (in mV). The sensitivity to the condensate is given by the change in reflectance (in mV) per change in condensate
coverage (in μg/cm$^2$) and depends on the morphology of the condensate as well as on the implemented reflectance detection
scheme (e.g., LED with different wavelength or intensity) (Vetelino et al., 1996; Vömel et al., 2016). In CFH, the mirror is
illuminated by a light-emitting diode (LED) in the near-infrared (850 nm), whose light is specularly reflected by the mirror (at
5° incidence angle) and weakened by around 12% by the thin condensate (a few μg/cm$^2$ "thickness"). In the following we
continue to use the term "thickness" for the sake of simplicity, although this does not adequately describe the condensate that
can consist of various types of droplets or ice crystals. Trace gas interferences (such as co-condensation of HNO$_3$ or CO$_2$) are
not an issue at atmospherically relevant concentrations (Thornberry et al., 2011; Vömel et al., 2016). A photodetector measures
this reflected light (indicated by the voltage $U_m$) and is used to adjust the mirror temperature ($T_m$) within a feedback loop by
adjusting the current through a heater coil, which diminishes the continuous cooling of the mirror by the evaporating cryogen
(R23). A digital microcontroller manages this feedback loop, allowing the mirror to be heated and cooled rapidly so that its
temperature is as close as possible to the instantaneous frost point ($T_{fp}$). The amount of H$_2$O in the ambient gas phase is then
derived from the mirror temperature ($T_m$). The feedback loop aims to maintain constant thickness of the condensate, indicative
for thermodynamic equilibrium with the gas phase (Barrett and Herndon, 1951; Mastenbrook and Oltmans, 1983).

Under "good conditions", i.e. for measurements with very stable PID control (the Proportional – Integral – Derivative control-
ler used to regulate the mirror temperature), the total frost point uncertainty at the native 1-second resolution is given by $|T_m -
T_{fp}| < 0.2$ K (Fahey et al., 2014; Vömel et al., 2016). However, under rapidly changing conditions, the feedback controller can
induce temperature oscillations that are significantly higher than 0.2 K. Traditionally, the oscillations are mitigated by averag-
ing. For example, NOAA uses a 25-s boxcar filter for CFH (corresponding to ∼ 125 m vertical resolution, i.e. a half 'level' in
NOAA nomenclature), and 250 m (i.e. a full 'level', corresponding to ∼ 50-s temporal resolution) for FPH (which uses a



controller similar in operation but parameterized differently), resulting in a total uncertainty of the average frost point of typi-

cally $|\bar{T}_m - \bar{T}_{fp}| < 0.51$ K (Vömel et al., 2007a; Hall et al., 2016).

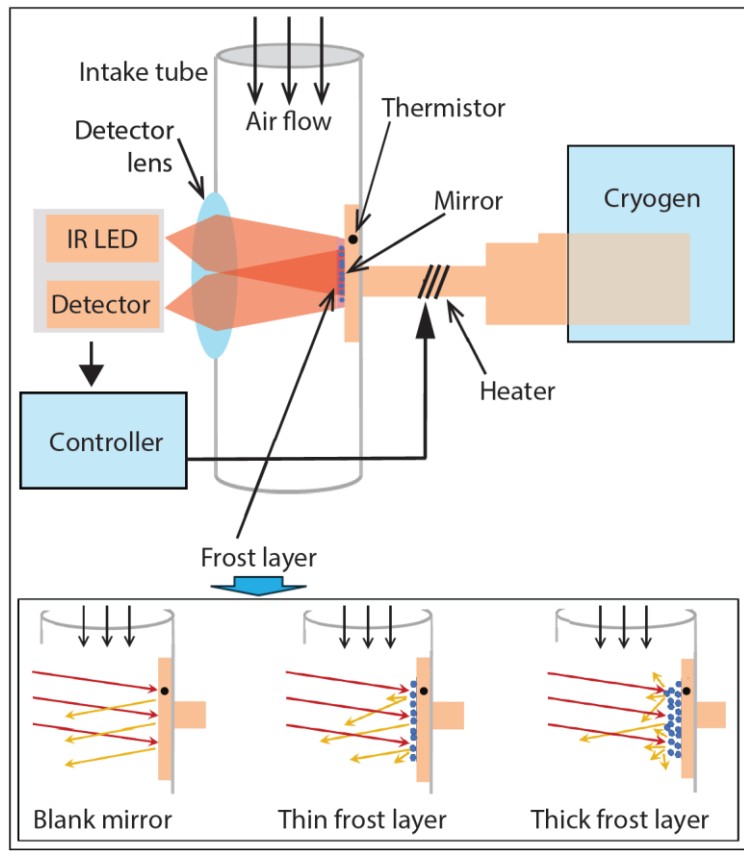

**Figure 1.** Frost point hygrometer and its various elements (not to scale). Top schematic: conceptual scheme similar to figures
shown by Vömel et al. (2007a) and Hurst et al. (2023). Bottom inserts illustrate qualitatively how deposit on the mirror in-
creases optical turbidity, thereby reducing the measured specular reflectance.

### 2.2.1 CFH feedback control and controller stability

CFH uses a digital PID controller that is tuned for fast response during ascent measurements (Vömel et al., 2007a). The control

(or "manipulated") variable is the duty cycle (pulse-width modulation, PWM, in % of the maximum 12-W heating power)

applied to the heater coil wound around the mirror stem. This results in a change in mirror temperature through Joule heating

of the cryogenically cooled cold finger. The process (or "feedback") variable is the specular reflectance $U_m$ of the mirror (with

setpoint 2.5 V, i.e. 88% specular reflection), and the mirror temperature $T_m$ is the measurement output (in K). The reflectance

and temperature measurements are such that they represent the inner 50 % of the mirror surface, thus avoiding the mirror edge

to influence the measurement. In order to achieve optimal control over the entire range of frost point temperatures expected





from the ground to the mid-stratosphere, the PID parameters vary in dependence of the frost point, using as input a slow moving average (in the order of 1 minute) of the mirror temperature, measured internally at 14 Hz (Holger Vömel, pers. comm., 2019). The feedback controller leads to slight oscillations around the true frost point of ambient air, with a typical amplitude < 1 K and oscillation period of around 25 s in the stratosphere (Vömel et al., 2007a). The phase sensitive detection (implemented at 5 kHz) also allows measurements during daytime, albeit with increased noise (Vömel et al., 2007a; Hall et al., 2016).

For long time series consisting of measurements from multiple balloon flights that have been downsampled to a coarse vertical resolution (e.g. as in Hurst et al., 2016), the error introduced by the controller oscillations is considered to be random (Vömel et al., 2016). During a single balloon flight, however, the frost point measurements (reported at 1-second resolution in the telemetry) are not randomly distributed around the true value, but exhibit a certain autocorrelation due to the oscillatory behavior of the PID controller (Vömel et al., 2016). Put simply, the reported mirror temperature periodically alternates between "swings" that are too warm or too cold. The period and amplitude of the oscillations depend on the controller's response to the transient dynamics of frost formation and evaporation, whose time scale is primarily governed by the difference between the ambient $H_2O$ partial pressure and the ice vapor pressure (time scale of kinetics $\tau_{kin} \sim p/\Delta p_{H2O}$). Further, it depends on the cryogen and ambient air temperatures, on the ventilation, and on the morphology of the frost coverage. This difficulty is only partially remedied by taking mean values of atmospheric layers, e.g. several 100 m thick, as done by Vömel et al. (2007a; 2016) and Hall et al. (2016).

For an optical chilled mirror hygrometer to provide reliable measurements, the following assumptions must be valid:

    (i)   the change in reflectance is a measure of the net $H_2O$ mass flux between the condensate and ambient air,

    (ii)   the phase of the $H_2O$ condensate is known (liquid or solid),

    (iii)   the mirror temperature is an accurate representation of the condensate temperature, and

    (iv)   the instrument is free of contamination by hydrometeors that collide with the air inlet.

While assumptions (i)-(iii) can be met relatively well by properly functioning instruments (Thornberry et al., 2011; Gao et al., 2016), assumption (iv) remains a concern even for the best hygrometers (Brunamonti et al., 2018), in particular when the payload below the balloon performs a strong pendulum motion in mixed-phased clouds, which increases the likelihood of droplets colliding with the inner walls of the inlet tubes and freezing on them (Jorge et al., 2021).

**2.2.2    CFH mirror temperature accuracy**

CFH uses a small (0.46 mm diameter) bead thermistor embedded into a gold-plated copper mirror disk of 1.27 mm thickness and 7 mm diameter (Vömel et al., 2007a). The thermistors are individually calibrated to a NIST-traceable temperature standard over the range -100°C to +25°C in a slowly warming alcohol bath that has been pre-cooled with liquid $N_2$. The NIST-traceable thermometer used for the calibration has an uncertainty ($u\_ref\_Therm$) of less than 0.02 K, including lag effects (Hall et al., 2016). The CFH uses a calibration curve for thermistors following Steinhart and Hart (1968) for the inverse absolute temperature $T^{-1} = c_0 + c_1\ln(R_T) + c_2\ln(R_T)^2 + c_3\ln(R_T)^3$ (in $K^{-1}$) with four calibration points (at -90°C, -65°C, -22°C and 18°C and



corresponding resistance values $R_T$ in Ohm) with a residual fitting error ($u\_Cal\_Fit$) < 0.04 K in the temperature range -94°C to +25°C, covering frost point temperatures from ground to the middle stratosphere (Vömel et al., 2007a). Thirty CFH mirror thermistors are calibrated in each calibration run, five of which are long-term working references that are recalibrated within
each run in order to monitor repeatability and long-term stability. Since 2004, a consistent series of CFH thermistor calibration runs has been performed exhibiting a repeatability ($u\_Cal\_Repeat$) < 0.02 K and no statistically significant decadal drift (Vömel et al., 2016). The measurement electronics introduce an uncertainty in the thermistor resistance measurement, which translates to an uncertainty ($u\_CFH\_ADC$) in the mirror temperature ranging from < 0.01 K in the stratosphere to < 0.16 K in the lower troposphere (i.e. 2σ of 0.005 K resp. 0.08 K, see Vömel et al. (2016)).

Thanks to the high thermal conductivity of copper (~ 400 W/m/K) and gold (~ 320 W/m/K), it can be assumed that thermal gradients on the mirror are small. The uncertainty in the homogeneity of the temperature across the mirror (termed $u\_Mirror\_Uniform$ by Hall et al. (2016)) is reported to be less than 0.1 K (Oltmans, 1985; Vömel et al., 2007a). The systematic error, i.e. the error component that cannot be reduced by averaging over multiple measurements, arises from the thermistor calibration, measurement electronics, and mirror temperature homogeneity uncertainties. It is estimated to be less than 0.11 K after a
variability analysis of dual CFH soundings (Vömel et al., 2016). Thermal lags between the mirror and the ice-gas interface are only relevant in cases of poor thermal conductivity due to thick coverage of more than a few tens of micrometers, which is unlikely for CFH. Also, delays in the electronics or in the data transmission are most likely small and not further considered. It is good practice to perform a ground check before each launch (Immler et al., 2010; Vömel et al., 2016). This provides traceability and allows early detection of instrumental problems, such as the thermistor installation issue that affected 15 % of
early CFH instruments (serial numbers 1L0001 to 2L2825, see Vömel et al. (2016)). Since all the CFH instruments analyzed in this work have serial numbers > 2L2825, no bias in CFH mirror temperature is suspected.

CFH performs a cleaning cycle with subsequent forced freezing (i.e. cooling the mirror to about 30 K below the frost point) twice per sounding (see e.g. Jorge et al. (2021), their Fig. 1). These cleaning cycles are performed at -15°C (or -12.5°C for early instruments) and at -53°C, the former to ensure the transformation of dew to frost and the latter to prevent the formation
of cubic or glassy (amorphous) ice (Vömel et al., 2007a). The phase of the condensate (liquid water or hexagonal ice) and associated saturation vapor pressure is therefore known at all times in CFH.

In summary, with stable PID control (i.e., within close range of thermodynamic equilibrium between the condensate and vapor phase), the dew resp. frost point accuracy of the CFH is excellent with an overall uncertainty of < 0.2 K with ice and < 0.3 K for liquid condensate, which corresponds to an error in the $H_2O$ mixing ratio of < 4 % up to the middle stratosphere (~ 28 km),
including the 0.3 hPa uncertainty of the RS41 radiosonde GPS-based pressure measurement (Vaisala, 2018b). This is also the basis for the temperature accuracy at the Golden Points presented in the next section. However, it should be noted that the accuracy of the frost point measurement depends not only on the instrument accuracy but also on how well the position of a Golden Point along a profile is determined, as a deviation of just a few meters can lead to errors of well over 4 %.



## 3 The Golden Points of chilled mirror hygrometry

In a Royal Society Bakerian Lecture in 1946, Dobson, Brewer and Cwilong (1946) described an instrument similar to a chilled mirror hygrometer for accurately determining the frost point. They wrote that the cooled surface of the instrument is too cold when the frost condensate is growing and too warm when it is shrinking, whereas the surface temperature provides a measure of the atmospheric frost point when the frost condensate does not change. In practice, they used the average of the two temperatures at which the condensate "just increases" and "just decreases", and noted further that given the low temperatures in

the UT/LS and the corresponding low $H_2O$ vapor pressures the growth/shrinking of the frost is a slow process. They found that this method is more accurate than the classic visual inspection of the dew point method, where the temperature is measured when the condensate first appears and then completely disappears, rather than when it grows/shrinks. This is because frost occurs only after the surface has cooled to 2 - 3 K below the frost point, i.e., massive supersaturation is required for frost formation, typically resulting first in the formation of supercooled dew, from which the ice then nucleates.

### 3.1 Definition of Golden Points

Manual operation and visual detection have since been replaced by automatic sensitive measurements with photo-electric detection of frost on a mirror and servo-loop mirror temperature control, but the fundamental principle of chilled mirror hygrometry remained the same: when the condensate grows, the temperature of the condensate is too low, when it shrinks, the temperature is too high. Therefore, an automatic frost point hygrometer attempts to control the heat load on its mirror such that

the condensate neither grows nor shrinks (Barrett and Herndon, 1951; Mastenbrook and Oltmans, 1983). However, to stabilize the condensate over time and properly control the instrument under rapidly changing conditions during a balloon sounding, it must regularly overshoot and undershoot the frost point by oscillations in the mirror temperature. In a stable laboratory environment and with a well-tuned feedback controller, perturbations lead to controller oscillations that decrease over time and may reach final amplitudes in the order of a few milli-Kelvin. In contrast, during a balloon sounding they may reach several

K and are often not symmetric about the frost point (Vömel et al., 2016). A chilled mirror hygrometer is therefore unable to keep the condensate in static equilibrium with the partial pressure of $H_2O$ in the ambient air ($p_{H2O}$). Instead, the condensate is quasi-periodically in a transient equilibrium with the gas phase $p_{H2O}$, namely only at the precise moment when the condensate changes from growing to shrinking or from shrinking to growing. At this moment, a chilled mirror hygrometer measures the true frost point.

We name all those measurements, where the condensate is in a transient equilibrium with the gas phase, the *'Golden Points'* of chilled mirror hygrometers. Golden Points occur ubiquitously in all chilled mirror hygrometer measurements, regardless of whether they are cooled cryogenically (e.g. Mastenbrook and Oltmans, 1983) or thermo-electrically (e.g. Fujiwara et al., 2003), and whether their feedback measurement for condensate growth is specular reflectance (e.g. Vömel et al., 2007a), light scattering (e.g. Brewer et al., 1951), phase/frequency shifts of a surface acoustic wave (e.g. Hansford et al., 2006) or energy

attenuation of alpha radiation (e.g. Rohrbough et al., 1967).



Golden Points are equilibrium points. Formally, the relationship for the Golden Points reads

$$T_m = T_{fp} \iff \frac{dU_m}{dt} = 0 \; , \tag{1}$$

where $T_m$ is the mirror temperature (an accurate representation of the condensate temperature), $T_{fp}$ is the frost point of the ambient air, $t$ is the time, $U_m$ is the feedback variable for the amount of condensate, and $dU_m/dt$ is the measure of condensate growth. In the case of CFH, $U_m$ is the mirror reflectance given by the voltage of the photodetector (in V), which in turn controls the mirror temperature $T_m$. We call Eq. 1 the "Golden Points equation".

Automatic chilled mirror hygrometers usually try to maintain their feedback variable $U_m$ as close as possible to a predefined "setpoint" $U_{\text{set}}$, which is determined by a constant reflectance chosen to avoid a sudden loss of condensate while preserving good sensitivity to changes in condensate thickness. A common misconception is that $U_m = U_{\text{set}}$ at a specific point in time implies equilibrium (i.e. $T_m = T_{fp}$) at that same point in time. This is generally not true, as $T_m = T_{fp}$ applies if and only if $dU_m/dt = 0$, which can occur far from the setpoint. Consequently, good chilled mirror instruments should have an optically sensitive ice film and measure mirror temperature and mirror reflectance simultaneously at high frequency, high accuracy, low noise and low reflex sensor drift. Hence, the requirements are a good measurement of $dU_m/dt$, sufficiently small $|dU_m/dt|$ and small $|dT_m/dt|$ with high temporal (i.e., altitude) resolution. This is because excellent accuracy of the frost point measurement can only be achieved if the position of the Golden Point along the balloon trajectory is also precisely determined with sufficiently small and well-quantified time offset between the temperature and reflectance time series. The accuracy is therefore not determined solely by the precision of the temperature sensor. Due to the large amplitude of the nonequilibrium oscillations of the controller, errors in the height determination of a Golden Point along a profile of only a few meters (resulting from timing uncertainty) can easily lead to errors in the $H_2O$ mixing ratio of well over 4 %. To meet the low noise condition, smoothing is often required. CFH typically requires smoothing of the reflection signal of 3 s for the liquid film and 15 s for the ice films (which takes account of changes in condensation rate and film morphology). The smoothing complicates determining the exact timing of the Golden Point and, thus, needs to be applied with care in order not to reduce the accuracy of the frost point $T_{fp}$.

Reporting the Golden Points to achieve maximum accuracy is not new: the first accurate measurements of stratospheric water vapor performed by Dobson and colleagues (1946) were essentially the Golden Points of their manually operated hygrometer. In that respect they are, to the best of our knowledge, the inventors of the Golden Points method. In the age of fully automated chilled mirror hygrometers, however, the method and its benefits fell into oblivion. In this paper, we recall this method and, in Section 4, present a new retrieval protocol based on a generalization of Eq. 1, which enables automated high-resolution balloon-borne frost point measurements in the UT/LS of unprecedented accuracy.



## 3.2 Application of the Golden Points method

A first example of using the Golden Points method is shown in Fig. 2, which compares the humidity measurements in the midlatitude tropopause region of three instruments: the CFH chilled mirror, the FLASH-B Ly-α fluorescence instrument and the thin-film humidity sensor of the RS41 radiosonde. Measuring the fine structure of UT/LS water vapor with high accuracy is still a challenge even for state-of-the-art instruments, especially around the tropopause, where an abrupt transition from moist tropospheric air to very dry stratospheric air takes place. Near the tropopause, the differences between the instruments

shown in Fig. 2 are more than 30 % in $H_2O$ mixing ratio, which is far from the target of < 4 % set by CIMO, let alone the target of 2 % set by GCOS. We show in this section that we can reduce these discrepancies to < 15 % when applying physically meaningful corrections to the thin-film and Ly-α instruments, based on the Golden Points of the chilled mirror instrument. As we will show in Section 5.2, we can further reduce the discrepancies to ~ 5 %, when CFH controller instabilities leading to strong nonequilibrium states of the mirror are corrected by means of a nonequilibrium correction of CFH.

Figures 2B and 2D show the original data of the three instruments as thick black lines for CFH and as green dashed lines for the other two instruments. Both FLASH-B and RS41 suggest a much drier upper troposphere and lower stratosphere than CFH. However, the extreme values of CFH mirror reflectance ($dU_m/dt = 0$) indicate seven Golden Points in the selected altitude range, whose altitudes can be determined with sufficient accuracy (within better than ± 5 m). At these points, CFH measured the frost point nominally with less than 0.2 K error. This corresponds to an uncertainty in the water mixing ratio at frost point

temperatures around 205 K and a pressure around 200 hPa of less than 0.5 ppmv (or 3 % in mixing ratio). Deviations in the original data of the other two instruments of several ppmv as shown in Fig. 2 therefore exceed the uncertainty at the Golden Points by almost an order of magnitude.

Interestingly, FLASH-B in Fig. 2B (FLASH instrument "FI05/12") can be corrected to match the Golden Points by assuming a constant offset (78.5 counts/s) in the fluorescence count rate prior to multiplication with the calibration constant (0.0256

ppmv/(counts/s)). As discussed in Section 2.1.3, FLASH-B requires recalibrations between flights, a large part of which can be corrected by a constant offset in fluorescence counts (Poltera, 2022). Here, we determine the size of this offset correction from the CFH Golden Points. We find that reducing the fluorescence counts by 78.5 counts/s (compared to typical count rates in the tropopause region of a few hundred counts/s) results in near perfect agreement with the CFH water vapor mixing ratio at all seven Golden Points. The presence of such a signal offset is puzzling. While it is known (Sitnikov et al., 2007) that the

airplane version of the instrument, FLASH-A, which resides in a closed compartment, must be corrected for stray light, resulting in an offset, the balloon instrument measures in an open configuration with downward-facing optics. Therefore, no offset would be expected (Fahey et al., 2014). However, the offset estimated here (78.5 counts/s) must be considered appropriate for this particular flight, as also confirmed by a FLASH PI (Sergey Khaykin, personal communication, 2021). This suggests



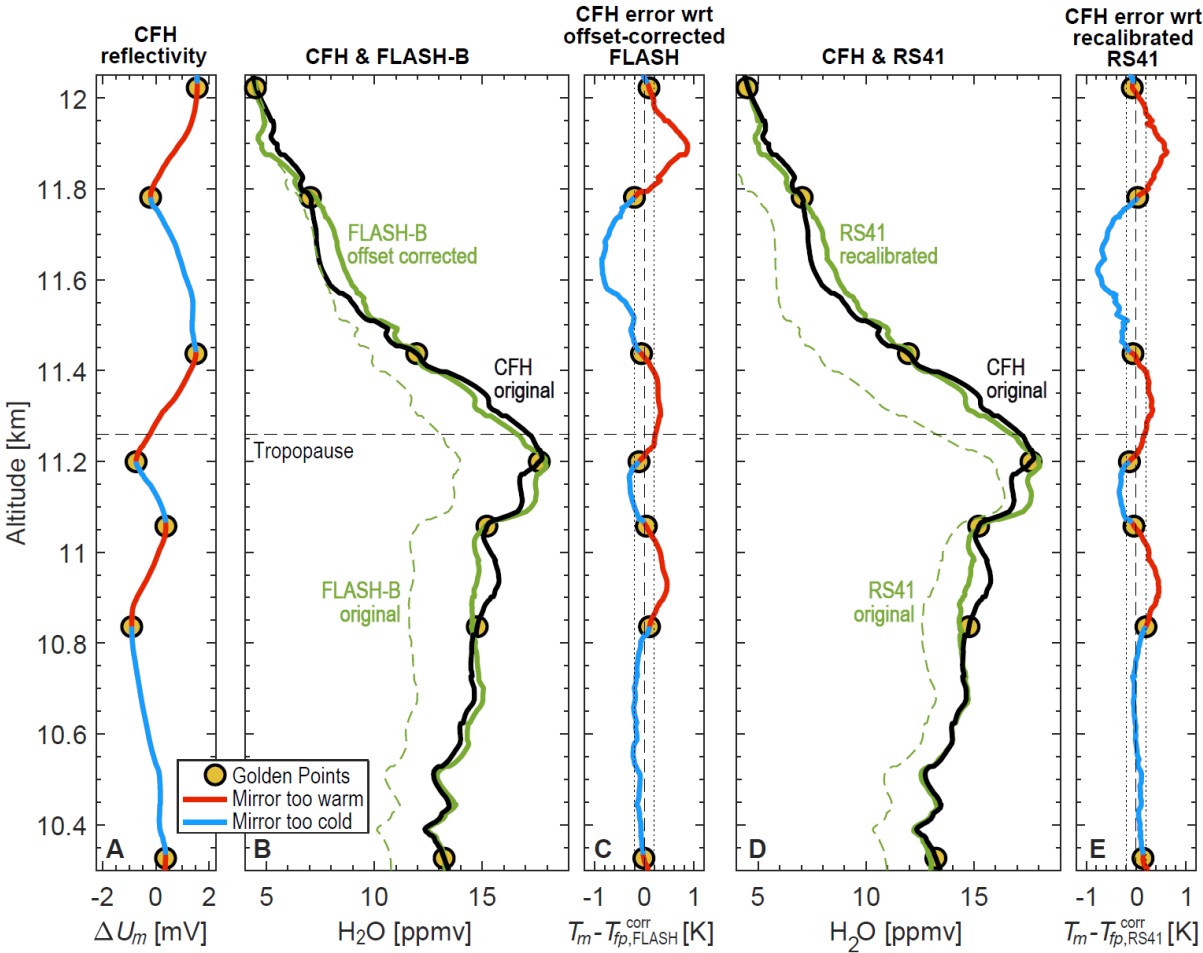

**Figure 2.** Sounding of the tropopause region over Lindenberg, eastern Germany, on 13 March 2017, 23 UT. Humidity was measured by three hygrometers on the same payload: CFH chilled mirror, FLASH-B Ly-α fluorescence instrument, and the thin-film polymer sensor of the RS41 radiosonde. **(A)** Mirror reflex signal (detector voltage in mV) with the Golden Points located at the extreme values, where $dU_m/dt = 0$. **(B)** Original data of CFH (thick black curve) and FLASH-B (thin dashed green curve) as well as offset-corrected FLASH-B data (thick green curve). **(C)** Error of CFH frost point data with respect to the offset-corrected FLASH-B. **(D)** Original data of CFH (thick black curve, same as in panel B) and RS41 (thin dashed green curve) as well as time-lag and bias-corrected RS41 data (thick green curve). **(E)** Error of CFH frost point data with respect to the time-lag and bias-corrected RS41.

that the standard calibration of FLASH-B, which assumes a simple proportionality of $H_2O$ mixing ratio and count rate at 50 hPa, was inadequate for the measurements shown in Fig. 2. We identified two other flights where offset corrections of the fluorescence signal were required, albeit to a much lesser extent than in Fig. 2 (about 2.5 counts/s (with calibration constant 0.1812 ppmv/(counts/s)) for the FLASH instrument "FI07/49" flown in Ny-Ålesund on February 21, 2020, and about 27.1 counts/s (with calibration constant 0.0318 ppmv/(counts/s)) for "FI05/16" on October 7, 2021, in Lindenberg, resulting in



errors of up to 10 % and 20 %, respectively, see Fig. 3.9 of Poltera (2022)). Nevertheless, it is currently unclear whether an offset correction is only required for certain specimens of this instrument or is a general property of FLASH-B.

RS41 typically requires corrections of the time-lag and bias of the thin polymer film measurement. Very often the RS41 shows good agreement with CFH in the lower and middle troposphere. At the low relative humidities and low temperatures found in the UT/LS, the RS41 generally still remains sensitive to moisture, but residual time-lag and bias errors, alongside sonde-to-sonde variability, decrease the accuracy of the RS41 measurement (Poltera, 2022). Therefore, we correct the Vaisala output

by performing the following steps (see Section 4.2 for details):

- we apply a time-lag correction to the raw data of relative humidity from RS41 according to Miloshevich et al. (2004), using a 2-parameter fit for a time constant of Arrhenius type (see Eq. 9);

- we smooth the time-lag corrected data with a boxcar filter (in this case with a full width of ± 10 s corresponding to about ± 50 m in altitude);

- we apply a bias correction using a 6-parameter fit that minimizes the root mean square error (RMSE) between the frost points determined from the time-lag corrected and smoothed RS41 data and the CFH Golden Points (see Eq. 10).

This procedure shifts the RS41 data from the profile originally provided by the Vaisala software (green dashed curve in Fig. 2D) to the time-lag and bias corrected profile (green solid curve in Fig. 2D). Thereafter, at the Golden Points the remaining error between the corrected RS41 and CFH frost points remains below ± 0.2 K (vertical dotted lines in Fig. 2E).

While the agreement between CFH on the one hand and the offset-corrected FLASH-B or the time-lag and bias corrected RS41 on the other hand is excellent at the Golden Points, there are still discrepancies up to ± 0.9 K away from the Golden Points (blue and red lines in Fig. 2C, 2E). As we will argue below, these remaining discrepancies are due to the CFH controller oscillations. As detailed in the next section, they can be further decreased to ± 0.4 K by applying a nonequilibrium correction to the CFH data.

**4    Reflectance-based nonequilibrium correction away from the Golden Points**

This section presents details on the nonequilibrium correction, first introduced by Poltera et al. (2021, 2022) and Poltera (2022). By adjusting the mirror temperature, $T_m$, chilled mirror hygrometers aim at establishing equilibrium between the $H_2O$ partial pressure in the ambient air and the $H_2O$ vapor pressure of the condensate (frost or dew). The vapor pressure satisfies the Clausius-Clapeyron equation $dp_{vap}/p_{vap} = \Delta H_{vap}/R \cdot dT/T^2$, where $\Delta H_{vap}$ denotes the heat of evaporation ($\approx$ 51 kJ/mol for ice)

and $R$ the universal gas constant ($\approx$ 8.3145 J/(K mol)). For constant $\Delta H_{vap}$, the Clausius-Clapeyron equation has the Arrhenius-type solution $p_{vap} \propto \exp(-\Delta H_{vap}/RT)$. When the temperature of the mirror is above the frost (or dew) point, sublimation (evaporation) occurs, and when it is below the frost point, deposition (condensation) occurs. Only when it is exactly at the frost (or dew) point, the condensate on the mirror is in equilibrium with the ambient water partial pressure, and the net flux of $H_2O$ molecules onto the surface is zero (Barrett and Herndon, 1951; Mastenbrook and Oltmans, 1983).



## 4.1 Quantifying the nonequilibrium

Chilled mirror hygrometers rely on the principle that the frost or dew condensate is in equilibrium with the $H_2O$ partial pressure in the atmosphere at a unique temperature, the frost (or dew) point. Under equilibrium conditions, the net flux of $H_2O$ molecules onto the mirror vanishes. Under nonequilibrium conditions, when the mirror is too cold, the net $H_2O$ flux towards the mirror is positive, the condensate grows, and the mirror reflectance decreases ($T_m > T_{fp} \Rightarrow dU_m/dt < 0$). Conversely, when the mirror is too warm, the net $H_2O$ flux is negative, the condensate shrinks, and the mirror reflectance increases ($T_m < T_{fp} \Rightarrow dU_m/dt > 0$). While detecting equilibrium according to Eq. 1 is straightforward, it is a corollary of a more general nonequilibrium equation. Quantifying the nonequilibrium relies on three assumptions, which we describe in the following subsections.

### 4.1.1 Assumption I: linear response of the mirror reflectance to the $H_2O$ molar flux

The relationship $U_m(M_{H_2O})$ between the amount of specularly reflected light given by the voltage $U_m$ (in V) and the amount of condensate on the mirror $M_{H_2O}$ (in mol/m²) is a monotonically decreasing function. We assume that the relationship is linear over a small range of mirror reflectances around the setpoint $U_{set} \pm \Delta U$ (Barrett and Herndon, 1951):

$$\frac{dU_m}{dt} = \frac{dU_m}{dM_{H_2O}} \cdot \frac{dM_{H_2O}}{dt} \equiv -A' \cdot \frac{dM_{H_2O}}{dt} \ . \tag{2}$$

We term $A' = -dU_m/dM_{H_2O}$ the "morphological sensitivity" of the mirror condensate (in V/(mol/m²)), which depends on the morphology of the condensate, e.g. ice crystal size and shape. Under many circumstances, within a flight segment, $A'$ is approximately constant. For CFH, $\Delta U = |U_m - U_{set}| < U_{set}/100$ for ice condensates and $\Delta U = |U_m - U_{set}| < U_{set}/10$ for liquid water condensates, with $U_{set} = 2.5$ V. The assumption that $A'$ is constant means that the morphology of the condensate near the setpoint remains stable over time, whereas abrupt morphology changes, such as the tearing off of pieces of the ice film violate this assumption. A similar linear relationship near the setpoint is assumed for chilled mirror hygrometers using light scattering from the condensate instead of specular reflectance from the mirror as feedback variable, such as SKYDEW (Sugidachi, 2014; Sugidachi et al., 2025).

### 4.1.2 Assumption II: net H2O flux is governed by diffusional law

A thin layer of thickness $\Delta x_{diff}$, Prandtl's boundary layer, is located above the mirror and its condensate, in which molecular gas phase diffusivity determines the time scale of mass transfer between the flowing air and the condensate. It can be approximated as $\Delta x_{diff} = \Delta x_{diff}(L,Re) \approx \frac{5}{3}L/Re^{1/2}$ (Schlichting and Gersten, 2017), where $L$ is the distance from the mirror edge to its center. We further simplify this to $\Delta x_{diff} \approx 150 \ \mu m \times (\rho(0)/\rho(z))^{1/2} \times (v_0/v)^{1/2}$, where $\rho(z)$ is the air density at altitude $z$ and $\rho(0) \approx 1.3$ kg/m³, $v$ is the balloon vertical velocity and $v_0 = 5$ m/s. For most of the applications within the upper troposphere, $\Delta x_{diff} \approx 250 \ \mu m$ is a sufficient approximation. Fick's first law (1855) for the $H_2O$ molecular flux onto the mirror reads (Taylor and Krishna, 1993):





$$\frac{dM_{H_2O}}{dt} = \frac{D_{H_2O}(\bar{p}, \bar{T})}{\Delta x_{diff} \cdot R\bar{T}} \cdot \left( p_{H_2O} - p_{vap}(T_{ice}) \right) , \qquad (3)$$

where $dM_{H_2O}/dt$ is the $H_2O$ molar flux onto the mirror, $D_{H2O}$ is the diffusion coefficient of $H_2O$ molecules in air (in m²/s), $R$ is the ideal gas constant, $\Delta x_{diff}$ is the boundary layer thickness, $p_{H_2O}$ is the $H_2O$ partial pressure in the ambient air and $p_{vap}(T_{ice})$ the $H_2O$ vapor pressure of the ice film on the mirror with temperature $T_{ice}$ (both in Pa), or of the liquid in the case of dew. Furthermore, $\bar{p}$ and $\bar{T}$ are the average pressure and temperature in the diffusion layer between the ice surface and the ambient air (Taylor and Krishna, 1993), taken here as $\bar{p} \approx p_{air}$ and $\bar{T} \approx T_{air}$. Finally, note that the $H_2O$ partial pressure $p_{H_2O}$ satisfies $p_{H_2O}$

$= p_{vap}(T_{fp})$.

### 4.1.3    Assumption III: mirror temperature equals condensate temperature

Our last assumption is that the measured mirror temperature $T_m$ is an accurate representation of the condensate temperature $T_{ice}$ with negligible time-lag:

$$T_{ice} = T_m . \qquad (4)$$

Vömel et al. (2016) estimated temperature inhomogeneities to be less than 0.1 K, owing to the high heat conductivity (about 400 Wm⁻¹K⁻¹) of the gold-plated copper mirror of CFH (Vömel et al., 2007a) and the only few micrometers thick ice crystals on the mirror (Leu and Kayser, 2009) after the CFH cleaning cycles at -15°C resp. -53°C frost point. Convective heat transfer from the flowing air to the mirror is not negligible and must be taken into account in the implementation of the PID controller (Vömel and Jeannet, 2013). Nevertheless, the temperature of the mirror surface and the ice film equilibrate within milliseconds

when the total heat load changes. Therefore, inhomogeneities in the mirror temperature and uncertainties of the thermistor are covered by the overall mirror temperature uncertainty of at most 0.11 K (Vömel et al., 2016).

### 4.1.4    The nonequilibrium equation

When the deviation from the mirror reflectance setpoint (2.5 V for CFH) is sufficiently well constrained by the micro-controller, the change in mirror reflectance is proportional to the net flux of $H_2O$ molecules with a timescale governed by diffusive

transport in the thin layer over the mirror. Then, combining Eqs. 2-4 yields the "nonequilibrium equation":

$$\frac{dU_m}{dt} = A' \frac{D_{H_2O}(\bar{p}, \bar{T})}{\Delta x_{diff} \cdot R\bar{T}} \cdot \left( p_{vap}(T_m) - p_{vap}(T_{fp}) \right) , \qquad (5)$$

Equation 5 is the quantification of nonequilibrium conditions on the mirror. If and only if the mirror reflectance assumes an extreme value (i.e., a local maximum or minimum, $dU_m/dt = 0$), the mirror temperature is at the frost (dew) point ($T_m = T_{fp}$ or $T_{dp}$) and the nonequilibrium vanishes, i.e., Eq. 5 relaxes to the Golden Point equation.





At the Golden Points, the mirror temperature is the most accurate estimate of the true atmospheric frost (dew) point within the
sampling period of $\pm 1$ s. In contrast, between the Golden Points the mirror is in a nonequilibrium state. Equation 5 provides a
fundamental mathematical relationship between mirror temperature and mirror reflectance:

-   When the mirror reflectance increases ($dU_m/dt > 0$), the mirror coverage decreases, the condensate evaporates, the mirror
    is too warm (i.e. the mirror temperature experiences a warm 'excursion').

-   When the mirror reflectance decreases ($dU_m/dt < 0$), the mirror coverage increases, the condensate grows, the mirror is
    too cold (i.e. the mirror temperature experiences a cold 'excursion').

-   When the mirror reflectance passes through a minimum ($dU_m/dt = 0$, $d^2U_m/dt^2 > 0$), the condensate changes from growth
    to evaporation as the mirror temperature is crossing the frost (dew) point.

-   When the mirror reflectance passes through a maximum ($dU_m/dt = 0$, $d^2U_m/dt^2 < 0$), the condensate changes from evap-

oration to growth as the mirror temperature is crossing the frost (dew) point.

-   If the reflectance of the mirror remains constant, the condensate neither grows nor evaporates, and the mirror temperature
    is the frost (dew) point, a condition that rarely exists due to the constantly changing atmospheric conditions.

During a balloon sounding, chilled mirror hygrometers with calibrated mirror temperature sensors and reflectance measure-
ments with good signal-to-noise ratio pass through a number of Golden Points with accurate $T_{fp}$ measurements. The number

of Golden Points depends on:

-   the performance of the feedback controller (fast vs. slow response)

-   the airflow through the inlet tube (good vs. poor ventilation)

-   the morphology of the condensate (fine vs. coarse frost)

-   the state of the atmosphere (slowly vs. rapidly varying frost point).

For CFH, the temporal sequence of Golden Points typically ranges from a few seconds in the lower troposphere to a few tens
of seconds in the UT/LS. As we will see in the case studies in Section 5, the Golden Points enable the in-situ/flight recalibration
of other sensors flown in tandem with CFH, such as the RS41, which then allows applying a nonequilibrium correction to the
CFH data.

### 4.1.5   Linearized form of the nonequilibrium correction

In order to improve applicability in control theory with the heat balance equation ($dT_m/dt \propto$ applied heat) and facilitate the use
for frost point hygrometers, which measure temperatures, we rewrite the nonequilibrium equation (Eq. 5) by means of the
linearized form of the Clausius-Clapeyron equation, $p_{\mathrm{vap}}(T_m) - p_{\mathrm{vap}}(T_{fp}) \approx \Delta H_{\mathrm{vap}}/R \cdot p_{\mathrm{vap}}(T_m)/T_m^2 \cdot (T_m - T_{fp})$, valid for $\left| T_m - T_{fp} \right|$
smaller than $1 - 2$ K, to obtain a linearization of $dU_m/dt$ in terms of the temperature deviation:

$$\frac{dU_m}{dt} \approx A' \cdot \frac{D_{\mathrm{H_2O}}(\bar{p}, \bar{T})}{\Delta x_{\mathrm{diff}} \cdot R\bar{T}} \cdot \frac{\Delta H_{\mathrm{vap}}}{R} \cdot \frac{p_{\mathrm{vap}}(T_m)}{T_m^2} \cdot \left( T_m - T_{fp} \right) . \tag{6}$$





Once the morphological sensitivity, $A' = -dU_m/dM_{H_2O}$, of the mirror condensate is known, Eq. 6 can be used to quantify the temperature error of the mirror, $T_m - T_{fp}$, as all the other terms in this equation are known physical constants or measured quantities. The least well-known quantity is the thickness of the boundary layer, which during balloon ascent in the upper troposphere is typically $\Delta x_{diff} \approx 250$ μm (see Section 4.1.2). However, $\Delta x_{diff}$ can be thicker in situations of poor ventilation ($\Delta x_{diff} \propto v_{flow}^{-1/2}$), such as under prelaunch conditions, or thinner in situations of strong ventilation, such as directly after the

burst of the balloon.

We can further simplify Eq. 6 by approximating the gas phase diffusion coefficient of $H_2O$ as $D_{H_2O}(p,T) \approx D_0 \cdot (p_0/p) \cdot (T/T_0)^2$, where $D_0 = 0.21$ cm$^2$/s, $p_0 = 1013$ hPa and $T_0 = 273$ K (Pruppacher and Klett, 2010). This yields

$$\frac{dU_m}{dt} \approx A \cdot \left( \frac{\bar{T}}{T_m^2} \cdot \frac{p_{vap}(T_m)}{\bar{p}} \right) \cdot (T_m - T_{fp}) \approx A \cdot \frac{\bar{T}}{T_m} \cdot \chi_{H_2O}(T_m) \cdot \frac{T_m - T_{fp}}{T_m} , \qquad (7)$$

where $A$ is the "effective sensitivity". It is a constant, $A = A' \cdot D_0 \cdot p_0 \cdot \Delta H / (\Delta x_{diff} \cdot (RT_0)^2) \approx A' \cdot 84$ mol/(m$^2$s) in V/s, which combines

the morphological sensitivity $A'$ (in V/(mol/m$^2$)), the standard diffusion coefficient $D_0$, the thickness of Prandtl's boundary layer ($\Delta x_{diff} \approx 250$ μm), the heat of evaporation of ice ($\Delta H_{vap} \approx 51$ kJ/mol) and the universal gas constant ($R = 8.3145$ J/(K mol)). Furthermore, $\chi_{H_2O}(T_m) = p_{vap}(T_m)/\bar{p}$ is the $H_2O$ mixing ratio estimated from the mirror temperature. The correction of the mirror temperature is then easily written as

$$T_{fp}(t) \approx T_m(t) - \frac{B(t)}{A} \cdot \frac{dU_m}{dt}(t) , \qquad (8a)$$

where $B(t) = T_m^2/(\bar{T} \cdot \chi_{H_2O}(T_m))$ in K is a function of temperatures and of the water vapor mixing ratio.

Equation 8a allows to perform a nonequilibrium correction of the CFH measurement between the Golden Points. However, this requires knowledge of the morphological sensitivity $A' = -dU_m/dM_{H_2O}$, which can differ by more than an order of magnitude from flight to flight and even changes after each CFH cleaning cycle (as shown in Section 6.1). Different $A'$ reflect large variability in ice film morphology due to slightly different conditions at the time of ice nucleation and growth (and

possibly sintering) on the mirror following the cleaning cycles. Therefore, obtaining $A'$ typically requires measurements by a second, independent instrument on the same platform, such as RS41 or FLASH, to establish an in-flight nonequilibrium correction of CFH.

However, there are a number of causes of residual errors that are not corrected by the nonequilibrium correction, even when excluding measurements affected by contamination of the inlet tube by hydrometeors or outgassing of the balloon skin, or by

strong electronic interference. Such residual errors include: (i) electronic delays or data transmission problems, which are usually small; (ii) uncertainties in $dU_m/dt$ due to electronic noise, which must be reduced by smoothing, e.g. with a 21-s half-width Savitzky-Golay filter; (iii) uncertainties in $dU_m/dt$ due to stray light, which can be largely avoided by flying only at



night; (iv) abrupt jumps in $U_m$ due to break-off of parts of the ice film, deposition of cloud particles on the mirror, or the sudden onset of specular reflection of individual crystal faces, all of which occur very rarely; (v) an uncertainty in $A$ due to a lack of

responsiveness of the second sensor to humidity changes, which is regularly the case for RS41 when the relative humidity above the tropopause drops below 2 %RH, so that the correction of CFH in the stratosphere must rely on the sensitivity $A$ determined for the second ice film in the upper troposphere. While point (v) may appear to be a major drawback in using RS41-derived sensitivities $A$ to correct CFH in the stratosphere, this work will show below that $A$ derived in the upper tropo-sphere can be used very well deep in the stratosphere (provided the thickness of the Prandtl Boundary layer, $\Delta x_{\mathrm{diff}}$, is adapted

appropriately, see Section 5.5.1).

## 4.2  Using a recalibrated RS41 for the nonequilibrium correction of CFH

Throughout the troposphere and stratosphere, the CFH measurements are excellent at the Golden points, i.e. $T_{fp}$ accurate to $\pm 0.2$ K, corresponding to errors in $H_2O$ vapor pressure or mixing ratio of less than 3.5 % (except where measurements are contaminated (Jorge et al., 2021)). By contrast, nonequilibrium states of the controller may cause much larger errors just a few

meters above or below the Golden Points. Realizing the CIMO and GCOS goals requires measurements of the atmospheric frost point with low uncertainty and high vertical resolution. For this, we need to correct the nonequilibrium errors between the Golden Points, which can be achieved by a second sensor on the same payload, e.g., the humidity sensor of the RS41. As we will discuss in Section 4.4, under certain circumstances, correction can be achieved without the help of other sensors, i.e. by means of a self-correction of the frost point hygrometer. However, for CFH with their widely spaced Golden Points, this is

usually not possible. Instead, another well recalibrated sensor is needed to correct the measurement errors.

Using an RS41 for this purpose, we apply the nonequilibrium correction according to the following scheme:

(1)   identify the Golden Points along the humidity profile measured by a chilled mirror instrument;

(2)   construct an a-priori estimate of the frost point profile by means of an RS41 on the same balloon payload, which is

-   synchronized to the CFH telemetry (i.e. correcting for a mismatch of typically about $\Delta_{\mathrm{tele}} = 3.5$ s (Holger Vömel,

pers. comm., 2019; Poltera, 2022);

-   time-lag-corrected by means of a 2-parameter Arrhenius-type equation ($\sim \exp(\Delta E/RT)$) for the time constant, see Eq. 9 below, which captures the physics of the instrument better than polynomial or exponential-type fits used previ-ously;

-   and bias-corrected by means of the Golden Points of the chilled mirror instrument using 6 parameters (in the form

of an offset and a scaling-factor motivated by ice chamber laboratory experiments described in Miloshevich et al. (2001)), see Eq. 10;

(3)   use this a-priori estimate to determine the effective sensitivity $A$;

(4)   construct an a-posteriori by correcting the CFH mirror temperature using $A$ in Eq. 8a, called "nonequilibrium corrected CFH".




The a-posteriori is our best approximation for the "true" frost point. We use the deviation between the a-priori and the a-posteriori as an estimate of the uncertainty of the frost point resulting from residual errors in the nonequilibrium correction of CFH and in the time-lag and bias correction of the RS41 data based on the CFH Golden Points.

The effective sensitivity $A$ can then be determined from a least squares fit using the scaled time derivative of the mirror reflectance $B \cdot dU_m/dt$ as function of the difference of time-lag and bias-corrected RS41 data, $T_{fp,\mathrm{RS41}}^{\mathrm{corr}}$ and the CFH mirror temperature $T_m$:

$$B \cdot \frac{dU_m}{dt} \approx A \cdot \left( T_m - T_{fp,\mathrm{RS41}}^{\mathrm{corr}} \right). \tag{8b}$$

Note that Eq. 8b is equivalent to Eq. 8a, but with the still unknown true frost point $T_{fp}$ approximated by the known time-lag and bias corrected RS41 ($T_{fp,\mathrm{RS41}}^{\mathrm{corr}}$). Figure 3D below shows a textbook example of the determination of $A$ immediately before the launch of a balloon sonde (for the liquid film, i.e. with $T_{dp,\mathrm{RS41}}^{\mathrm{corr}}$ instead $T_{fp,\mathrm{RS41}}^{\mathrm{corr}}$), resulting in $A$ = 773 V/s. In Section 6, we estimated $A$ in flight for 70 soundings from the tropics to the polar regions, and found for the liquid film and the first and second ice films a median value of $A \approx 6000$ V/s, albeit with a high degree of variability (700-34000 V/s, see Fig. 11). An effective sensitivity of 6000 V/s is a canonical value for a good ice film and translates to a morphological sensitivity $A' \approx 71.4$ V/(mol/m$^2$) $\approx 40$ µV/(ng/cm$^2$) under typical balloon ascent ventilation conditions. This corresponds to about 1.2 mV per $H_2O$ molecular monolayer. With detectable changes (at S/N = 2) in CFH reflectance of about 20 µV/s during nighttime flights, this is comparable to precision mass balances (Ward and Buttry, 1990) and translates to an absolute sensitivity of CFH of about 0.1 ppmv or about 0.1 K under good measurement conditions, a value consistent with the 0.11 ppmv average precision at 2σ reported for CFH under laboratory conditions representative of the UT/LS (Fahey et al., 2014).

### 4.2.1 Time-lag correction for RS41

The RS41 humidity measurement can be used to construct the CFH nonequilibrium correction, even though it is often less accurate in the upper troposphere and in the stratosphere (e.g. Brunamonti et al., 2019; Sun et al., 2021; Lee et al.; 2021; Khordakova et al., 2022) and fails to reproduce the CFH Golden Points, as shown by the green dashed in Fig. 2C. Therefore, RS41 must first be corrected for time-lag and bias by means of the Golden Points, before it can be used to correct the nonequilibrium states of CFH. This utilizes the high precision of the corrected RS41 data.

The capacitive RS41 humidity sensor operates on the principle of the change in the electrical capacitance of a thin polymer film due to moisture absorption. When atmospheric humidity changes, the $H_2O$ molecules must diffuse into or out of the polymer film, which takes time, causing the sensor to react with a diffusive delay and with a damped response to changes in the external humidity. This is best described by in terms of an Arrhenius-type temperature dependence (e.g. Denton et al., 1985) of the characteristic time for diffusion through the layer



$$\tau_{\text{RS41}}(T) = \tau_\infty \cdot \exp\left(\frac{E_a}{RT_{\text{RS41}}}\right) , \tag{9}$$

where $\tau_{\text{RS41}}$ is the corresponding time constant, $T_{\text{RS41}}$ is the temperature of the RS41 humidity sensor in K (aimed to be at 5 K above ambient temperature), $E_a$ is the diffusion activation energy (in J/mol) and $\tau_\infty$ is a pre-factor (in s). The Arrhenius description ($\sim\exp(\Delta E/RT)$) captures the physics of the instrument better than a polynomial dependence on temperature, as previously assumed by Miloshevich et al. (2004) or a simple exponential dependence ($\sim\exp(-cT)$) used by Vaisala (Antikainen and Paukkunen, 1994) and the research community (Leiterer et al., 2005; Dirksen et al., 2014). In particular, at low temperatures, the Arrhenius dependence provides a more realistic description and appears to better reproduce recent laboratory measurements at temperatures lower than previously tested (Von Rohden et al., 2021, 2022; see also Dirksen et al. (2020)). Since $E_a$ and $\tau_\infty$ may vary slightly between different copies of RS41 due to small differences in design, such as film thickness, we fit Eq. 9 individually for each flight along with other parameters and use this to correct RS41 for its time-lag. The equation for time-lag deconvolution is found in Miloshevich et al. (2004), their Eq. 4.

### 4.2.2 Bias correction for RS41

The bias is corrected by means of the 6-parameter fit mentioned in Section 3.2, namely in terms of an offset $S_0$ and a scaling factor $\xi$, following the rationale of the correction of the temperature dependence in ice chamber laboratory experiments applied by Vaisala to their radiosonde humidity sensors in the past (Wang et al., 2002; Miloshevich et al., 2001; 2004):

$$\begin{aligned} S_{\text{corr}}(S,T) &= (S - S_0) \times \xi \\ &= \left(S - (a_0 + a_1 T_{\text{RS41}} + a_2 T_{\text{RS41}}^2)\right) \times (b_0 + b_1 T_{\text{RS41}} + c_0 \cdot S) , \end{aligned} \tag{10}$$

where $S$ is the $H_2O$ saturation ratio (with respect to liquid water or ice, depending on the condensate on the mirror) measured with RS41 (after time-lag correction), $T_{\text{RS41}}$ is the temperature of the RS41 humidity sensor (ambient + 5 K), and $a_i$, $b_i$ and $c_0$ are six bias correction parameters. Equation 10 is also fitted individually for each flight together with other parameters and this is used to correct RS41 for its bias. This procedure allows the time-lag and bias corrected RS41 profile to match CFH's Golden Points. In addition, it provides information on the behavior of the RS41 in environments that are challenging to reproduce in the laboratory, such as the UT/LS.

### 4.2.3 Details of the implementation of the nonequilibrium correction

To account for changes of the film morphology after each cleaning cycle of each sounding, we subdivide each flight into three segments: the liquid film segment (from balloon launch to the first CFH cleaning cycle at $T_m = -15$ °C), the segment with the first ice film (from the first to the second cleaning cycle), and the segment with the second ice film (from the second cleaning cycle at $T_m = -53$ °C to the hygropause). We conservatively limit the analysis to below the hygropause, since the detection limit of 1–2 %RH of the RS41 polymer humidity sensor (e.g. Brunamonti et al., 2019; Vömel et al., 2022) was reached just above the hygropause in some of the Lindenberg flights and because the uncertainty of the time-lag correction of capacitive





thin-film sensors is greatest during desorption from the thin-film polymer in the first kilometers above the hygropause (Dirksen
et al., 2014). Therefore, applications above the hygropause in the stratosphere must rely on the nonequilibrium parameters
obtained in the upper troposphere below the hygropause, which is justified as long as the morphology of ice film does not
change during further ascent.

For each flight segment, we extract the local minima and maxima after smoothing the reflectance signal (7-point second-order
Savitzky-Golay filter for the liquid film and for the first ice film, 31-point second-order Savitzky-Golay filter for the second
ice film). This filter aims to reduce random noise (e.g., microscopic fluctuations in ice crystal shape and size accompanied by
sudden changes in specular reflection, stray light from the ambient or from the instrument's light source, and noise of the
electronics), while not causing a time shift, i.e. keeping the Golden Points at their original position in the profile. To avoid
local extrema due to noise, neighboring maxima and minima are required to differ from each other by at least 3% of the range
of all reflectance values in the flight segment. Golden Points and portions of flight segments where the mirror is suspected to
be blank or contains only very little condensate are discarded (i.e., the reflectance must be greater than $U_m = 2.5 \text{ V} + 0.95 \cdot$
$(U_{m,\text{clean}} - 2.5 \text{ V})$, where $U_{m,\text{clean}}$ is the signal value of the blank mirror determined during instrument preparation).

We then employ a numerical optimization solver (Storn and Price, 1996) to adjust the RS41 data to best match the CFH Golden
Points. This involves fitting the following ten correction parameters for the RS41:
- the two time-lag correction parameters $\tau_\infty$ and $E_a$ (Eq. 9);
- six bias correction parameters $a_0$, $a_1$, $a_2$, $b_0$, $b_1$, $c_0$ (Eq. 10);
- the width of a boxcar filter for flight segments with ice, $N_{\text{boxcar}}$, that is used to smooth the CFH and RS41 relative humidity
data in the same manner;
- a time-delay parameter for flight segments with liquid condensate, $\Delta_{\text{delay}}$, which accounts for any sub-second mismatch
in time between mirror temperature and reflectance.

The solver is given lower and upper bounds for each of the parameters (Poltera, 2022, Table A.1), and then searches for the
parameters that best align the RS41 with the CFH data at all Golden Points within the flight segment. In the solver, the RS41
raw data of RH is first time-lag corrected (as in Miloshevich et al. (2004)) and then smoothed with the boxcar filter. The
resulting smoothed RH profile is then bias-corrected and transformed to frost point. The CFH mirror temperature is first con-
verted to RH to be comparable with the RS41 measurements; then the CFH RH is synchronized (sub-second time shifted) with
RS41 by means of $\Delta_{\text{delay}}$ (after $\Delta_{\text{tele}}$ has been taken into account), smoothed with the same boxcar filter as the RS41, then
transformed back to frost point. The formulation by Murphy and Koop (2005) is used for the saturation vapor pressure over
ice, and the ITS-90 compatible form of Wexler's formulation (Hardy, 1998) for the saturation vapor pressure over water, which
is the saturation vapor pressure of liquid water used by Vaisala for the RS41 (Survo et al., 2014). For air pressure, the RS41
GPS-based data product from Vaisala is used. The solver minimizes the RMSE (or, equivalently, the standard deviation)
between the corrected RS41 frost point ($T_{fp,\text{RS41}}^{\text{corr}}$) and the CFH frost point ($T_m$) at the location of the CFH Golden Points.



For flight segments with ice films we prescribe $\Delta_{\text{delay}} = 0$, while for flight segments with liquid films we prescribe $N_{\text{boxcar}} = 1$, because sub-second delay errors are less significant than time-lag errors at lower temperatures, while the opposite is true at higher temperatures. The solver is used to fit the remaining 9 parameters simultaneously to obtain the best match with the CFH Golden Points. From the 70 soundings discussed in Section 6, the residual fitting error is 0.1 K (1σ) for the Golden Points.

After that, the derived corrected frost point $T_{fp,\text{RS41}}^{\text{corr}}$ serves as the atmospheric reference frost point profile, from which we derive the effective sensitivity parameter $A$ of CFH. Finally, we calculate the effective sensitivity $A$ by minimizing the residual error of Eq. 8b, i.e. of $A \cdot (T_m(t) - T_{fp,\text{RS41}}^{\text{corr}}(t)) = B(t) \cdot dU_m/dt(t)$.

## 4.3 Using a recalibrated FLASH for the nonequilibrium correction of CFH

With its fast response and a measurement precision of about 5.5 %, FLASH-B is particularly suited for the study of fine-scale
processes in the UT/LS and stratosphere, such as horizontal moisture transport and overshooting convection in the tropics (Khaykin et al., 2009; 2016; Liu et al., 2010), or water vapor (re-)distribution and PSC-dehydration in the Arctic (Maturilli et al., 2006; Karpechko et al., 2007; Lukyanov et al., 2009; Khaykin et al., 2013; Engel et al., 2014). FLASH-B is, however, not an absolute water vapor instrument and needs to be calibrated before flight. The calibration procedure is described by Vömel et al. (2007b) and Lykov et al. (2011) and uses a reference-grade laboratory chilled mirror hygrometer (MBW 373L) to deter-
mine the calibration constant (in ppmv/(counts/s)), which reflects the performance (such as output intensity, window transmissivity and quantum efficiency of the photo-multiplier) of the measurement optics. FLASH-B is calibrated at CAO (Central Aerological Observatory, Russia), or, if the station has the necessary calibration equipment, such as the Lindenberg Observatory, directly at the sounding station (Lykov and Khaykin, 2018). Alternatively, in-flight calibration by comparing the FLASH-B fluorescence counts with an accurate estimate of the water vapor mixing ratio at 50 hPa (e.g. from CFH or Aura/MLS)
during balloon descent (Sergey Khaykin, personal communication, 2021), or at 300 hPa (e.g. from CFH or RS41) during balloon ascent (Peter Oelsner, personal communication, 2022) is also possible. However, large discrepancies (Dirksen, 2020) between pre-flight calibration coefficients derived in the laboratory and in-flight calibration coefficients derived from comparison with RS41 and CFH have cast doubt on the general validity of the FLASH-B calibration procedure, unless a constant offset in fluorescence count rate is taken into account (Poltera, 2022), see below.

Although the accuracy of the original FLASH-B measurement cannot reproduce the CFH Golden Points, as shown by the green dashed line in Fig. 2B, the FLASH humidity measurement can be used to construct the CFH nonequilibrium correction by means of the introduction of an offset in fluorescence count rate. This requires fewer recalibration parameters than the RS41 recalibration (2 vs. 10) and without residual time-lag uncertainty, since FLASH is an optical measurement.

### 4.3.1 Offset-correction of FLASH

The original FLASH-B equation (Lykov et al., 2017; Lykov and Khaykin, 2018) can be extended by introducing a constant offset in fluorescence count rate as additional calibration parameter:





$$S = S_{\text{offset}} + \frac{1}{K_{\text{cal}}} \cdot T_Q(p,T) \cdot T_{H_2O}(\chi_{H_2O}, p, T) \cdot T_{O_2}(p,T) \cdot \chi_{H_2O} , \tag{11}$$

where $S$ is the fluorescence counts signal (in counts/s) and $\chi_{H_2O}$ is the H$_2$O mixing ratio. The O$_2$ and H$_2$O Beer-Lambert transmission terms satisfy $\ln(T_{O_2}) = -\sigma_{O_2} \cdot 0.21 \cdot n_{\text{air}} \cdot L$ and $\ln(T_{H_2O}) = -\sigma_{H_2O} \cdot \chi_{H_2O} \cdot n_{\text{air}} \cdot L$, where $n_{\text{air}} = p/(k_B T)$ is the air molecule number density and 0.21 is the mixing ratio of oxygen. The effective Lyman-α absorption cross sections $\sigma_{O_2}$ and $\sigma_{H_2O}$ (i.e., the convolution of the line-shape emission of the lamp and the molecular absorption cross sections of O$_2$ and H$_2$O) and the Lyman-α absorption path length $L$ for FLASH-B are $\sigma_{O_2} = 1.00985 \cdot 10^{-20}$ cm$^2$, $\sigma_{H_2O} = 1.4552 \cdot 10^{-17}$ cm$^2$ and $L = 5$ cm. The quenching term is $T_Q(p,T) = Q(p,T)/Q(1000$ hPa, 0°C$)$, where $Q(p,T) = n_{\text{air}}/(1+n_{\text{air}} \cdot k_{q,\text{air}}/A_{0,0})$ with quenching rate coefficient $k_{q,\text{air}}$ and the Einstein transition probability $A_{0,0}$ taken as $k_{q,\text{air}} = 2.3 \cdot 10^{-11}$ cm$^3$/s and $A_{0,0} = 1.26 \cdot 10^6$ s$^{-1}$ (Lykov et al., 2017). Equation 11 has been solved by an approximate fit by Lykov et al. (Lykov et al., 2017), which limits the applicability to $p < 300$ hPa. However, Eq. 11 has an analytical solution, which extends the range of applicability:

$$\chi_{H_2O} = -\frac{1}{\sigma_{H_2O} \cdot n_{\text{air}} \cdot L} \cdot W_0\left(-\frac{\sigma_{H_2O} \cdot n_{\text{air}} \cdot L}{T_Q \cdot T_{O_2}} \cdot K_{\text{cal}} \cdot (S - S_{\text{offset}})\right) , \tag{12}$$

where $W_0(x)$ is the 0$^{\text{th}}$ (i.e. principal/upper) branch of the *Lambert W*-function, which solves $W_0(x) \cdot \exp(W_0(x)) = x$ (Poltera, 2022). The solution for the upper branch $W_0$ of the *Lambert W*-function is only valid when $W_0(x) \geq -1$ and $x \geq -e^{-1}$, i.e. when $\chi_{H_2O} \cdot \sigma_{H_2O} \cdot n_{\text{air}} \cdot L \leq 1$ and when $K_{\text{cal}} \cdot (S - S_{\text{offset}}) \cdot \sigma_{H2O} \cdot n_{\text{air}} \cdot L \leq e^{-1} \cdot T_Q \cdot T_{O_2}$. These inequalities provide an atmospheric lowest level limit for the validity of the solution in a particular sounding around 500 hPa (somewhat better than the usually stated limit of 300 hPa and tested for individual FLASH-B soundings (Poltera, 2022)).

The FLASH-B calibration parameters $K_{\text{cal}}$ in 1/(counts/s) and $S_{\text{offset}}$ in counts/s can be fitted by Eq. 12 within the extended measurement range of FLASH-B ($p < 500$ hPa and $\chi_{H_2O} < 1000$ ppmv), using a reference water vapor mole fraction $\chi_{H_2O} = \chi_{H_2O,\text{ref}}$. This allows for in-flight calibration of FLASH-B with a tandem hygrometer used as reference. For the soundings shown in Figs. 2 and 4, we have used the tandem CFH water vapor mixing ratio at the Golden Points as reference $\chi_{H_2O,\text{ref}}^{\text{Golden Pts}}$, together with the GPS-based air pressure and temperature measurements of the RS41, and have fitted $K_{\text{cal}} = 0.0256$ ppmv/(counts/s) and $S_{\text{offset}} = 78.5$ counts/s. FLASH does not directly monitor the lamp power, but the PMT voltage, lamp current and internal temperature typically remains very stable during the soundings at air pressures below 500 hPa (as in this sounding), indicating that the intensity of the hydrogen-glow discharge lamp is stable and justifies the use of a constant value for $K_{\text{cal}}$. However, the large positive offset of 78.5 counts/s required to minimize the RMSE of the fit warrants a brief discussion. Hydrogen-glow discharge lamps with a mixture of hydrogen and helium are known to suffer from interfering light (often termed 'stray light' in the literature) of the helium line at 318 nm, which overlaps with the spectrum of measured hydroxyl fluorescence after exposition of the water molecules to Lyman-α radiation. The FLASH-B lamp uses a special MgF$_2$ window filter aimed at suppressing the 270–320 nm band emission while still allowing 50 % transmission at the 121.6 nm line. Despite



the presence of this filter, in a closed chamber with reflecting surfaces, the reflection of light from the helium line introduces an additive offset to the fluorescence signal (Vömel et al., 2007b; Fahey et al., 2014). We speculate that (i) also reflections from surfaces of the FLASH balloon instrument could cause a constant positive offset in the fluorescence counts, resulting in

the observed offset, or (ii) the offset originates from the instrument's electronics.

### 4.4    Nonequilibrium correction of CFH without other sensors

Sections 4.2 and 4.3 show how the nonequilibrium correction of chilled-mirror instruments like FPH or CFH can be achieved using another sensor by recalibrating it using the Golden Points. This could be a sensor with higher vertical resolution than the Golden Points, but possibly of lower accuracy than FPH or CFH. Sensors with sufficient capabilities to measure the low

stratospheric humidities include FLASH or the radiosonde RS41, while certain other sondes may not be suitable, because they have insufficient stratospheric sensitivity, such as the often used iMet-1 radiosonde (e.g. Hurst et al., 2011b).

Besides the use of a second instrument for the nonequilibrium correction, there are also other possibilities. These rely on the fact that, for a given ice film, a reference is required between (at least two) Golden Points in order to assess the sensitivity parameter $A$ of the ice film and then perform the nonequilibrium correction. These possibilities are:

-    Use well-defined conditions such as a pre-launch procedure for which a constant dew point can be assumed or conditions in clouds for which 100 %RH can be assumed and used to constrain the nonequilibrium parameters. See example in Section 5.1.

-    Apply self-correction of the chilled mirror instrument when descent data are of good quality in addition to ascent data and can be used to identify features in humidity in the uppermost kilometers of a flight. See example in Section 5.3.

-    Use cases with extreme conditions, such as after the Hunga Tonga eruption, where precise $H_2O$ mixing ratios between the enhanced plume regions are not of great importance compared to advantages of applying the nonequilibrium correction. See example in Section 5.5.2.

However, these possibilities apply only to particular circumstances. A further possibility of general applicability is to change the properties of the feedback controller of the chilled mirror instrument in order to obtain much more densely spaced Golden

Points than with the controller implementations currently used by CFH. This relaxes the requirement for an external reference and enables a nonequilibrium self-correction. This is the case for the recent development of SKYDEW (Sugidachi et al., 2025), see Section 5.6. The SKYDEW (FW v1.0) data product, however, which is based only on the Golden Points, has a vertical resolution of 150 m - 250 m, whereas a much higher vertical resolution (< 50 m) and similar accuracy (< 0.5 K) could be achieved with a proper nonequilibrium correction.



## 5    Nonequilibrium correction under various conditions

We illustrate the application of the nonequilibrium correction using a selection of case studies representing different scenarios during a balloon sounding: prelaunch or "laboratory-like" conditions in a static atmosphere, measurements in the UT/LS in a dynamic atmosphere, and the comparison of ascent and descent data to validate measurements in the stratosphere. These case studies demonstrate that

- crossing the setpoint of the controller, i.e. the preset reflectance value, does not imply that the condensate is in equilibrium with the gas phase, but only that the condensate has the predefined thickness at this instant;
- chilled mirror instruments do not exhibit a first-order time-lagged response, so that the concept of an e-folding time constant is not appropriate;
- smoothing the signal with a low-pass filter and using the residuals as an estimate for the controller error does not always provide an accurate representation of the measurement uncertainty.

Not all instruments or flight segments suffer from nonequilibrium errors to the same extent. While the nonequilibrium errors of some soundings are less than 0.1 K, so that the total uncertainty of the frost point measurement at 1 s resolution (including the uncertainty of the mirror calibration) is better than 0.2 K, the nonequilibrium errors can reach several degrees Kelvin for hundreds of meters for other soundings, so that the conventional technique of segment-wise averaging does not allow to retrieve the frost point if no correction of the nonequilibrium error is applied (see Section 5.4 on low-pass filtering).

### 5.1  Nonequilibrium correction under prelaunch conditions

Figure 3 shows CFH prelaunch data in outside air shortly before the balloon was released. During prelaunch, a small fan is temporarily installed on the upper inlet tube (Vömel et al., 2016), but the ventilation it creates is less than the airflow during the balloon ascent, resulting in more pronounced oscillations in the CFH measurement. At the Golden Points, CFH compares well with the humidity measured by RS41 (after applying the above-mentioned 3.5-s synchronization shift $\Delta_{tele}$). In this example with a nearly constant ambient dew point, it is clear that the that the strong oscillations observed in the signal are caused by mirror temperature oscillations resulting from a controller instability, i.e., "nonequilibrium excursions", and not by a time-lagged response to changes in the dew point. In this special case without external dew point changes, the nonequilibrium errors in mirror temperature are nearly symmetric, but with a slight negative bias due to the nonlinearity of the Clausius-Clapeyron



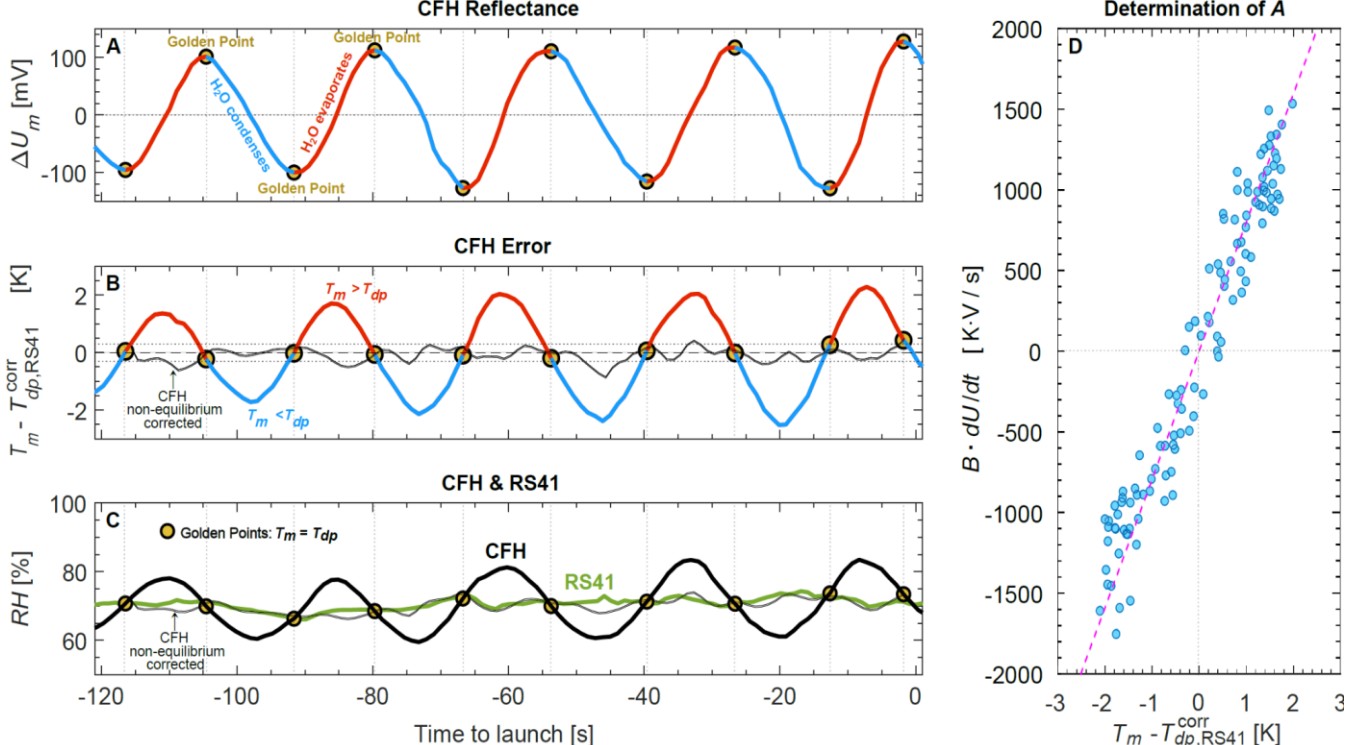

**Figure 3:** Nonequilibrium correction applied to prelaunch measurements of CFH and RS41 before the sounding in Lindenberg on 13 March 2017. **(A)** CFH mirror reflectance deviation from setpoint $\Delta U_m = U_m - 2.5$ V. Color coding: red where reflectance increases as the condensate evaporates, blue where reflectance decreases as the condensate grows. Circles: Golden Points,
where the reflectance reaches an extreme value as the condensate reaches equilibrium with ambient air. **(B)** Nonequilibrium error of CFH, i.e., difference between mirror temperature $T_m$ and the dew point from RS41 (in K). Horizontal dotted lines at $\pm$ 0.3 K: $2\sigma$ accuracy that we assert for the Golden Points of the CFH instrument when the coverage on the mirror is liquid dew instead of solid frost (i.e., in the lower troposphere). **(C)** Green line: RH (in %) from Vaisala's RS41 output. Thick black line: RH from CFH calculated using CFH mirror temperature $T_m$ and air temperature (4.3 °C, measured by RS41). Circles: Golden
Points indicating the times when CFH mirror temperature is at the dew point. Thin black lines in (B) and (C): RH from CFH corrected for nonequilibrium errors according to Eq. 8b (without correction of RS41). **(D)** Scatter plot of the scaled time derivative of CFH mirror reflectance $B \cdot dU_m/dt$ (see Eq. 8b) versus the difference of the CFH mirror temperature and the dew point derived from the RS41 measurement (blue circles). Magenta line: fitted effective sensitivity $A = 773$ V/s.

equation. If, instead of nonequilibrium correction, the $H_2O$ mixing ratio or relative humidity is averaged over two periods (e.g., Vömel et al., 2007b), this would provide an accurate representation of the dew point, but the temporal resolution would still be about three times coarser than the Golden Points sampled with Eq. 1. A proper nonequilibrium correction using Eq. 8a further increases the temporal resolution (here by a factor of 3 compared to using solely the Golden Points), enabling dew/frost point retrievals with both low measurement uncertainty and high temporal resolution. The relative humidity of the





nonequilibrium-corrected CFH agrees well with RS41, even though both instruments might actually measure slightly different air due to small disturbances caused by the handling of the payload during the launching procedure.

The controller of the CFH is programmed so that the deviation of the reflectance from the setpoint, $\Delta U_m = U_m - 2.5$ V, is as low as possible, and thus a certain amount of condensate is targeted. However, when the reflectance passes through the setpoint, this does not indicate that thermodynamic equilibrium is attained. For example, in Fig. 3 the reflectance is at the setpoint at $t$
$= -20$ s, yet the mirror temperature is at maximum distance from the dew point. Thermodynamic equilibrium is only reached when the reflectance remains constant or when it reaches a minimum or a maximum (e.g., at $t = -40$ s).

Figure 3D shows how the nonequilibrium correction term can be determined by means of a scatter plot of the change in reflectance (scaled by $B$) over the mirror temperature deviation from the RS41 dew point. The slope of the magenta line is the resulting effective sensitivity $A = 773$ V/s with an uncertainty estimated at 10 %.

**5.2 Nonequilibrium correction under in-flight conditions**

Figure 4 shows the results of the March 2017 intercomparison sounding above Lindenberg of CFH, FLASH and RS41, which we already discussed in Fig. 2, but now including the nonequilibrium-corrected CFH data. We applied the scheme outlined in Section 4.2 to the CFH original data, i.e., the nonequilibrium correction of CFH using the time-lag and bias corrected RS41 measurements of the same flight. The differences between the CFH original data (data points on black curve) and the nonequi-
librium-corrected CFH data (blue curve) amount to 6 %. Although this correction is small compared to the recalibrations of RS41 and FLASH-B (dashed and solid green curves in Fig. 4), a 6 % change still exceeds the CIMO and GCOS goals.

While the flight section between 11.1 km to 11.4 km could at first be mistaken with a time-lag error of CFH, the undershoots between 11.55 km and 11.8 km and between 10.8 and 11.1 km show that the original CFH data suffer from controller/mirror oscillations, and not from a slow response of its mirror temperature signal. Conversely, the RS41 and FLASH-B hygrometers
provide information about the true humidity structure, but suffer from a dry bias in this flight section and had to be corrected first. Figure 4 also shows the profiles of the time-lag and bias-corrected RS41 (green curve) and the offset-corrected FLASH-B (orange curve). The discrepancies between these corrected profiles and the original CFH amount to ~ 15 %, which is reduced to ~ 5 % by the nonequilibrium correction of CFH. This demonstrates the necessity and validity of the nonequilibrium correction of CFH for quantifying the frost point.





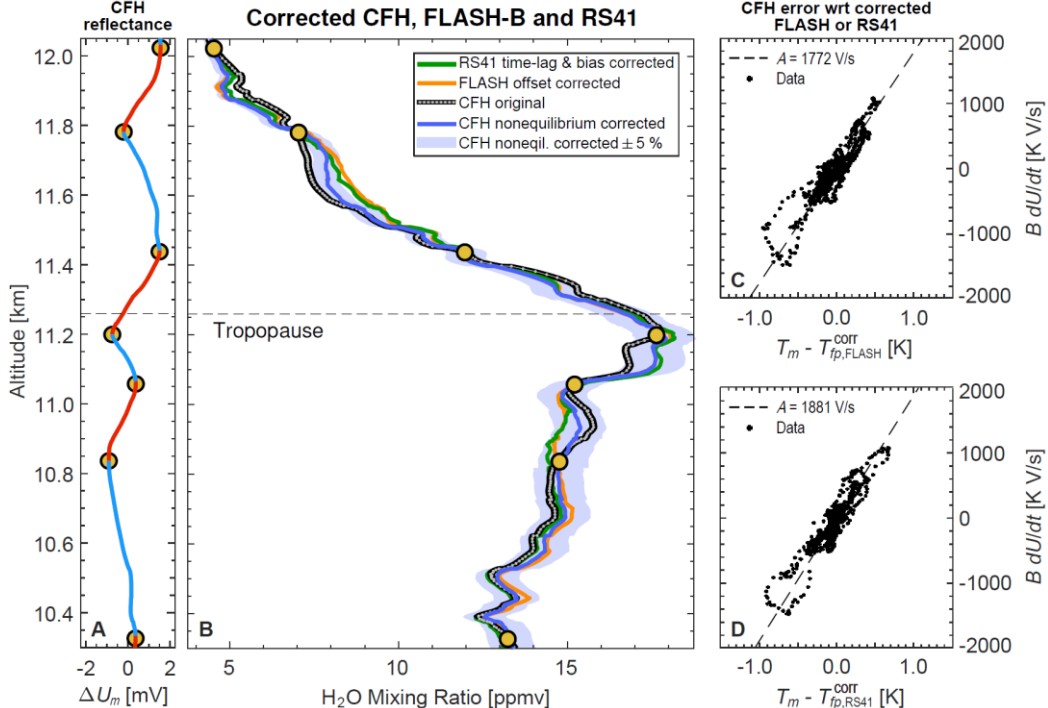


**Figure 4.** CFH-FLASH tandem sounding launched with a 2000 g rubber balloon on 13 March 2017 at 23 UT from Lindenberg Observatory (52.21°N, 14.12°E) together with a Vaisala RS41 radiosonde. **(A)** CFH mirror reflectance voltage $U_m - 2.5$ V with Golden Points at the extremes (maxima and minima of $U_m$) separating sections of the balloon trajectory where the CFH mirror is too warm (red) or too cold (blue). **(B)** $H_2O$ mixing ratio measured by CFH, FLASH and RS41. Orange curve: offset-

corrected FLASH. Green curve: time-lag and bias corrected RS41. Black curve with gray data points: CFH original data. Blue curve: nonequilibrium corrected CFH. Light blue shading: ± 5 % around nonequilibrium corrected CFH. **(C)** Scatter plot of the scaled time derivative of CFH mirror reflectance $B \cdot dU_m/dt$ (see Eq. 8b) over the difference of the CFH mirror temperature and the offset-corrected FLASH frost point measurements yielding an effective sensitivity $A = 1772$ V/s. **(D)** Same as (C) but using the bias- and time-lag corrected RS41 frost point measurements yielding a very similar effective sensitivity $A = 1881$

V/s.

The two examples in this and the previous section show that the use of mirror reflectance is a powerful diagnostic tool. Not only can the atmospheric frost point be determined at the Golden Points, but the time derivative of the reflectance also shows whether the mirror is too warm or too cold, allowing for a corresponding correction. We therefore strongly recommend that chilled mirror instruments always provide the measured reflectance signal in addition to the mirror temperature.

**5.3 Nonequilibrium correction versus treating CFH as time-lagged instrument**

Chilled mirror hygrometers are actively controlled instruments and typically follow a second order response with multiple overshoots (e.g. Jacobs, 1993), i.e., the mirror temperature satisfies a second-order differential equation of the form $\tau^2 d^2 T_m/dt^2$



$+ 2\,\zeta\,\tau\,dT_m/dt + T_m = f_{\text{ext}}(t)$. Here, $\tau$ is the system's natural period of oscillation and $\zeta$ is its damping factor, which are defined, among other things, by the Joule heating of the mirror along with its thermal inertia, the PID settings, the $H_2O$ gas phase

kinetics, as part of the control system. Further, $f_{\text{ext}}$ is the external forcing that results from the changing ambient humidity. Typically, the PID parameters are tuned such that the resulting system follows a damped oscillator response. This $2^{\text{nd}}$ order differential equation with oscillatory behavior and the nonequilibrium equation (Eq. 5) are coupled and must be satisfied simultaneously. As $dU_m/dt$ satisfies the physical constraint given by Eq. 5, this could be erroneously interpreted as $1^{\text{st}}$ order behavior unless one recognizes that the mirror temperature is actually a function of the reflectance signal, $T_m = T_m(U_m)$, via the

heating power applied by the controller (in response to ambient frost point changes).

Depending on the size of the damping factor $\zeta$, the controller behavior can be overdamped ($\zeta > 1$, sluggish approach to the instantaneous equilibrium without oscillations), critically damped ($\zeta = 1$, faster approach to equilibrium without oscillations), underdamped ($1 > \zeta > 0$, oscillatory approach to equilibrium with over- and undershoots), or unstable ($0 > \zeta$, oscillations increase), assuming the forcing term to be constant. Figures 3 and 5 show that CFH often operates in the underdamped regime.

However, due to changes in the forcing term, there are situations lasting for a few seconds, where the mirror temperature oscillations are strongly underdamped with oscillations approaching instability. Conversely, there are situations where the mirror temperature oscillations are overdamped and very close to a first order response. Under these latter conditions, a simple correction can be used to describe CFH, mimicking a time-lag similar to the behavior of RS41, and an exponential approximation to equilibrium is assumed (as in Miloshevich et al. (2004)). This approach has been applied to chilled mirror hygrom-

eters in selected studies in the past (e.g. Vömel et al., 2007b; 2016; Hasebe et al., 2013) and allowed to capture the stratospheric humidity characteristics measured during both balloon ascent and parachute descent.

We revisit a CFH flight from Lindenberg in May 2014 that was discussed by Vömel et al. (2016). There, humidity features measured in the stratosphere during both ascent and descent have been matched in altitude by correcting the mirror temperature assuming a first-order time-lag response with a time constant of 10 seconds. Assuming a simple time-lag between the mirror

temperature and the true frost point, for whatever reason, means that there is a time-lag relationship $T_{fp}(t) = T_m(t) + \tau_{\text{CFH}}\,dT_m/dt(t)$, where $\tau_{\text{CFH}}$ is the time constant for the apparent time-lag (here 10 s), or equivalently $\chi_{H_2O}(t) = \chi_{H_2O,m}(t) + \tau_{\text{CFH}}\,d\chi_{H_2O,m}/dt(t)$ for the water vapor mixing ratio. This formula is mathematically equivalent to the phase correction applied to capacitance humidity sensors by Miloshevich et al. (2004). It corresponds to a first-order response, as it has also been assumed by Hasebe et al. (2013). When making this assumption, however, it should not be forgotten that

the CFH is not a capacitive humidity sensor with a diffusively delayed and damped response of the polymer film. Rather, CFH is an actively controlled sensor, with the mirror temperature and mirror reflectance interacting in a feedback loop. The physical system of CFH has in fact physical time constants, i.e., it is characterized by the Joule heating and thermal inertia of the mirror (with characteristics times of a few seconds to achieve a target temperature), the thermistor measurement of the mirror condensate temperature (characteristic time in the order of the ms, i.e. negligible), and by the kinetics of the diffusion of $H_2O$



molecules through the gas phase of the Prandtl boundary layer above the mirror and the growth/evaporation of a measurable
amount of ice on the mirror (with widely varying characteristic times between 1 ms and a few seconds, depending on the
amount of $H_2O$ in the gas phase and ice film morphology). However, the output system of CFH, i.e. its temporal response, is
characterized by controller oscillations in the mirror temperature aiming at maintaining a constant thickness of the mirror
condensate.

Figure 5 compares the raw data from this 2014 flight in Lindenberg (panel A) with the time-lag correction $\tau_{CFH}$ = 10 s (panel
B), and the nonequilibrium correction, Eq. 8a (panel C), which is based on the Golden Points and nonequilibrium correction
(panel D). We determine the CFH nonequilibrium correction by using the CFH descent data to establish the a-priori for the
ascent data and vice versa. Figure 5 reveals that at altitudes where both the ascent and descent exhibit simultaneous Golden
Points, also both frost points agree well, and without any lag. The nonequilibrium correction provides a better match of the

ascent and descent features in stratospheric humidity than the time-lag correction. This is because the nonequilibrium correc-
tion makes use of the mirror reflectance measurement through Eq. 8a. We have excluded the first 400 m of descent, like in
Vömel et al. (2016), as they may be affected by an unstable airflow and short-term contamination from the payload. We have
derived the effective sensitivity for the ascent, $A_{ascent}$ = 800 V/s and for the descent, $A_{descent}$ = 2300 V/s, by fitting for the best
match between ascent and descent humidity features. Because of the higher ventilation during descent (~ 20 m/s descent rate

measured in this section of the profile, compared to 5 m/s ascent rate), the skin layer over the mirror is shallower, thus the
exchange of water molecules with the flowing air is faster, resulting in a larger effective sensitivity during descent compared
to during ascent. We can achieve a similar result by assuming the skin layer thickness to evolve as $(\rho v)^{-1/2}$ and assuming the
same underlying morphological sensitivity $A'$ of the condensate for both ascent and descent.

The nonequilibrium correction achieves a better match between ascent and descent profiles than the time-lag correction. This

is because taking the reflectance measurement into account (as time dependent 'pseudo time constant') allows to correct for
the main reason for the ascent/descent features mismatch, which is CFH's controller induced nonequilibrium oscillations of
the mirror temperature, and not a slow response of the mirror temperature measurement. The nonequilibrium correction is
valid everywhere, whereas applying a time-lag correction is valid only within flight sections where the nonequilibrium excur-
sions can be approximated by a first order response, such as in the flight shown in Fig. 5. To illustrate that the nonequilibrium

error is qualitatively different than a time-lag error, but may appear locally as a time-lag error, we rewrite the nonequilibrium
correction (Eq. 8a) in the mathematical form of a first order response: $T_{fp}(t) = T_m(t) + \tau_{noneq} \, dT_m/dt$ with the nonequilibrium
'pseudo time-constant' $\tau_{noneq} = B(t)/A \cdot (-dU_m/dT_m)$. The term $-dU_m/dT_m$ (in V/K) contains the reflectance measurement $U_m$.
Therefore, $\tau_{noneq}$ (in s) can take both positive and negative values and is highly variable in time, which is not a physically
meaningful quantity. Hence, applying a time-lag correction is generally not appropriate, although some sub-sections of the

sounding profile (where $\tau_{noneq}$ is positive and approximately constant) may resemble a time-lag error. As shown above, the true
nature of mirror temperature oscillations is not a first order response (i.e. time-lag), but rather a second order response.





The median value of the pseudo time-constant $\tau_{noneq}$ of 11 seconds agrees well with the 10 seconds estimated by Vömel et al. (2016), which explains why in the examined altitude section their time-lag correction produced results similar to the nonequilibrium correction. In contrast to the assumed time-lag of fixed 10 seconds, $\tau_{noneq}$ varies strongly with time, with a robust
standard deviation of 37 seconds (calculated as half the difference between the 84.13% and the 15.87% percentiles) around the 11 seconds median (see Fig. 5E).

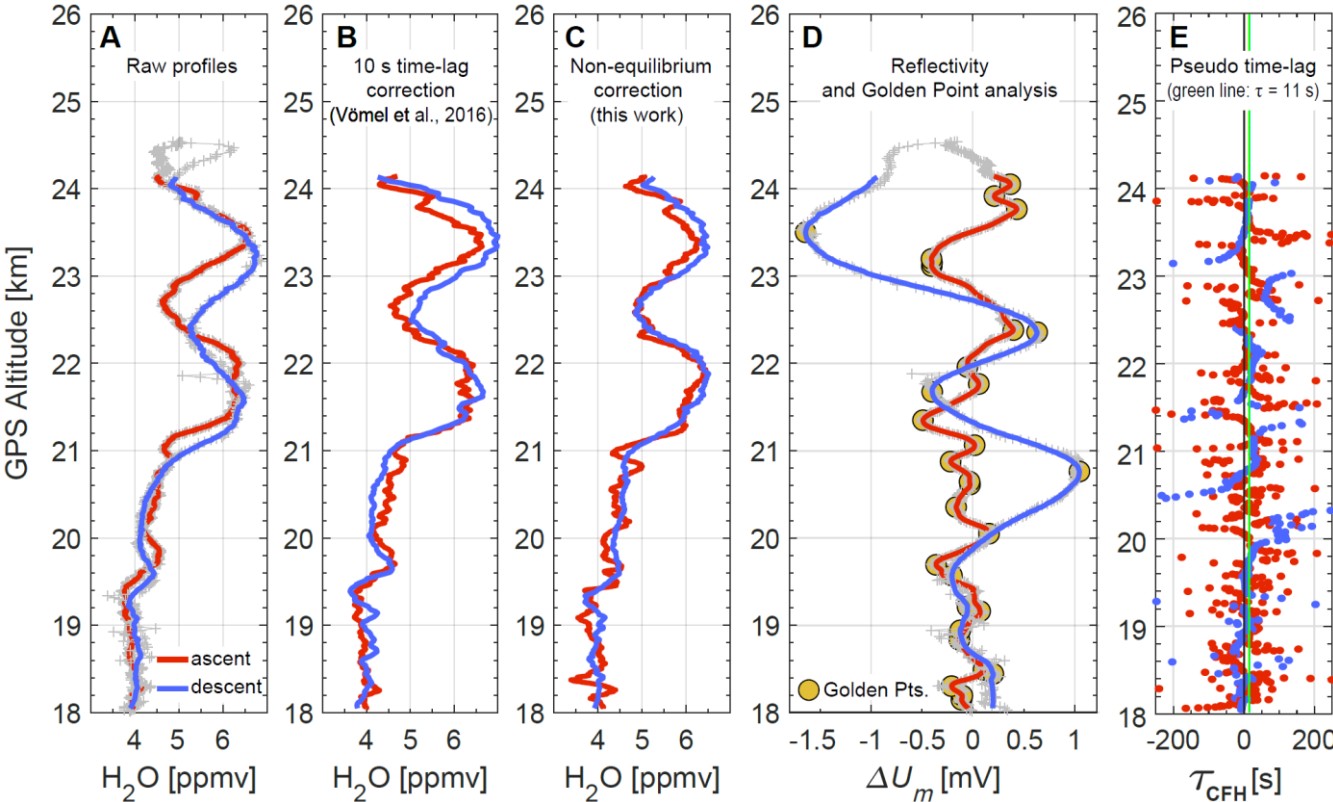

**Figure 5.** Water vapor mixing ratios during ascent and descent measured by CFH on 20 May 2014 22UT over Lindenberg (serial 2L3407 with FW 6.40 on LI133). **(A)** Raw signal of $H_2O$ mixing ratio ascent with smoothed mirror temperature (using
a second-order, 15 s half-width, Savitzky-Golay filter) versus GPS altitude (air temperature and air pressure from iMet1-RSB radiosonde, serial 14246; telemetry software (Strato 9.54) applied an offset of 2.9 hPa to the pressure measurement). **(B)** Time-lag corrected mixing ratio profiles, using an e-folding time constant of 10 s, as in Vömel et al. (2016). **(C)** Nonequilibrium corrected mixing ratio profiles, using an effective sensitivity $A = 800$ V/s for ascent and $A = 2300$ V/s for descent. **(D)** Raw signal of mirror reflectance profiles (smoothed with a second-order, 21 s half-width, Savitzky-Golay filter) and associated
Golden Points. **(E)** Ascent and descent profiles of the pseudo time lag constant $\tau_{noneq}$, i.e. the mathematical time constant that would be required to reproduce the atmospheric frost point profile after applying the nonequilibrium correction. The median of all $\tau_{noneq}$ happens to be 11 s (green line), i.e. close to the time lag applied by Vömel et al. (2016).



For these reasons, treating or reporting the response of chilled mirror instruments in the same manner as the response of polymer thin film sensors (as in e.g. WMO, 2023, their Table 12.7) is misleading and generally not appropriate, because high-
quality chilled mirror hygrometers do not suffer from time-lag of their mirror temperature, but from nonequilibrium errors.

## 5.4 Nonequilibrium correction versus low-pass filtering

Atmospheric water vapor profile measurements with an uncertainty of less than 4 % (2σ) in the $H_2O$ mixing ratio in the stratosphere have been defined as a goal by CIMO (WMO, 2023). This is currently within reach for only a few instruments, including the CFH (Bodeker et al., 2014). In line with this, the AquaVIT-1 intercomparison of atmospheric water vapor meas-
urement techniques considered CFH as one of the 'core' instruments to construct a metrological reference (Fahey et al., 2014), and the SPARC water vapor assessment II (WAVAS), which investigated biases and drifts of water vapor satellite retrievals with respect to frost point hygrometers, accepted CFH as one of the reference instruments (Kiefer et al., 2023), as did the WMO 2022 Upper-Air Instrument Intercomparison Campaign (Dirksen et al., 2024).

Due to the generally high quality, it is often not necessary to apply corrective measures for errors of the CFH. Yet, averaging
over extended sections (> 125 m in the stratosphere) of the balloon ascent profile by applying a Gaussian filter is part of the standard CFH data processing procedure. This is meant to quantify and correct errors, as described by Eqs. 1-5 in Vömel et al. (2016). Figure 6 compares the averaging by Gaussian filtering with the nonequilibrium correction using the example of a CFH-RS41 tandem flight on February 21, 2020 in Ny-Ålesund on Spitsbergen (Svalbard, Norway). The figure shows a nearly isothermal profile section (208 K - 210 K air temperature) in the lowest stratosphere in polar winter. Due to the sufficiently
high relative humidity above ice, $S_{ice} > 0.1$, the RS41 radiosonde remains sensitive to moisture (e.g. Survo et al., 2015) and the humidity measurement of the RS41 can be used as a reference for estimating the effective sensitivity parameter $A$.

The effect of averaging and smoothing with the Gaussian filter can be seen by comparing the original CFH data (gray crosses) and the pink "CFH NDACC" curve (with a width of fitting kernel as in the NDACC Revision 0 data (NDACC, 2020), i.e. linearly interpolated between 4 s at $T_m = 20°C$, 12 s at $T_m = -75°C$ and 100 s at $T_m = -95°C$, resulting in a vertical resolution
(obtained by multiplying the width of the kernel with $(2\pi)^{\frac{1}{2}} \times 5$ m/s) ranging from 137 m to 261 m for the flight section shown in Fig. 6). Gaussian filtering reduces the noise in the mirror temperature, but does not correct the controller's nonequilibrium errors. The changes due to Gaussian filtering amount to a maximum of ±0.2 K in the frost point temperature and about ±1 % in $S_{ice}$. In contrast, the nonequilibrium errors actually caused by controller oscillations are about a factor of 5 larger, as the comparison of the original CFH data (gray crosses) and the black curve "$T_{fp}$" shows. The nonequilibrium corrected CFH profile
is closer to the moisture characteristics measured by the RS41 than to $T_{m,raw}$. This example also shows that the actual atmospheric humidity profile between the Golden Points can be highly variable (e.g. the two moist features between 9.5 and 9.8 km and between 10.6 and 10.9 km, highlighted by red boxes in Fig. 6A) and that the correction of the mirror temperature profile by simple interpolation between the Golden Points is not necessarily accurate.



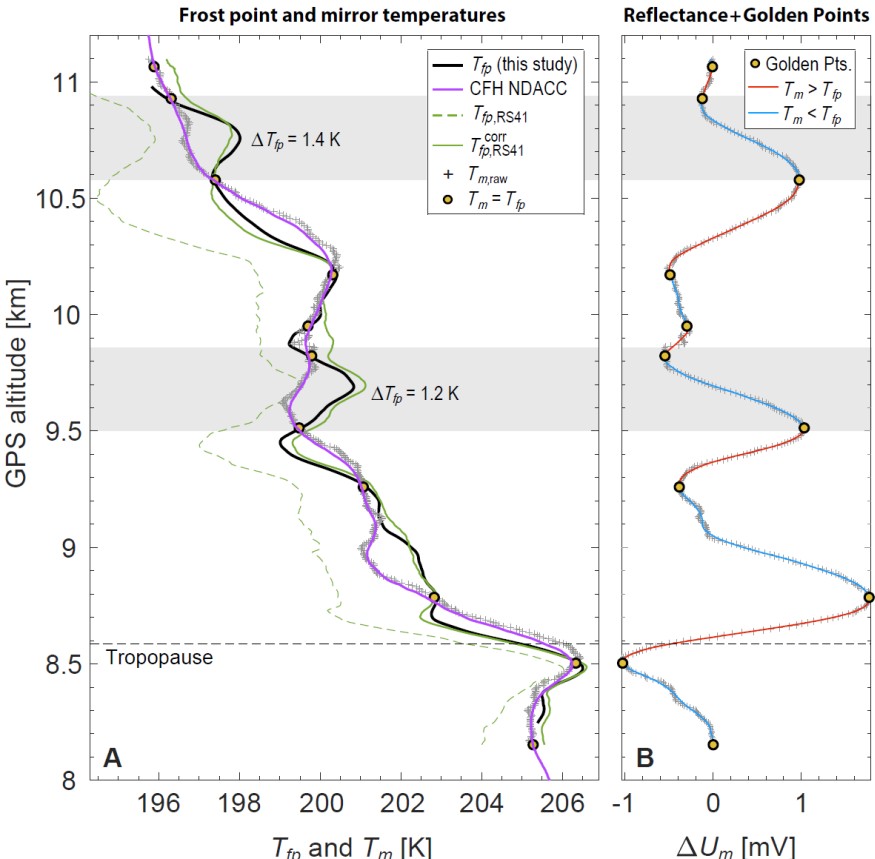

**Figure 6.** Frost point profiles in the lowermost stratosphere from the sounding in Ny-Ålesund on 2020-02-21 19UT. **(A)** Raw CFH mirror temperature (gray +), CFH Golden Points (yellow circles), RS41 frost point (Vaisala product, dashed green), RS41 time-lag and bias-corrected frost point for best fit at the CFH Golden Points (solid green), nonequilibrium corrected CFH mirror temperature using the effective sensitivity $A$ calculated from the corrected RS41 (solid black). **(B)** Mirror reflectance. Raw CFH mirror reflectance (gray +), smoothed CFH mirror reflectance colored red when mirror too warm and blue when mirror too cold, CFH Golden Points (yellow circles). The nonequilibrium corrected CFH profile (black solid line) fits to the stratospheric humidity features captured by the RS41 (green solid lines) much better than the NDACC Revision 0 (NDACC, 2020) profile (purple line), in particular in the gray shaded regions.

In summary, in Fig. 6 neither a Gaussian smoothing of the CFH mirror temperature nor a first-order time-lag correction (Vömel et al., 2016) capture detailed structures of atmospheric humidity and correct for nonequilibrium controller errors. If 250-m boxcar averages (as defined by NOAA and termed 'levels' in their nomenclature) were applied to the mirror temperature centered at 9.75 km and 10.75 km, this would result in a dry bias of about 1 K in frost point and the associated uncertainty, i.e. the standard error of the 250 m linear fit (Hall et al., 2016), would be underestimated. However, using the measurements of a second, independent sensor with greater temporal resolution than the CFH Golden Points, is the appropriate method to





determine $T_{fp}$ between the Golden Points. The second instrument for a nonequilibrium correction could be an RS41 that is pre-
corrected for time-lag and bias by means of the CFH Golden Points, as described in Section 4.2. Conversely, a mere linear
interpolation between Golden Points would fail to capture the structure of the atmospheric humidity field (nor would variations
thereof, e.g. the average of the maxima-only and minima-only linear interpolations, or low-pass filtering of the linear interpo-
lation).

Good control of the frost on the mirror is a tradeoff between producing many Golden Points ($dU/dt = 0$) and minimizing the
nonequilibrium error (small $|dU/dt|$). A strongly oscillating PID controller producing Golden Points at <100 m vertical resolu-
tion, e.g. SKYDEW with its current PID implementation (Sugidachi et al., 2025), would be able to capture detailed humidity
structures as in Fig. 6. However, PID controllers with strong oscillations tend to have a larger oscillation amplitude and thus a
larger uncertainty (~ 0.5 K) at the Golden Points (defined as the maximum mirror temperature difference ±1 s around Golden
Points), as well as larger nonequilibrium errors. On the other hand, CFH has a well-tuned controller for background conditions
of tropospheric and stratospheric humidity, with mostly small nonequilibrium errors. Nevertheless, rapid frost point changes
can cause large nonequilibrium errors, resulting in Golden Points more than 100 m apart. Thus, it is unclear whether PID
control is the most appropriate approach to frost control of chilled mirror hygrometers.

## 5.5 Flight-to-flight variability

An operationally well-adjusted CFH is one for which the sum of all sources of uncertainty in frost point (or dew point) meas-
urements is better than 0.2 K, and systematic errors, which most likely affect long-term climate series, are demonstrably smaller
than 0.1 K (Vömel et al., 2016). In fact, the nonequilibrium analysis of this work confirms this performance for many of the
analyzed CFH soundings. However, there are notable exceptions. For 11 % of the about $10^5$ measurement points analyzed in
Section 6, the nonequilibrium analysis revealed errors in the frost point estimated from the raw mirror temperature (i.e., $|T_m -
T_{fp,RS41}^{corr}|$) of more than 0.2 K, for 5.5 % of more than 0.3 K, for 2.2 % of more than 0.5 K, and for 0.6 % larger of more than 1
K. For the subset of points near the tropopause (about 23000 points), 18 % have errors in frost point larger than 0.2 K, 10 %
larger than 0.3 K, 4% larger than 0.5 K, and 1 % larger than 1.0 K. Since 40 of the analyzed 70 soundings have been launched
from Lindenberg with under the optimal launch conditions at the GRUAN Lead Center and the moderate midlatitudinal flight
conditions, these statistics might be slightly worse for other stations and flight campaigns with lower $H_2O$ mixing ratio, such
as in the tropical UT/LS. As mentioned earlier, not all instruments or flight legs are equally affected by nonequilibrium errors.
As shown by Eq. 5 ($1/A' \cdot dU/dt \sim \chi_{H2O}(T_m) - \chi_{H2O}(T_{fp})$), the reasons for this are the quality of the ice film, which is determined
by the properties and cleanliness of the mirror, by the condensate thickness at the PID setpoint and by the cleaning cycles (all
described by $A'$), as well as the properties of the vertical $H_2O$ profile ($\chi_{H2O}$), which determines the history in the integral part
of the PID with a fixed setpoint (at low $H_2O$ mixing ratios, a minute change in reflectance corresponds to larger deviations
from frost point than at high $H_2O$ mixing ratios). Subsequently, we discuss a few selected cases with widely varying perfor-
mance and in Section 6 we discuss the statistics of the 70 soundings.





Figure 7 shows a collection of nonequilibrium errors obtained from six flights. The magnitude of the error in frost point temperature increases from left to right and varies between less than 0.1 K to several K, and from being symmetric to highly asymmetric in terms of frost point.

**Figure 7:** Four-kilometer slabs of various soundings with CFH, sorted according to nonequilibrium error. Slabs start after the second cycle (at $T_{fp}$ = -53°C) and end at the hygropause. We use the higher reliability of RS41 below the hygropause to correct CFH in panels **A-E**. An exception is panel **F**, which shows the CFH measurement in the core of the Hunga Tonga-Hunga Ha'apai plume, for which CFH was corrected assuming a background mixing ratio of 5 ppmv. Panels A and F will be discussed in detail in the following subsections.



### 5.5.1 Case Study: Stratospheric streamer with enhanced H₂O over Lindenberg

Figure 7A shows the CFH measurements at Lindenberg (52.21°N, 14.12°E) on 19 April 2018 below the hygropause. About 10 km higher, in the stratosphere, the same sounding showed a significant increase in the $H_2O$ mixing ratio of 1.5 ppmv above the background value of 4.8 ppmv at an altitude of 22.0 km - 22.5 km. This anomaly is unusual and the question arises as to whether this is a false measurement. In order to directly compare the measurements during the ascent and descent, we show the $H_2O$ profile as a function of potential temperature in Fig. 8. In fact, the anomaly can also be found in the descent data between 535 K and 550 K. The CFH nonequilibrium correction corroborates the very good altitude agreement during ascent and descent. This is further confirmed by a measurement with an aerosol backscatter sonde.

For the nonequilibrium correction of the ascent, we use the morphological sensitivity $A'$ determined for the upper troposphere in Fig. 7A. The validity of this procedure is not self-evident, as the morphology of the ice film may have changed during the 10 km interval of ascent. However, the RS41 provides no information above 20 km, so assuming the same morphology is the most useful assumption. We also use a constant morphological sensitivity $A'$ for the descent, but the much higher ventilation due to the fast descent has to be taken into account. We take account of the about five times faster descent rate resulting in enhanced ventilation of the mirror by virtue of $\Delta x_{\mathrm{diff}} \propto Re^{-1/2} \propto v_{\mathrm{flow}}^{-1/2}$, which reduces the Prandtl layer thickness $\Delta x_{\mathrm{diff}}$ and enhances the effective sensitivity $A$ by $5^{1/2}$ relative to the ascent of the balloon. The excellent agreement between the humidity measured during ascent and descent strengthens the validity of these assumptions.

The aerosol backscatter probe COBALD (Brabec et al., 2012) and an electrochemical ozone probe (ECC) flew on the same payload. The COBALD data show a minimum in the backscattering (both in the 940 nm and 455 nm channels) in the same layer as the $H_2O$ anomaly (Fig. 8D). CFH and COBALD measurements consistently show that the air mass at 540 K has a different origin than the rest of the profile. High humidity and low number density of aerosol particles point to the polar vortex as a possible region of origin. The air from there is "old", meaning that methane oxidation is advanced, which increases the $H_2O$ mixing ratio, and the particle number density is low because large particles have precipitated on the transport route to the poles. This is confirmed by the COBALD color index, which shows a minimum of about 4-5 in this layer. To embed the results in a microphysical background, we carried out Mie calculations using a lognormal distribution of aerosol particles with a width $\sigma = 1.6$ and a mode radius of 40-50 nm that result in a color index 4-5, whereas the typical mid-latitudinal mode radius of 60-80 nm yields a color index of about 6-7, in agreement with the measurements.

The RS41 was not able to detect the stratospheric $H_2O$ anomaly, not even qualitatively, as it already lost sensitivity to humidity at lower altitudes (i.e. the humidity signal cannot be distinguished from the noise in the capacitance measurement). Finally, an electrochemical ozone sonde also on board suggests some evidence of this anomaly with a local decrease in ozone concentration around 540 K, but unfortunately it suffered from technical problems and will not be discussed further.



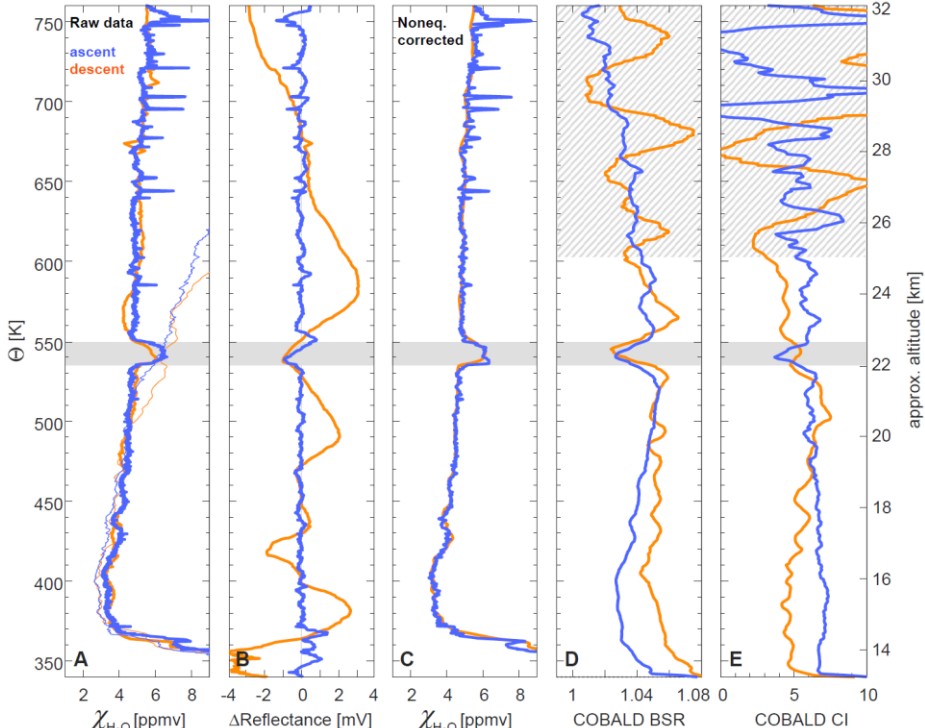

**Figure 8:** Balloon sounding on 19 April 2018 in Lindenberg (52.21°N, 14.12°E) showing remarkable $H_2O$ and particle backscatter anomalies (gray shading) in measurements by CFH and the backscatter sonde COBALD during ascent (blue lines) and descent (orange lines). **(A)** $H_2O$ mixing ratios retrieved from CFH raw data. Thin lines show the RS41 data (Vaisala product). In this sounding, RS41 lost sensitivity above potential temperature level 500 K (≈ 20.5 km altitude). **(B)** CFH reflectance signal. **(C)** Nonequilibrium correction of the CFH mixing ratio, making the blue ascent and the orange descent lines almost coincide. **(D)** COBALD backscatter ratio ($BSR_{455nm}$) at 455 nm, with 10 K (potential temperature) moving average. **(E)** COBALD Color Index CI = ($BSR_{940nm}$-1)/($BSR_{455nm}$-1), also with 10 K moving average. Hatched area: above 600 K the backscatter signal is too noisy during descent, possibly due to excessive swinging of the dangling payload.

So, the CFH / COBALD tandem with nonequilibrium CFH correction reveals that locally elevated $H_2O$ over Lindenberg at about 540 K (22.5 km) together with a significant change in the aerosol properties is not an artifact. Rather, it is a true atmospheric signal most likely caused by an erosion of the polar vortex in spring 2018 creating a thin layer with high water vapor and low aerosol and ozone concentrations.

### 5.5.2 Case Study: Hunga Tonga-Hunga Ha'apai stratospheric plume

Figure 9A shows the CFH measurements from Maido Station on Reunion Island (21°S, 55°E) on the evening of 22 January 2022. The massive explosive eruption of the submarine volcano Hunga Tonga-Hunga Ha'apai (HTHH) (21°S, 175°W) injected 50-150 Tg $H_2O$ (e.g. Vömel et al, 2022; Khaykin et al., 2022; Millán et al., 2022) and about 0.4 Tg $SO_2$ into the stratosphere



(Millán et al., 2022). The Tonga volcano Rapid Response Experiment (TR²Ex) at the Maïdo observatory enabled in-situ bal-loon-borne measurements in the core of the main volcanic plume as it moved over Réunion Island 6 to 8 days after the eruption. The humidification of the stratosphere is thought to have led to a rapid depletion of ozone (5% within one week of stratospheric

$O_3$ above the tropical southwestern Pacific and Indian Ocean region). Dips in the $O_3$ profile correspond with peaks in both aerosol backscattering and $H_2O$ mixing ratios, leading to the conjecture that volcanic $H_2O$ and $SO_2$ injection led to fast $SO_2$ oxidation (through an OH enhancement), subsequently new particle formation (binary nucleation of $H_2SO_4$ and $H_2O$) and growth of preexisting particles (via condensation of $H_2SO_4$ and $H_2O$) and, possibly, particle coagulation (Zhu et al., 2022; Asher et al., 2023). The increase in total reactive surface area enables heterogeneous catalytic reactions causing $O_3$ destruction

at relatively high (~ 220 K) temperatures (Evan et al., 2023; Zhu et al., 2023).

The $H_2O$ profile shown by Evan et al. (2023) is the thin dotted line in Fig. 9B; it has only a coarse vertical resolution (see polygonal line with 250 m linear elements in Fig. 3 of Evan et al.) and exhibits only the two $H_2O$ maxima. The original data show much more structure, but also a minimum that extends to an unrealistic 1 ppmv (bold black line in Fig. 9B). The corre-lation with the aerosol measurements (Fig. 9C) has a linear correlation coefficient $R^2 = 0.61$ (in log $BSR_{940\ nm}$ vs log $\chi_{H2O}$).

Applying the CFH nonequilibrium correction to the data with high vertical resolution increases the level of correlation between the aerosol and $H_2O$ profiles to $R^2 = 0.86$.

The nonequilibrium correction allows us to compute the RH profile and compare it with the backscatter ratio (BSR, Fig. 9C) and color index (CI, Fig. 9D) of the COBALD sonde, both in high resolution. It is interesting to compare the three layers with increased $H_2O$ mixing ratio between 19-22 hPa (layer 1), 24-26 hPa (layer 2) and 28-29 hPa (layer 3). In layer 1, there is a

clear increase in the color index to values of CI = 8.5-10, which indicates effective particle radii (3$V$/$A$) of 0.35-0.5 µm (ac-cording to Mie calculations assuming a lognormal distribution with width $\sigma = 1.6$ of $H_2SO_4$-$H_2O$ droplets at $RH_{ice} = 40$ %). In layer 2, the color index is slightly lower, with values around 8, in line with effective particle radii of about 0.3 µm. In layer 3, there is no clear correlation between $H_2O$ mixing ratio and color index, but CI ~ 7 suggests an effective particle radius of about 0.15 µm (similar to unperturbed regions with CI ~ 6, $\sigma = 1.8$ and $RH_{ice} = 1$ %). While the backscatter ratio is influenced by

mixing with unperturbed air taking place during atmospheric transport time of about one week since the eruption occurred, the color index has the advantage of being virtually unaffected by mixing. Therefore, the difference between the color indices of the three layers suggests that much more of the emitted $SO_2$ was converted into sulfuric acid in layer 1 than in layer 3. The $H_2SO_4$ then condenses onto existing particles or forms new particles that coagulate with existing ones, both of which contribute to particle growth. The likely reason for the faster conversion of $SO_2$ to $H_2SO_4$ is the higher humidity in layer 1, which exceeds

that in layer 3 by about an order of magnitude, leading to a faster formation of hydroxyl radicals and thus to a faster formation of $H_2SO_4$. The combination of the high-resolution CFH and COBALD measurements in Fig. 9 and their interpretation is fully consistent with and supports the work of Asher et al. (2023).



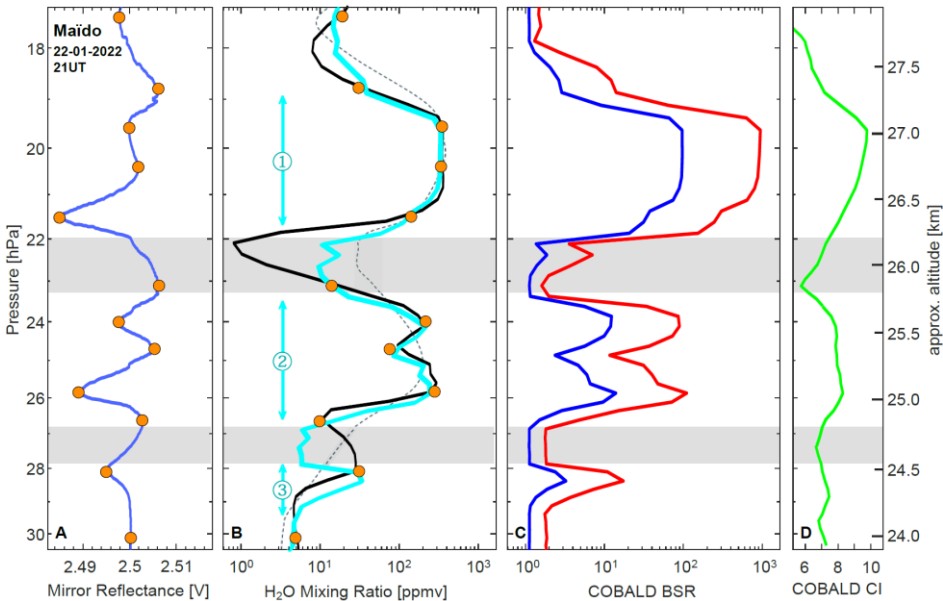

**Figure 9:** CFH-COBALD tandem measurements launched from the Maïdo Observatory in Réunion Island (21°S, 55°E) as-
cending into the main plume of the volcanic eruption of the Hunga Tonga–Hunga Ha'apai (HTHH) submarine volcano on 22
January 2022 at 21UT. **(A)** CFH reflectance signal (blue line) and Golden Points (orange circles). **(B)** CFH raw data of $H_2O$
mixing ratio (black line); nonequilibrium-corrected CFH mixing ratio (cyan); CFH data with 250-m smoothing (thin dotted
line) as used by Evan et al. (2023). **(C)** Aerosol backscatter ratio measured by COBALD at 940 nm ($BSR_{940nm}$, red line) and
455 nm ($BSR_{455nm}$, blue line). **(D)** Aerosol Color Index measured by COBALD, CI = $(BSR_{940nm}-1)/(BSR_{455nm}-1)$. Gray shaded
areas mark the largest errors in the $H_2O$ profile that occur in the $H_2O$ minima (i.e. in regions of the atmosphere unperturbed by
the volcanic plume). $H_2O$ mixing ratios (this work), backscatter ratios and color index in panels B, C, and D are displayed with
0.25 hPa (i.e., 50 - 100 m) binning.

Since the CFH reflectance measurement and COBALD's backscatter measurement arise from completely different measure-
ment principles, the strong correlation is an impressive indicator for the validity of the nonequilibrium correction. The gray
shaded areas indicate the largest errors in the $H_2O$ profile, which occur in the $H_2O$ minima. In their paper, Vömel et al. (2022)
show the profile of the raw $H_2O$ mixing ratio in a figure that ranges between 10 ppmv and 4000 ppmv, so the minimum values
in the profile at 26 km remain invisible. Of course, the resulting errors of up to 20 K in the frost point and more than a factor
of 10 in the $H_2O$ mixing ratio between the $H_2O$-enriched layers 1 and 2 are in flagrant violation of the target set by the WMO.
However, it must also be clear that ignoring the unperturbed regions outside of the $H_2O$-enriched layers, as done by Vömel et
al. (2022) and Evan et al. (2023), has little influence for the general assessment of this extraordinary eruption.

### 5.6 Nonequilibrium self-correction

As mentioned in Section 4.4, a much denser array of Golden Points along the vertical profile of a balloon ascent than is possible
with CFH can eliminate the need for a nonequilibrium correction or enable a self-correction of a chilled mirror instrument if



even higher vertical resolution is desired. Self-correction accomplishes independence of other instruments that have higher

vertical resolution (at lower accuracy), such as FLASH-B or RS41. Recently, there has been a development in this direction in the form of the Peltier-based chilled mirror hygrometer SKYDEW (Sugidachi et al., 2025). While the Golden Points of CFH are typically separated by a few hundred meters (see multiple examples in Figs. 2-9), SKYDEW's Golden Points are typically 15 m apart in the troposphere and 20-25 m in the stratosphere. Figure 10 illustrates this feature of SKYDEW that has the potential to lead to a better constraint of the frost point. However, Fig. 10B also shows that the very large amplitude of the

nonequilibrium excursions leads to strong temporal changes, $dT_m/dt$, at the Golden Points. This renders the retrieval of precise timings of the Golden Points along the balloon trajectory difficult. Even sampling time differences between $T_m$ and $U_m$ of less than 1 s cause SKYDEW to suffer large uncertainties in the Golden Point timing, leading to incorrect Golden Points temperatures. This demonstrates that the perceived advantage of a denser Golden Point array, in terms of vertical resolution, can easily become a disadvantage in terms of frost point accuracy on instruments with 1 Hz data and large nonequilibrium excursions.

In Fig. 10C, we have shifted the mirror temperature data by 0.4 s (about 2 m vertical distance) relative to the reflectance signal, which can remove most of the atmospherically meaningless oscillation of the Golden Points (i.e., $T_m$ is shifted such that the RMS difference between the maxima-only and minima-only polygonal lines is minimized). After this synchronization of $U_m$ and $T_m$, the Golden Points of SKYDEW can be used without further nonequilibrium correction, albeit with a vertical resolution of 15-25 m and a sampling uncertainty of ±1 s in the Golden Points (Sugidachi et al., 2025) related to the steep slope $dT_m/dt$.

For finer structures to be resolved, SKYDEW requires a nonequilibrium-correction, e.g. by means of a time lag- and bias-corrected RS41 in the same way as done for CFH (violet solid line in Fig. 10D).

Alternatively, SKYDEW can be self-corrected: by using the polygonal line defined by the neighboring Golden Points as a-priori reference in the nonequilibrium correction procedure, we can determine the sensitivity $A$ and obtain the dashed red line in Fig. 10D. The good agreement with the nonequilibrium corrected profile based on RS41 demonstrates the high potential of

1195 this procedure, which allows SKYDEW to be nonequilibrium corrected without a second independent sensor. However, due to the steep slopes $dT_m/dt$ at the Golden points, even with synchronized $U_m$ and $T_m$, the noise in the measurements leaves the Golden Points with finite errors, whose quantification is beyond the scope of this paper. This calls for a better feedback control to reduce the amplitude of the nonequilibrium excursions, or for sub-second telemetry. In comparison, for CFH in its current implementation, using another sonde with appropriate capabilities remains the best option for nonequilibrium correction.





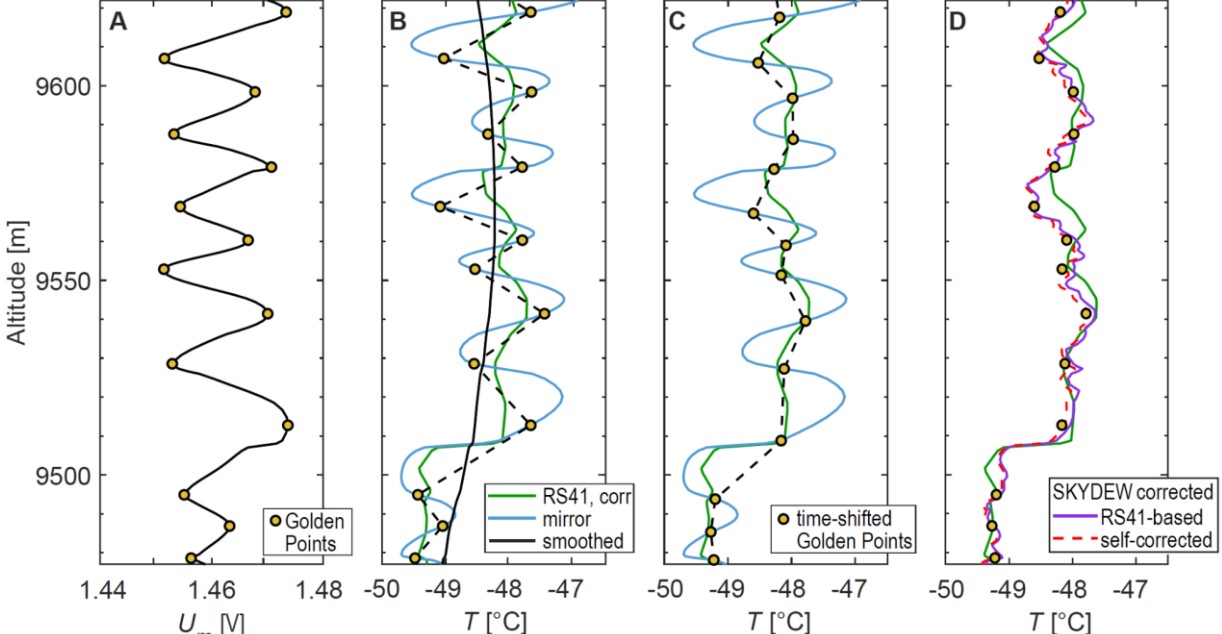

**Figure 10:** Self-correction procedure of SKYDEW (serial 20570, firmware 1.0), launched in tandem with a Vaisala RS41-SGP (V1021230) from the Lindenberg Observatory on 18 July 2023 at 21UT. **(A)** Skydew mirror signal (upscaled from 1 Hz to 5 Hz with linear interpolation to allow sub-second synchronization of $U_m$ and $T_m$, then smoothed with a gaussian filter with an equivalent sample size of about 2 s in order to retain high temporal resolution) and the detected Golden Points (minima/maxima). SKYDEW uses light scattering instead of reflectance as its feedback signal, such that, in contrast to CFH, the mirror signal increases as the mirror condensate is growing and vice-versa. **(B)** Skydew mirror temperature (blue) with Golden Points marked in yellow. The linear interpolation between Golden Points is shown as a black dashed line. The Golden Points 'zig-zagging' effect is not an atmospheric effect and is due to a misalignment of 0.4 s between the mirror temperature and mirror signal values provided in the telemetry frame. The black line is a gaussian smoothing of the Golden Points linear interpolation (with an equivalent sample size of about 25 s or 125 m), similarly to Sugidachi et al. (2025) for the upper troposphere. **(C)** Effect of correcting for a 0.4 s delay in the mirror temperature telemetry signal on the position of Golden Points and linear interpolation thereof **(D)**. SKYDEW nonequilibrium corrected mirror temperature, using RS41 time-lag and bias-corrected, and SKYDEW smoothed (panel (B)) as a-priori reference to determine $A$.

# 6   CFH nonequilibrium error statistics

In order to statistically evaluate the nonequilibrium error of CFH, we have analyzed a total of 70 RS41-CFH nighttime tandem soundings according to the procedure described in Section 4.2.3. A complete list of these 70 soundings is provided in the Supplementary Material (Table S1). Of these, 26 were performed during the StratoClim 2016-2017 campaigns in India and Nepal (Brunamonti et al., 2018), 40 during routine soundings in Lindenberg flown together with a COBALD backscatter sonde (Brabec et al., 2012; Vömel et al., 2016) between March 2016 and May 2019, and 4 during development and test flights for other instruments in Ny-Ålesund in February 2020 (Jorge et al., 2020). Except for one flight in Lindenberg, the CFH PID





controller version is '6.44', which is the standard version (since late 2014). Before data analysis, we eliminated flight segments that were rejected based on an NDACC quality check (Revision 0), e.g. due to mirror cleaning cycles, contamination or temperature/reflectance artifacts (NDACC, 2020; see also Vömel et al. (2016)). Furthermore, we restricted the analysis to night time measurements, since CFH daytime profiles tend to suffer from incomplete solar filtering and are more susceptible to noise

(Hall et al., 2016), and thus there is a higher risk of measurement artifacts, especially in the reflectance signal that would affect the quality of the reflectance-based statistical analysis.

From these 70 soundings, we show in Section 6.1 that there is a large film-to-film variability of the effective sensitivity parameter $A$, which makes a separate treatment of each sounding inevitable. Section 6.2 shows that the deviations between the reported mirror temperature and the estimated true frost point are typically better than 0.5 K (corresponding to uncertainties <

9% in the $H_2O$ mixing ratio in the lower stratosphere). However, at times when the mirror temperature deviates significantly from the true atmospheric frost point, deviations of more than 5 K are possible, which typically occurs when there are patches of coarse ice crystals on the mirror and/or when there are large changes in the atmospheric $H_2O$ mixing ratio. In these cases, the nonequilibrium correction can remove 80 % - 90 % of the nonequilibrium error. This, in turn, allows a reduction in the uncertainty (at a given vertical resolution) of the frost point estimates that occur under nonequilibrium conditions, and helps

discern true atmospheric features from measurements artifacts.

## 6.1 Large film-to-film variability of the effective sensitivity parameter $A$

Figure 11 shows the effective sensitivity parameter $A$ derived for the three condensate layer types liquid film, first and second ice film for each of the 70 flights. The values span about two orders of magnitude and vary from film-to-film and from flight-to-flight. This large variability does not allow the use of a fixed value of $A$ for all flights, not even for instruments that have

been recovered and flown again, using the identical mirror and optical hardware. This complicates the design of feedback controllers of consistently good performance, i.e. with the same sufficiently large $A$. Most likely, the variability is due to different film morphologies, themselves associated with the mirror surface properties (impurities, coatings, etc.) and to the prevailing temperature, ventilation, ambient $H_2O$ molecule number density, cooling rate, supersaturation and possibly coalescence/sintering experienced during film formation. Variability in balloon ascent rate might play a minor role as well, as the

boundary layer thickness of the airflow over the mirror is part of the effective sensitivity and depends on the flow conditions inside the inlet tubes. Over 95% of the derived effective sensitivity values are larger than 1000 V/s in the CFH data set we analyzed, which generally results in small mirror temperature deviations from equilibrium, as $(p_{vap}(T_m) - p_{vap}(T_{fp}) \propto 1/A$ $dU_m/dt)$.



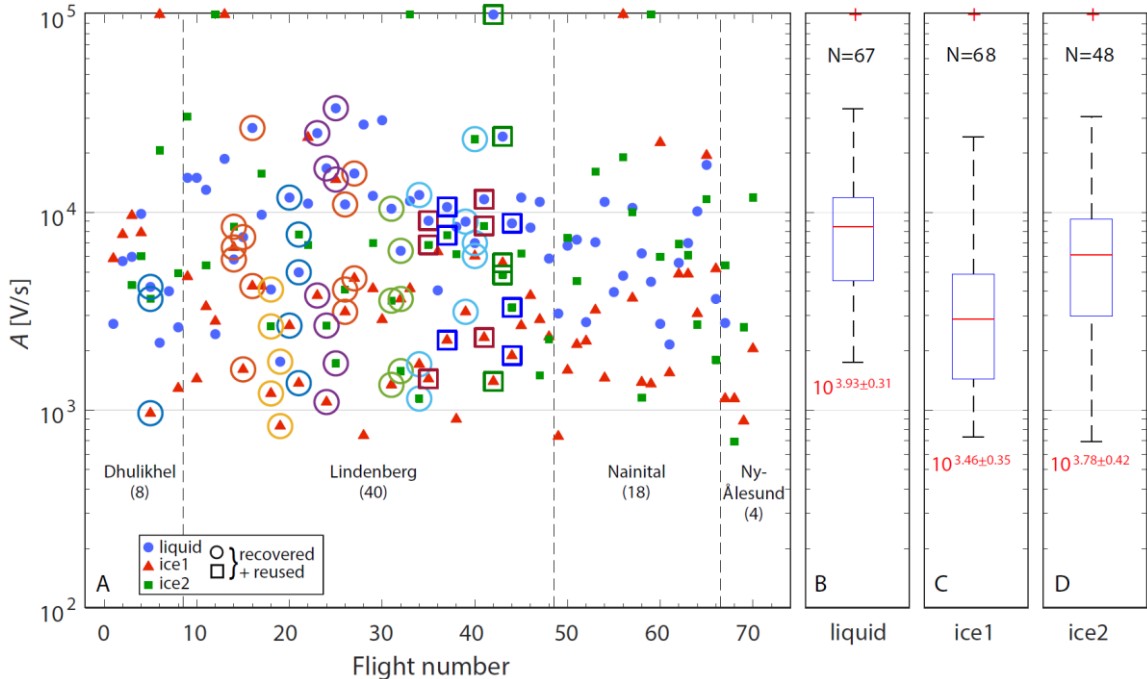

**Figure 11:** Sensitivities (in V/s) derived from 70 CFH-RS41 tandem soundings. **(A)** Sensitivity $A$ of each individual soundings for the liquid film (blue), first ice film (red) and second ice film (green). Soundings with recovered and reused instruments are shown with frames. The sensitivities span about 2 orders of magnitude and vary from film-to-film and from flight-to-flight. **(B)-(D)** Distribution of the sensitivities shown as boxplots in log-scale for each of the three condensate layers. The box shows the median (horizontal red line, second quartile $q_2$), the 25% ($q_1$) and the 75% ($q_3$) percentiles, i.e. 50 % of all data are in the interquartile range $q_3$-$q_1$ (blue box) The whiskers show the mild outliers of $A$ in logarithmic scale between 1% and 99% (or, more accurately, between $q_1$-1.5×($q_3$-$q_1$) and $q_3$+1.5×($q_3$-$q_1$)). Numbers in red display are the median +/- the robust standard deviation (defined as half the difference between the 84.13% and the 15.87% percentiles). The first ice film has the lowest median effective sensitivity (~2884 V/s), the liquid film the highest median effective sensitivity (~ 8511 V/s), and the second ice film an intermediate ~6026 V/s.

The first ice-film (triggered by the cleaning cycle at -15°C mirror temperature) has the smallest effective sensitivity, likely due to an ice film with fewer and larger ice crystals, which is potentially "patchier". Films with large ice crystals are known to deteriorate the detection limit and response time of optical frost point hygrometers (Pragnell, 1993; Vömel et al., 2016). The second ice film (formed at -53 °C mirror temperature) has the largest variability in effective sensitivity. This might be due to large flight-to-flight differences in prevailing conditions during condensate formation, mirror impurities, or increased variability in boundary layer thickness on the mirror, which increases as the Reynold's number decreases in the UT/LS (Jorge et al., 2021; see also Section 4.1.2). The liquid film (at mirror temperatures above -15°C) has the highest effective sensitivity and the least scatter in effective sensitivity, possibly because of a higher consistency of dew morphology.



## 6.2 Correction statistics of CFH vs. RS41

For each of the analyzed flight sections (liquid layer, first and second ice layer), the optimization solver described in Section
4.2.3 provides individual time-lag and bias parameters for the RS41, from which we calculate a corrected RS41 profile, the
effective sensitivity $A$ and the nonequilibrium-corrected CFH profile (and for liquid films also the time delay). Ideally, corrected RS41 and CFH profiles should coincide, as both sensors fly in tandem on the same balloon. However, measurement
noise, residual correction errors, contamination, and sub-scale atmospheric variability introduce some uncertainty. Nevertheless, after time-lag and bias correction of RS41, frost point differences > 1 K that occur sometimes are unlikely to originate
from a faulty RS41 humidity sensor or from sub-scale variability. Rather, in nonequilibrium regions, strong deviations of CFH
from true frost point are a cause of large discrepancies that can be corrected for.

Figure 12 shows the difference between individually corrected RS41 and CFH with and without nonequilibrium correction for
all 106814 data points. The data have been retrieved with 1-s resolution between the ground and the tropopause, where the
corrected RS41 provides measurements with reliable precision. In order to ensure flight-to-flight consistency in vertical reso-
1280 lution and improve the accuracy of the estimated nonequilibrium error when it is large, the boxcar filter parameter $N_{\text{boxcar}}$ is
kept constant (5 s for liquid films, 25 s for ice films) and the nonequilibrium error is derived from the vapor pressure space
instead of the frost point space (i.e. Eq. 5 instead of Eq. 6). The error due to the nonequilibrium of CFH (light gray points) is
generally small, but there are cases with errors of 1-5 K or > 10% in mixing ratio. These measurement errors (which have a
strong negative bias, as visible in Fig. 12B) benefit most from the nonequilibrium correction, which can reduce the error by
1285 up to 80% (change from gray points to black points in Fig. 12A). Measurements with the largest nonequilibrium errors occur
mostly in the presence of the second ice film (or UT/LS). They are clearly negatively biased (dry bias) due to the kinetics of
the ice film with slow condensate growth and faster condensate evaporation (Clausius-Clapeyron equation). However, the
mean bias of the original CFH measurements for the second ice film in these 70 soundings remains low, namely only -21 mK,
and reduces further to -6 mK after the nonequilibrium correction. Similarly, the standard deviation reduces from ±305 mK for
the original data to ±157 mK for the nonequilibrium corrected data and ±96 mK for the Golden Points.

The text insert in Fig. 12 indicates the mean bias and standard deviation, corresponding to the root mean square error (RMSE)
between corrected RS41 and CFH (original, corrected and Golden Points) for all three condensate types. For the second ice
layer (UT/LS) the nonequilibrium correction achieves about 45% reduction in RMSE, for the first ice layer 30% reduction,
and for the liquid layer about 20% reduction. Interestingly, for the liquid film the nonequilibrium correction is the least efficient
and the RMSE at the Golden Points is the largest. Despite the time-delay correction in the optimization routine described in
Section 4.2.3, the ± 1 s uncertainty in sampling the Golden Points likely plays a bigger role for the liquid film, as CFH typically
oscillates much more strongly (i.e. large $|dT_{\text{m}}/dt|$ at the Golden Points). In addition, the noise in the resistance measurement
induces an additional random uncertainty in the lower troposphere (Section 2.2.2).



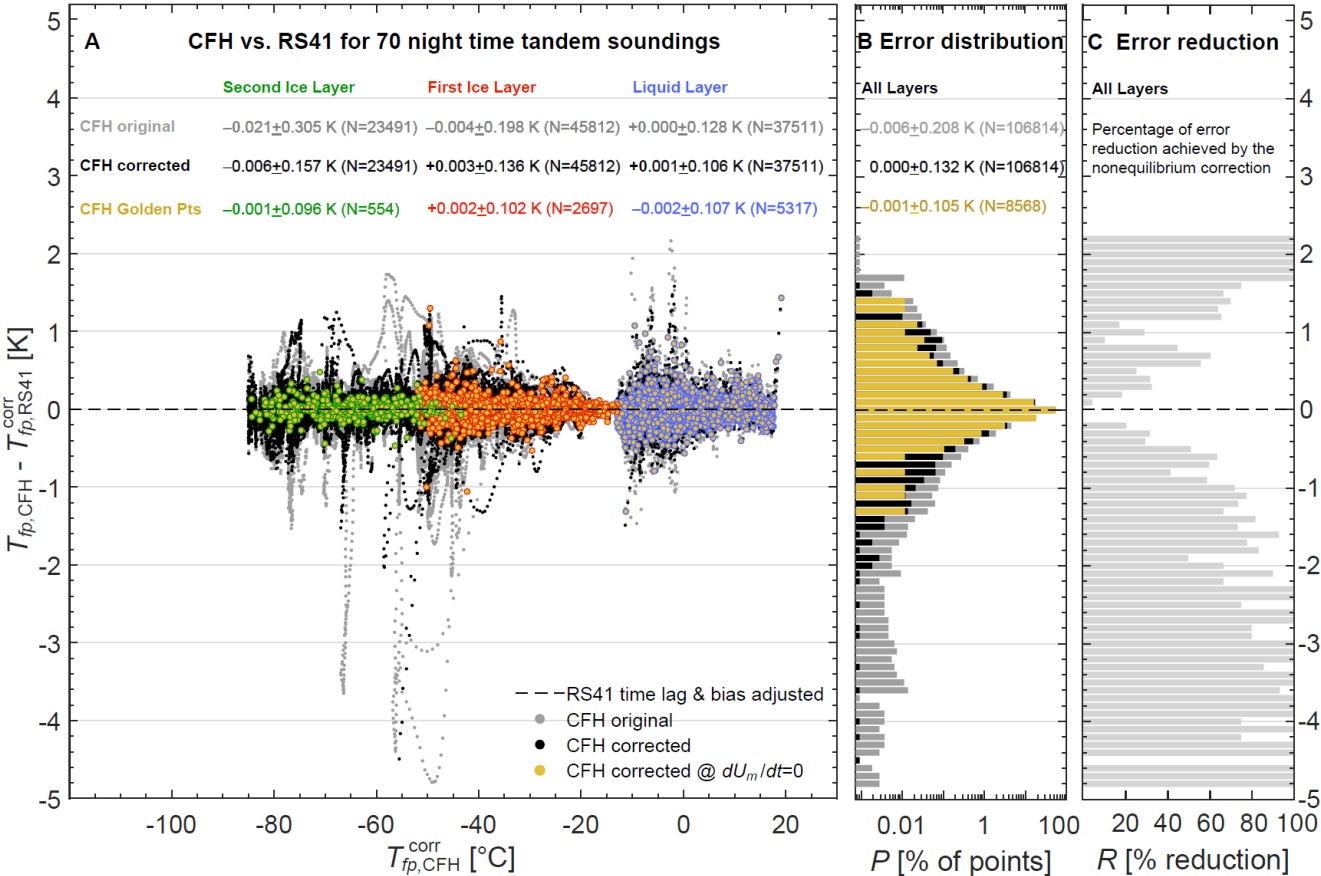

**Figure 12:** Error statistics based on 106814 CFH-RS41 tandem measurement data points. **(A)** Frost point differences (in K) between CFH (denoted by $T_{fp,\text{CFH}}$) and RS41, which was individually time-lag and bias-corrected by means of the CFH Golden Points ($T_{fp,\text{RS41}}^{corr}$), as a function of temperature. Gray points: uncorrected CFH mirror temperature ($T_{fp,\text{CFH}} = T_m$). Black points: nonequilibrium-corrected CFH ($T_{fp,\text{CFH}} = T_{fp,\text{CFH}}^{corr}$). Yellow circles with colored borders: CFH Golden Points for the different condensate types (liquid film blue, first ice film red, second ice film green). Text inserts: mean values ± standard deviations for each data type $T_{fp,\text{CFH}}$ and each type of condensate. **(B)** Probability distribution of frost point differences (in K) across all condensate types for original data (gray), nonequilibrium-corrected data (black), and Golden Points (yellow). **(C)** Reduction of the probability of a given error to occur in percent calculated as ratio of the occurrence with and without nonequilibrium correction $R = 1 - P(T_{fp,\text{CFH}}^{corr} - T_{fp,\text{RS41}}^{corr})/P(T_m - T_{fp,\text{CFH}}^{corr})$.

Figure 12B depicts the logarithmically plotted distribution of CFH-RS41 errors across all condensate types, demonstrating the effectiveness of the nonequilibrium correction (change from gray to black histogram) and the superior quality of the Golden Points (yellow histogram). In particular, note the effectiveness in eliminating most of the extremely dry outliers. The figure also reveals that nonequilibrium errors do not affect measurement accuracy when averaged over a large number of data points (- 6 mK bias over all data points over all layers), which is indicative of appropriate PID control on average, as noted by Vömel et al. (2016). The asymmetry in the distribution with a pronounced cold bias is due to the instrument needing more time to



increase the condensate thickness than to decrease it to meet the setpoint (owing to the asymmetry in the Clausius-Clapeyron equation). Lowering the setpoint of the feedback controller in the UT/LS might improve this issue (Sugidachi et al., 2025). The effectiveness of the nonequilibrium correction is summarized in Fig. 12C, which shows that nonequilibrium errors greater than 1 K are typically reduced by 60-90 % by applying the nonequilibrium correction. The remaining 10-40 % of the deviations could be attributed to various causes, such as contamination of the inlet, problems with data synchronization, uncertainties in $dU_m/dt$ due to electronic noise or stray light, or residual errors in the corrected RS41 data. Nevertheless, the effectiveness of the nonequilibrium correction remains striking.

## 7    Conclusions

Chilled-mirror hygrometers with precise SI-traceable sensors for mirror temperature and with transmission of synchronized, good reflectance data with a sensitive condensate film and a well-tuned feedback controller, such as CFH, serve as scientific reference instruments for measuring water vapor in the atmosphere. In addition, they are used to calibrate and/or validate other scientific instruments, such as Lyman-α hygrometers and absorption spectrometers on in-situ platforms, or microwave radiometers and Raman LIDARs on ground- and satellite-based platforms. This article shows how occasional measurements with errors in the frost point > 0.2 K, which are the largest source of uncertainty in chilled mirror instruments and are often oscillatory in nature and caused by controller instabilities leading to nonequilibrium states of the chilled mirror, can be systematically corrected.

Mirror temperature oscillations around the dew/frost point are inherent to the chilled mirror measuring technique and cannot be avoided. The mirror temperature is often the only data reported, while reflectance is traditionally considered only as housekeeping data and not used in scientific publications and data products (e.g., from NDACC). Based on this work, we advocate that the chilled mirror hygrometer measurement should be considered strictly as a composite of both the mirror temperature $T_m$ and reflectance $U_m$. Both, $T_m$ and $U_m$ should be reported and the $H_2O$ partial pressure should be calculated according to the nonequilibrium equation $p_{vap}(T_m) - p_{vap}(T_{fp}) \propto (p/A')(dU_m/dt)$ (Eq. 5). The reason for this is the often overlooked property of chilled-mirror hygrometers that the true dew/frost point is only measured when the condensate (liquid water or ice) is in equilibrium with the ambient gas phase, i.e. at the Golden Points ($dU_m/dt = 0$). Away from the Golden points, when the condensate is not in equilibrium ($dU_m/dt \neq 0$), the measurement error is proportional to the change in mirror reflectivity per time, independently of the controller action. When the reflectance increases, the condensate is evaporating and the mirror is too warm. Conversely, when the reflectance decreases, the condensate is growing and the mirror is too cold. The Golden Points are a powerful feature of chilled mirror hygrometers as they provide a physical confirmation of thermodynamic equilibrium. As such, they strengthen the use of chilled mirror hygrometers as reference instruments with high accuracy and negligible drift, suited for long-term records of $H_2O$ in the lower stratosphere.



Therefore, the central idea of this paper is the following: when CFH is flown together with an additional humidity sensor, whose temporal resolution is better than the spacing of the Golden Points, the additional sensor can first be recalibrated using the CFH Golden Points and the recalibrated sensor can in turn be used to correct the nonequilibrium states of CFH. Such a second sensor can be a high-performance thin film sensor like the RS41 or an optical hygrometer like FLASH-B. The recalibrated second sensor enables estimating the effective sensitivity $A$, which contains information about the frost (or dew) mor-

phology on the mirror and allows to correct the CFH nonequilibrium error. In contrast, attempts to consider cooled mirror instruments as time-lagged (i.e. following a first-order response) and correctable using a "time constant" are physically unfounded and generally inappropriate. This is because cooled mirrors are actively controlled devices and usually follow a higher order response, often with multiple overshoots. Attempts to remove nonequilibrium errors with downsampling techniques or low-pass filtering over several hundreds of meters are prone to bias in case of large and/or asymmetric nonequilibrium errors

and reduce the vertical resolution.

The morphology of the ice layer, the detection limit of the reflectance measurement and the tuning of the feedback controller are decisive factors for the chilled mirror response. Deviations from equilibrium can be corrected using the nonequilibrium correction (Eq. 5). However, even if measurements affected by hydrometeor contamination of the inlet tube or by the outgassing balloon skin or by strong electronic interferences are excluded, a number of causes for residual errors remain that are not

corrected by the nonequilibrium correction, e.g. uncertainties in data synchronization, in $dU_m/dt$ and in $A$.

In this work, 70 soundings have been analyzed revealing that the nonequilibrium correction achieves a reduction of the frost point error by 60-90 % for errors larger than 1 K. While a $2\sigma$ uncertainty of 10 % in mixing ratio or relative humidity is specified for chilled mirror instruments, such as CFH, this work shows that this uncertainty can be much exceeded in certain flights. However, at the Golden Points the $2\sigma$ uncertainty is less than 4 % in mixing ratio (for well-calibrated ambient air

temperature sensors and modern GPS-based pressure measurements). Away from the Golden Points, the use of RS41-assisted nonequilibrium correction enables a $2\sigma$ uncertainty during night flights of less than 0.4 K, or less than 4 % from the ground to the middle stratosphere when downsampled to 500 m vertical resolution. This shows that the target uncertainty of < 4 % in mixing ratio aimed by the WMO in 2023 is achievable. We therefore recommend combining instruments with different measurement principles, such as the RS41 capacitive thin-film sensor or the FLASH-B Lyman-α sensor and the CFH chilled mirror,

for research applications where low uncertainty is required, as they can be used to mutual advantage.

From an instrumental point of view, it is now clear that a small error, e.g. $|T_m - T_{fp}| \propto |dU_m/dt| / A' < 0.2$ K (see Eq. 8a), is obtained when the morphological sensitivity $A'$ is large, the change in reflectivity $|dU_m/dt|$ is small, and the frequency of Golden Points along the flight path is high. We note that simultaneous compliance with small $|dU_m/dt|$ and high frequencies of Golden Points is only possible if the amplitude of $U_m$ is sufficiently small. This, in turn, is hampered by the fact that even under very

stable control conditions, a small amplitude of $U_m$ needs to compete with inherent noise in the measurement electronics. This requires meaningful smoothing of $U_m$, which makes it difficult to determine the exact timing of the Golden Points; however,



overall, with small $|dU_m/dt|$, the measurement will benefit from small nonequilibrium errors $|T_m - T_{fp}|$ along the full flight path. Further, choosing a suitable reflectance set point (i.e., desired ice coverage), which is most sensitive to changes in ice thickness, and choosing a high cooling rate during ice formation after the cleaning cycles increases the morphological sensitivity $A'$ of

the ice films. Finally, maximizing $A'$ through an improved mirror design, such as a hydrophilic micropattern structure printed or deposited on the mirror, might increase the consistency and morphological sensitivity of the mirror condensate.

The hygrometer CFH (with standard firmware 6.44, cooled with R23) has in fact a large $A'$ and small $|dU_m/dt|$ under background conditions, often with several hundred meters between adjacent Golden Points. Conversely, SKYDEW (with standard firmware 1.0, cooled thermo-electrically) has only 10-20 m between adjacent Golden Points and a large $A'$, but also a very

large $|dU_m/dt|$ (due to the large oscillation amplitude). For CFH, a-priori profiles derived from the hygrometer's Golden Points alone are not useful since they do not capture the underlying atmospheric humidity features (see Fig. 6). Therefore, they require a second sensor on the same payload. For SKYDEW with fast oscillations due to an aggressive PID controller, the very large $|dU_m/dt|$ makes it difficult to correct small, sub-second synchronization errors between $T_m$ and $U_m$ (see Fig. 10). Here, low-pass filtering of the Golden Points track may be useful in atmospheric background conditions, but since low-pass filtering does not

necessarily correct for nonequilibrium (see Fig. 6 and Fig. 10), attempts to resolve fine atmospheric structures with low-pass filtering in general remains bias-prone.

As an outlook, it would be interesting in the future to investigate new hygrometer designs, which maximize the use of the Golden Points concept. New designs with live detection of the Golden Points by a microcontroller (i.e., scanning $T_m$ and $U_m$ at much faster rate than 1 Hz and sending these data in telemetric batches) could reduce the localization uncertainty of the

Golden Points and be useful for sensors with large $|dU_m/dt|$. Also, reducing $|dU_m/dt|$ by establishing an adjustable setpoint can reduce the nonequilibrium deviations and, thus, improve the performance of chilled mirror hygrometers. Autonomous nonequilibrium correction, i.e., generating mirror temperature oscillations at a fixed frequency (e.g. 1/60 Hz), and determining $A'$ such that this artificial (i.e. non-atmospheric) frequency is suppressed in the nonequilibrium-corrected data, is another approach for improved measurements.


**Code and Data availability.** The data related to this article will be available online at: https://hdl.handle.net/20.500.11850/732964 (Poltera et al., 2025).

**Supplement.** The supplement related to this article is available online at: doi:xyz/xyz-supplement.

**Author contributions.** Yann Poltera prepared the manuscript with contributions from all co-authors. Beiping Luo provided
support for the data analysis, Thomas Peter for developing the figures, and Frank Wienhold for providing much of the underlying data. All coauthors proofread the text.



**Competing interests.** The authors declare that they have no conflict of interest.

**Acknowledgements.** We gratefully acknowledge funding and support from the Swiss National Fund (grant no. 2000021_159950/2), the Swiss Federal Institute of Technology in Zürich (ETHZ), the University of Bern, MeteoSwiss, the GRUAN Lead Center, the Federal Office of the Environment (Project UTF-No. 705.25.22-PCFH), and GAW-CH & GCOS-CH under the "Swiss $H_2O$ Hub: High-quality water vapor measurements from ground to space" project. We are grateful to our colleagues at DWD (Lindenberg, Germany), IITM (Pune, India), ARIES (Nainital, India), DHM (Kathmandu, Nepal), KU (Dhulikhel, Nepal), AWI (Potsdam, Germany) and AWIPEV (Ny-Ålesund, Svalbard), LaCy (La Réunion, France) and ETH Zürich (Switzerland) for performing the balloon soundings analyzed in this work. We thank many colleagues for fruitful discussions, in particular Simone Brunamonti (Empa), Teresa Jorge (Hitachi Energy), Ruud Dirksen and Peter Oelsner (GRUAN Lead Center, DWD), Marion Maturilli (AWI), Holger Vömel (NCAR), Sergey Khaykin (LATMOS/IPSL), Ulrich Krieger (ETH Zürich), Bertrand Calpini and Gonzague Romanens (MeteoSwiss), and Thomas and Steven Brossi from mylab Elektronik GmbH (Switzerland).

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
