# Peer review of "The "Golden Points" and nonequilibrium correction of high-accuracy frost point hygrometers"

_EGUsphere, 2025_

## Referee Comment (RC1)

Review report

The "Golden Points" and nonequilibrium correction of high-accuracy frost point hygrometers
By Yann Poltera, Beiping Luo, Frank G. Wienhold, and Thomas Peter

This paper proposes a very active use of the mirror reflectance data (Um) of the balloon-borne Cryogenic Frostpoint Hygrometer (CFH) measurements to obtain much more information than previously done, resulting in less uncertainty and higher temporal/vertical resolution of the measured water vapor concentrations. The central ideas include the so-called Golden Points where the time derivative of Um becomes zero, and moreover, the correction for the data points between these Golden Points, the latter of which is termed as nonequilibrium correction. The theoretical background and actual implementation procedures of this nonequilibrium correction are fully explained and discussed in Section 4; and, the theory with several key assumptions look very reasonable to me. Then, the correction is applied to a large number of (~70) CFH sounding data taken from various locations from the tropics, through Northern Hemisphere (NH) midlatitudes, to NH winter polar region, showing that the proposed method actually results in less uncertainty and higher temporal/vertical resolution of the CFH measurements. I believe that this paper manuscript would mark a very important step historically for balloon-borne chilled-mirror hygrometry.

While the manuscript is very well written after Section 3, I was a little bit 'uneasy' while reading through Sections 1 and 2. This is mainly because there is no explanation on the Golden Points nor on the nonequilibrium correction until Section 3. Instead, there are rather lengthy (in-depth) reviews on atmospheric water vapor, observational challenges, etc. (I also notice that there is duplicated information in several places.) I am afraid that this style may even discourage some readers to read it through. I personally think that the authors can go more quickly to the definition of the Golden Points and to the explanation of the nonequilibrium correction in the manuscript. At least, at the end of Section 1, the authors should explain the definition of the Golden Points and the overview of the nonequilibrium correction and describe what will be discussed in each of the following sections. This will make the introductory part even longer, thus the authors may consider shortening the review part and removing duplicated information. (Please let me note that the review part itself is very interesting, but probably is less relevant to the main points of this manuscript.)

Please see below for specific comments and suggestions.

- Section 1: Please see the above comments for this section. (I do agree that this is a very

good set of reviews on the matter, but perhaps less for this particular manuscript.)

- Section 2: I think that the authors can shorten this part as well, describing only the key technical aspects of CFH, RS41, and FLASH-B which are directly relevant to the discussion in and after Section 3. (The main topic of this paper manuscript is the application of Golden Points concept and nonequilibrium correction.)

- Paragraph at lines 171-177: SKYDEW may also be mentioned already here.

- Line 213: Could you add a few more words to explain what does "auto-correlated error component" mean?

- Line 353: "resp." should be "or" (there are few other places where "resp." appears)

- Line 387: Sugidachi et al., 2025 is also a good example.

- Line 434: "less than 0.2 K error" – please add an explanation why (although this may have already been explained implicitly somewhere). Also, add "vapour" to water mixing ratio.

- Paragraph starting from line 438: The technical information on FLASH-B may be moved to Section 2, so that we can concentrate on the Golden Points concept itself here.

- Paragraph starting from line 466: The technical information on RS41 may be moved to Section 2, so that we can concentrate on the Golden Points concept itself here. Also, for RS41 data provided from the ground-receiving system, a time-lag correction may have already been applied within the manufacturer software. Does the sentence mean that the authors make an additional time-lag correction? In other words, do you mean that the manufacturer's time-lag correction is insufficient? Please clarify this in the text.

- Line 520: Please add one sentence to justify this simplification.

- Lines 563-573: This part is related to the Golden Points. I think that this may be moved to Section 3.1, so that we can concentrate on the nonequilibrium equation here.

- Line 587: The term "standard diffusion coefficient" should appear here, not at line 590.

- Section 4.2 (corrections to RS41 data) and Section 4.3 (corrections to FLASH-B data) may be moved to Section 2 or to Appendix/Supplement, because this manuscript mainly discusses the CFH data corrections (and the readers would be primarily interested in those, and probably not the details of the RS41 and FLASH-B corrections although they are important). Having rather long subsections here may disturb the logical flow.

- Lines 942-944: This last sentence, starting from "However" is not clear to me in terms of the logical relationship with the previous sentences. What is the point?

- Lines 1004-1005. I am confused here. Is 10% RH "high"? Probably, additional explanation is needed why this number can be regarded as high.

- Lines 1046-1047: "Thus, it is unclear whether PID control is the most appropriate approach to frost control of chilled mirror hygrometers.": Do you mean that there could be other, more sophisticated and/or appropriate control methods for upper-air chilled mirror hygrometry? Please add a few more words to explain this.

- Section 5.5.1, in particular, the part "High humidity and low number density of aerosol particles point to the polar vortex as a possible region of origin.": Just out of curiosity, I looked at a reanalysis data set MERRA-2 (https://gmao.gsfc.nasa.gov/reanalysis/MERRA-2/docs/) whether this could be a case. NASA/GMAO has a website, https://fluid.nccs.nasa.gov/reanalysis/, where we can quickly see weather charts, for example, of potential vorticity at 50 hPa over the Arctic region:
https://fluid.nccs.nasa.gov/reanalysis/classic_merra2/?stream=MERRA2&field=epv&level=50&fcst=20180419®ion=nps&tau=00&track=none

[Figure]

Modern-Era Retrospective Analysis for Research and Applications, Version 2 (MERRA-2)   **GMAO**
50 hPa Abs Potential Vorticity [PVU], Heights [dam]

Thu 04/19/2018 00Z

The above figure shows that airmass with relatively high PV values (maybe in the form of differential advection) was actually coming to the western Europe region. The maximum PV stream (colored in yellow) over Europe is located somewhat to the east, but this may be related to difference in time (00 UTC for this figure), actual balloon trajectory, the choice of 50 hPa, etc. More investigation would be needed by downloading and analyzing full model-level reanalysis data (e.g., MERRA-2, ERA5, etc.) to investigate whether the hypothesis is true, although I am not suggesting to do so for this paper manuscript.

- Line 1185: "0.4 s" – perhaps add "for this particular case"

---

## Author Comment (AC1)

**Review of**

**'The "Golden Points" and nonequilibrium correction of high-accuracy frost point hygrometers' by Poltera et al.**

**Author response to reviewer comments**

*We would like to thank the two anonymous reviewers and Takuji Sugidachi for their constructive and critical evaluations. Some of their comments were challenging and required considerable effort, but they also helped to improve the quality and readability of our manuscript. The reviewers' comments are listed below in black font, and our responses are in italicized blue font. We would like to thank our reviewers once again, as well as the editor for allowing us additional time for revision.*

**RC2 - Referee#2 Comment**

**'Comment on egusphere-2025-2003', Anonymous Referee #2, 20 Jul 2025**

**Citation: https://doi.org/10.5194/egusphere-2025-2003-RC2**

**Summary:**

The manuscript by Poltera et al. describes a different method for analyzing water vapor measurements using frostpoint hygrometers. The idea of their paper is that equilibrium points are defined as measurement points during which the reflectivity measured by the instruments does not change. They name these points golden points and use them to correct measurements by other instruments, such as the Vaisala RS41. After that, they use these corrected measurements to improve the CFH measurements between the equilibrium points.

This paper provides a novel idea and should help improve the analysis of frostpoint hygrometer observations. The authors recommend making measurements of the reflectivity public in quality controlled final data, most importantly the NDACC data set. This is a good suggestion.

However, the paper suffers from a lack of focus and a generally verbose style, which makes it hard to follow. The wordiness also hides some of the limitations of this approach and thereby its true value.

This manuscript requires major revisions, for which I detail my reasons below. Nevertheless, the idea is a significant progress for measurements of tropospheric and particularly

stratospheric water vapor. If the authors can agree to my recommendations, this study should become a significant contribution to this important observing technique.

*We would like to thank anonymous referee #2 for the critical and thorough but generally positive review as well as for the comments on our manuscript. Concerning the lack of focus and a generally verbose style, we have done our best by shortening the text and moving distracting but necessary parts into appendices, and hope for your approval.*

**Major comments**

**Major Comment #1 'not that different from low-pass filtering'**

1) In my opinion, the authors overstate their fundamental idea. Dew-point and frost-point hygrometry relies on the assumption that the temperature of the condensate layer is a measure of the vapor pressure in the overlying gas phase if a condensate layer does not change. This is also the definition of their golden points. It has long been recognized that this principle is difficult to implement and that an active controller must regulate the mirror temperature.

*Yes, $pH2O = pvap(Tf) = pvap(Tm) <=> dUm/dt = 0$.*

*However, it is difficult to implement, as the slightest disturbance in frost point (Tf) will change the mirror reflectance (Um), and the controller cannot predict the future in a balloon sounding with rapidly changing atmospheric humidity. However, the mirror reflectance signal allows us to determine when $dUm/dt = 0$ and thus $Tm = Tf$ is reached. In addition $dUm/dt$ can be used to correct the nonequilibrium error.*

In the early days, this was done manually (see for example the work by Brewer and colleagues) and later electronic implementations of that principle were built. In all implementations (manual or electronic readings), averaging is done to minimize the deviation of the amount of condensate from a pre-determined or manually observed amount condensate. The averaging then implies that the condensate temperature readings do not reflect the instantaneous vapor pressure, but rather a value averaged over the same time interval.

*Brewer and colleagues took the mean value of the mirror temperature when the condensate "just increases" and "just decreases", i.e., just before and just after a Golden Point. They did not average the mirror temperature over continuous time intervals.*

*If the averaging time interval encompasses large non-equilibrium sections, the averaged mirror temperature value does not necessarily represent the averaged frost point (i.e., it may be biased). The values of dUm/dt during the averaging time interval can be used to determine to what degree the mirror temperature average is biased.*

Their term "Golden Points" and the graphic representation implies that this averaging is not needed and that moments in time exist, when the instantaneous reading can be used to calculate the vapor pressure.

*Yes, in an ideal scenario, with very low noise in the mirror and temperature signals and high frequency data, averaging would not be necessary. However, in practice, the standard data rate of 1 Hz in balloon telemetry and the noise in the mirror and temperature signals require some averaging.*

However, in practice they find that this is not straight forward, and they again rely on averaging to smooth out the reflectance signal.

*Yes, averaging is required for reducing the noise in Um and thus reducing the uncertainty of the Golden Point timing, as in 'peak-finding' algorithms.*

They attribute the source of noise to electronic noise. However, it is more likely that this is noise in the overall measurement system, which includes the condensate layer. It should be possible to evaluate the contribution of electronic noise when no condensate is present, i.e. prior to filling cryogen and possibly during the mirror clearing cycles.

*This is a valid point. We have estimated the electronic noise also prior to cryogen filling. We find that the noise from the electronics alone is one order of magnitude less than the noise when ice condensate is present (for stratospheric stable conditions with good frost control). After filling, the noise combined together with the condensate could have multiple sources, such as 'sparkles' of the randomly oriented ice crystals facets when illuminated with a light source, mechanical vibrations from the payload motion, or stray light effects. However, the electronics of the instruments appears to remain a major source of noise under in-flight conditions. To illustrate this we added Figure D1, which shows a sounding with spectral noise of the condensate-substrate-electronics system with clearly visible regular oscillations in both the Tm and Um. In this case, the oscillations have a characteristic period of 5 s, in other soundings they can have a broad spectrum of periods, but are very often regular and visible by the bare eye. These oscillations are possibly due to some eigenfrequency of the controller and clearly not atmospheric in nature.*

*We estimate the detection limit of CFH in nighttime flights, due to both the electronic noise and condensate layer noise, to about 20 µV/s (Section 4.2) for good ice films.*

However, this does not solve the problem that averaging is needed to determine so called golden points.

*Yes, to remove false positives in the peak-finding algorithm and to obtain less noisy dUm/dt*

Since their method requires averaging as well, the term "points" is not quite appropriate and theirs is just another method of looking at averaged dew-points or frost-points. Calling this method "Golden" is somewhat overstated.

*We chose the term "Golden Points" to emphasize their importance in hygrometry, admittedly a bit flowery, like "whispering gallery", "sundog", or "ozone hole".*

*We call them "Golden" because the averaging is performed only in regions where $dU_m/dt \approx 0$, i.e. the averaged mirror temperature value has little to no residual bias as the controller-induced error is small. We call them 'points' because the net flux of $H_2O$ molecules over the mirror vanishes typically only for a very brief moment. The art is to capture these points.*

*The word "points" also emphasizes that the measurements result in non-continuous frost point profiles (i.e., with data gaps between points). Yet the data gaps can be filled by means of the nonequilibrium correction. We appreciate the argument that, due to the unavoidable smoothing, these cannot be points and cannot be accurate. However, thanks to the reviewer's critical eye we now perform a sensitivity study to examine the impact of choosing different half widths of the smoothing filters. We show that our results depend only weakly on the filter width. We added a considerable amount of new information on this (Section 5.5, Fig. 6C and Appendix D) and hope for the reviewer's approval.*

Any actively controlled system such as a chilled mirror hygrometer tries to minimize the error function $(U_m - U_{set})$, i.e. tries to find the point where $U_m = U_{set}$. Any deviation from that is used to counteract the disturbance.

*This is correct.*

In mathematical terms, $U_m \neq U_{set}$ is almost always true for instantaneous readings, but averaged over time, $=U_{set}=const$ and therefore $d/dt = 0$. This is equivalent to the smoothing of the detector signal done by the authors to find the points where $d/dt$ goes through zero.

*We disagree with this statement. The smoothing we apply on the reflectance signal $U_m$ (15 s after 2nd cleaning cycle) is typically far less than the smoothing required to keep $U_m \sim$ constant over time. Already our Fig. 2 shows this clearly, as do all other reflectance plots in the manuscript. In contrast, applying the smoothing necessary to make $avg(U_m) \sim$ constant can lead to a strong reduction in the vertical resolution of frost point hygrometers and possibly bias them too.*

*One of our statements is that Golden Points rarely occur at $U_m=U_{set}$, and the average of $U_m$ over short time intervals around equilibrium (i.e. Golden Points) is still generally not equal to $U_{set}$. Equilibrium occurs only when $dU_m/dt=0$.*

It is also true for any PID controller, which minimizes the error signal. Their misconception may lie in confusing instantaneous readings (either of the reflectance or of the mirror temperature) and readings after "sufficient" smoothing.

*We are not sure we understand what you mean, but we don't think this is a misconception. Our implementation of the Golden Points relies on the not yet smoothed 1-s CFH data. We have to apply some smoothing to eliminate the high-frequency noise and locate the Golden Points. Finding peaks in a noisy signal requires some smoothing, but this is more accurate than smoothing the signal without first searching for positions with dUm/dt = 0. We detail this in Appendix D.*

Brewer (A W Brewer et al 1948 Proc. Phys. Soc. 60 52), has an excellent description of the details of the instruments they used to make the first stratospheric frostpoint measurements. Brewer described the challenge of visually deciding, whether a condensate remains constant, when time constants were on the order of a minute. Their electric detector was a significant improvement and allowed a better determination of constancy of the frost-layer.

*Thank you for the reference, which is similar to the two years older Bakerian lecture. We are aware of it, it is an excellent description indeed.*

In effect, "Golden Points" is only a new name for the chilled mirror principle, which had certainly been around since the 1920s (see some references in Brewer et al, 1948), although not yet suitable for stratospheric dryness. That achievement is clearly that of Brewer, Dobson, and colleagues. However, they are not the inventor of the chilled mirror principle or any "Golden Points" by a different name.

*Controlling the mirror **temperature** such that the frost layer remains constant on 'average' requires oscillating around the frost point. The physical principle of chilled mirror hygrometry imposes this type of control. The extremes of the mirror **voltage** are the Golden Points. Dobson et al. (1946) detected those extremes not by measuring a voltage but visually, and then used the mean. However, with the advent of automatic instruments, the significance of the points with dU/dt = 0 may have been deemphasized. By referring to them as Golden Points, we would like to draw more attention to them again.*

*To our knowledge Dobson et al. were the first to consciously perceive the importance of ensuring that "the deposit is seen just to increase and just to decrease" (Dobson et al., 1946, bottom of p. 147). The references in Brewer et al. (1948) are either on dew point measurements at high temperatures or on frost point measurements but letting the condensate disappear and use the temperature at disappearance as frost point. Therefore, we continue to think that they are likely the inventors of the "Golden Points" method.*

Previous instruments worked on the same principle, but at warmer temperatures. To all scientists of that generation, it seems to have been clear that the necessary condition was a condensate that remains constant. The challenge was how to achieve that.

*The answer is that it is impossible to keep the condensate constant, as it is impossible to predict future changes along a balloon trajectory. Instead, oscillations around the frost point are inevitable, which means that the instrument only measures the actual frost point when dU/dt = 0 is satisfied. This fact is exploited in the Golden Points method.*

Having said that, there is great value in looking at the deviations of the reflectance from the expected value and to try to estimate a correction of the mirror temperature based on that error signal. This has to my knowledge not been done and should help particularly for instruments, which have larger uncertainties than others such as some CFH instruments or the SkyDew instruments.

*Thank you for the comment. We note here again that 'deviations of the reflectance from the expected value', i.e., Um-Uset, is not a measure of the frost point error, but dUm/dt is, and this is what we use.*

**Major Comment #2 'missing uncertainties'**

2) In several meteorological and metrological communities, establishing well justified uncertainties is as important as the measurement itself. This aspect of the measurement process is ignored by the authors, and they do not estimate any uncertainties in their method. This leaves the reader, who has no other information than the claims of the authors, without a metric whether their method is better than previous work. I see great potential in this method and believe it may improve what has been done so far. But without uncertainties it is difficult to compare their results with previous efforts. The authors ignored the work that was done in the NDACC data set of frostpoint based stratospheric water vapor measurements, even though they are clearly aware of it and use these profiles. In that data set, the data have been filtered and smoothed using a variable Gaussian filter, which the authors acknowledge or averaged into 250 m layers. However, the authors ignored the uncertainty estimates provided in this data set and not having any uncertainty estimates of their own, it is hard to quantify the benefit of their work.

*Thank you for your comment. This is an important argument that we are very concerned about and motivated to address.*

*In Section 4.2, we wrote that "we use the deviation between the a-priori and the a-posteriori as an estimate of the uncertainty of the frost point resulting from residual errors in the nonequilibrium correction", and we presented the resulting uncertainty for several specific soundings (green and black lines in Fig. 2, green and blue lines in Fig. 4, green and black curves in Fig. 6) and statistically for 70 soundings in Fig. 12.*

*The deviation between the a-priori and the a-posteriori provides a first metric for the uncertainty of the Golden Points method.  However, we do agree with the reviewer that this*

*does not allow for a complete assessment of the uncertainties in our methodology, nor does it allow for a comparison with the uncertainties provided by NDACC, because it does not test the assumptions made in the Golden Points method.*

Plots of differences for example should include uncertainty estimates that were provided to bring the differences into context. That would then allow them to discuss the details and potential shortcomings of the existing uncertainty estimates and propose a better uncertainty treatment. It could well be that their method is superior to previous work, but just saying so is not sufficient.

*We have now added Section 5.5 to the paper that discusses the uncertainty of our nonequilibrium correction method related to different assumptions underlying the method. We have also added the NDACC uncertainties to Figure 6 to enable a comparison of both methods.*

*Figure 6 showcases the limitations of low-pass filtering if the mirror reflectance information is not taken into account, and that it can lead to residual biases that are not covered by the NDACC uncertainties.*

*The NDACC dataset is extremely valuable, and we believe it could be further improved if the reflectance data were systematically included in the dataset and used in their processing routine. Figure 6 is an incentive for this. We refrain from performing further analyses on the NDACC data in this article. We believe that the NDACC processors are best suited to implement the use of reflectance in the NDACC processing pipeline.*

*The asserted 0.2 K uncertainty of the Golden Points (on frost layers, 0.3 K on dew layers) takes into account (in quadrature) the 0.11 K uncertainty of the thermistor measurement and < 0.17 K (2σ) uncertainty in Golden Point sampling (±1 s sampling uncertainty), as discussed in Section 5.5. The 0.2 K value is in line with the uncertainties with stable PID control stated by Fahey et al. (2014) and Vömel et al. (2016).*

**Major Comment #3 'text length'**

3) I strongly urge the authors to reduce the length of their manuscript. I estimate that the total page count can be cut in half without any significant loss of information. This should help the reader get a better insight into the details of this method.

*We have endeavored to reduce the length of the manuscript and we moved parts to Appendices in order to improve the readability. Sections 1 and 2 have been shortened by about 45 %, but overall, the reduction of the manuscript without the appendices is only 24 %. The reason for this is that we saw only few ways to shorten the core technical sections*

*without compromising comprehensibility and because we added new material such as the discussion of uncertainties.*

The majority of the manuscript focuses on the Cryogenic Frostpoint Hygrometer (CFH). They also discuss Skydew as another frostpoint instrument, and FLASH as another reference. To strengthen their discussion, I would suggest completely removing the discussion of FLASH (FLASH could be mentioned in one sentence in the conclusion section). I also suggest moving the Skydew discussion to the very end or potentially to delete it as well. It adds little to the value of their approach as a whole.

*Thank you for your suggestions. Following your advice we have moved the discussion of FLASH to Appendix C, where we also discuss the new offset correction for FLASH (Eq. C1) developed in this work.*

*SKYDEW has a very different PID controller than CFH, with a strongly oscillating behavior producing a large number of Golden Points. The SKYDEW discussion sheds light on the sub-second timing uncertainty in sampling the Golden Points when using a strongly oscillating but higher vertical resolution PID controller. This timing uncertainty is difficult to showcase on the CFH instrument, except perhaps on pre-launch time-series with large and nearly symmetric oscillations such as in Figure 3. We consider the discussion of SKYDEW to be valuable to the community (see e.g. [https://www.gruan.org/fileadmin/Resources/Public/Lists/Launches/Launches/SKYDEW_launch_launchList.csv](https://www.gruan.org/fileadmin/Resources/Public/Lists/Launches/Launches/SKYDEW_launch_launchList.csv)). In fact, both anonymous referees and the community commenter are intrigued by this timing uncertainty.*

To further reduce text, the authors should look at repeated discussions of the same point and consolidate all of these into one single discussion. I will provide examples below but cannot list all.
*We thank the referee for listing repeated discussions.*

**Detailed comments:**

Abstract, first sentence. Here the authors claim that they "introduce new retrieval protocol for … measurements in the upper troposphere/lower stratosphere of unprecedented accuracy". This claim is repeated in line 418 and again in line 218, here at least with the caveat "at the equilibrium points". However, nowhere in the manuscript do they actually discuss the accuracy of their method. They only show differences from the (I assume NDACC) data that without referring to the uncertainties provided in these files. Their discussion should be refined by deriving an accuracy of their method and comparing it with the existing estimates. This is clearly doable but has not been done.

*We claim the accuracy to be unprecedented under rapidly changing conditions (abstract 1st sentence), because with a well-calibrated CFH hygrometer, an accuracy better than 4% (or < 0.2 K in frost point) can be achieved from 0 km to 28 km at the Golden Points. This is because the controller nonequilibrium, which dominates the error budget (Vömel et al., 2016; Hall et al., 2016), is very small at those equilibrium points. Furthermore, the nonequilibrium correction reduces uncertainty away from the Golden Points, allowing layer averages of higher vertical resolution and with smaller residual errors than was possible previously.*

*We have now added a subsection in the paper (Section 5.5) that discusses the uncertainty of the nonequilibrium correction method and added Fig. 6C for the comparison with the uncertainty of the low-pass filtering used by NDACC.*

What they term as accuracy is the difference between the raw CFH measurements and corrected CFH measurements based on a comparison with another sensor, where the correction of both sensors has been combined into one higher order algorithm. This is not an accuracy estimate.

*To our knowledge, there is no other hygrometer that can achieve < 4% uncertainty in the UTLS in flight mode. The mirror temperature is SI-traceable, and therefore the Golden Points inherit SI-traceability but with a larger 'in-flight' 0.2 K systematic error than the 'laboratory' 0.11 K systematic error. Recalibration of RS41 or FLASH by means of the Golden Points is one of the few methods capable of providing a quality reference away from the Golden Points. In our opinion, the difference between the corrected auxiliary sensor and the corrected CFH profile, in addition to the CFH temperature accuracy (< 0.11 K), gives a very reasonable estimate for the accuracy of the final frost point product.*

Lines 29-31: This sentence represents the lack of understanding of the NDACC frostpoint data, which report uncertainty estimates. This statement should be made within the context of the uncertainties and vertical resolution reported therein. Are the large deviations that they find, covered by the larger uncertainties reported in the data files? I assume the large deviations also go along with large uncertainties in these files. If their method can reduce this uncertainty, or if their method shows that the existing uncertainty estimates appear to be incorrect, then this would be a very valuable contribution.

*Many thanks to the reviewer for this comment. At this point, we discuss CFH mirror temperature and mirror reflectance and it was not our intention to refer to the uncertainties and vertical resolutions provided by NDACC. However, upon your suggestion, we made an uncertainty estimate of nonequilibrium corrected frost point data for a flight with high nonequilibrium deviations and compared them with NDACC data (see Figure 6). Panel C shows the uncertainty calculated from the Golden Points method and nonequilibrium correction and reveals that the present NDACC uncertainty is incorrect for this sounding with an ice film of low sensitivity. We extended the non-equilibrium correction for this sounding also into the entire stratosphere (see Fig. E1). Nonequilibrium deviations of the*

*mirror temperature from the corrected frost points with amplitude larger than 1 K can be corrected. This confirms the detection of dehydrated layers in the lower stratosphere, a feature that was also suggested by RS41, but without correction could not have been identified from the CFH data given the vicinity of nonequilibrium oscillations.*

*Indeed, it would be useful to apply the presented method to the whole NDACC chilled mirror dataset in the future, and correct the occasional nonequilibrium deviations similar to cases presented in this work.*

Lines 185ff: This would be the place to introduce the uncertainties that are explicitly provided in the NDACC data files, rather than just citing upper estimate values from publications.

*Thank you. We have added this information.*

Line 206: Equating the amount of condensed material with "layer thickness" is not valid. The authors should not give the impression that variations in reflectance are equal to variations in "layer thickness". As they are aware, condensate can grow along preferred nucleation sites (mirror imperfections) or along small temperature gradients within the mirror. How well operators clean the mirror is also likely to play a role. The current frostpoint hygrometers are unable to derive a layer thickness from the reflectance. The term in parenthesis should be deleted here.

*We fully agree and have removed the controversial expression, and added the mirror imperfections and mirror gradients aspects in Section 2.3.*

Line 230ff: Do the authors imply, that the time lag correction applied by Vaisala is incomplete and hence apply a new correction? Is that a correction on top of a correction or a correction using the Vaisala raw data?

*The correction is applied using the Vaisala raw data (GRUAN/GUT, GRUAN/GU in Vaisala MWX files). We have clarified this in Section 3.2.*

*Vaisala (MW41) applies a time-lag correction, but it has residual errors due to time-constants assumed too large in certain temperature regimes (about -40 C to -60 C). In fact, they have recently (2022-08) updated the time-lag correction algorithm in the new MW51 telemetry and processing system for the RS41 radiosonde (https://www.vaisala.com/en/sounding-data-continuity).*

Line 298: Vömel et al. (2007) quote a range "between 0.1 K for well-behaved instruments to 1.0 K" for slightly unstable instruments.

*Thank you. We have amended the text accordingly*

Line 379ff: It may be worth mentioning again, that Vömel et al (2007) quote a value of "0.1 K for well-behaved instruments", implying that this may be a rough uncertainty for "well behaved" instruments, much less than the several K referred to here. While this is clearly not stable in a mathematical sense, it would meet WMO criteria. The authors show an example for such a profile in Figure 7. The statement "a chilled mirror hygrometer is therefore unable to keep the condensate in static equilibrium…" is too strong. Deviations of 0.1 K seem to be close enough to be able to call this "equilibrium" using the WMO criteria for measurement accuracy.

*We appreciate the comment. However, maintaining a static (or continuous) equilibrium is impossible since there are always disturbances in the atmosphere. Further, oscillations around the frost point are required for a responsive measurement and avoiding the risk of a blank or 'snowballed' mirror. The statement is correct, even for the high quality profile in Figure 7A, as oscillations and Golden Points are clearly detectable, although the deviations are less than 0.1 K. The Golden Points on such profiles have minimal possible error.*

Line 382ff: The term "transient equilibrium" feels like a contradiction in terms.

*We appreciate this comment. Although transient equilibrium might feel like a contradiction, strictly speaking it is not. Under the conditions of a Golden Point, the H2O molecules in the gas phase and in the condensate "are present in concentrations which have no further tendency to change with time, so that there is no observable change in the properties of the system" (Atkins, Phys. Chem.). If H2O molecules were forced to condense and empty the gas phase, they would tend to diffuse back and reestablish the equilibrium because p_vap > p_partial, and vice versa. At the Golden Points, the Gibbs free energy is at its minimum value. Only external forcings by the changing environment along the ascent trajectory or by controller instabilities would change this situation. And there is no physical constraint stating how long an equilibrium state needs to persist to be called an equilibrium state.*

*This said, we are not interested in a fundamental battle at this point and relaxed our statement considerably. We now highlight that these are points when "the vapor pressure of the condensate and the H2O partial pressure are equal (pvap = pH2O)" and talk about "equilibrium-like moments". Please see our revised text.*

Please note my discussion above, that the authors use averaging to define the equilibrium. Figure 10B shows that the equilibrium condition is not always perfect. In this case the authors explain the deviation by the insufficient temporal resolution and shift the profile by a fraction of a second. A similar effect can happen within the CFH at slower and well resolved oscillations if they are large enough. That may point towards the limitation of the assumption of equilibrium within the mirror system. While this does not contradict the basic idea, which I fully support, it warrants caution calling such points "golden". Here, the authors need to smooth the reflectivity signal by 15 s, which is within the range of oscillations by the CFH. This means that instantaneous measurements of the equilibrium (points) are insufficient. This entire discussion should be re-written.

*The Golden Points is a fundamental thermodynamical principle at the core of chilled mirror hygrometry. The principle remains valid on real measurement data, but many aspects (such as limited temporal resolution, noise, etc.) lead to uncertainties in the Golden Point temperatures. Our averaging by 15 s of the reflectance is typically required to eliminate high-frequency noise (electronic and other), see Fig. D1. This maintains the structure in the reflectance (i.e., $dU_m/dt \Box T_m - T_{fp}$).*

Lines 262ff: Is it a good assumption that the reflectance is directly given by a change in the frost coverage in ug/cm2 if it also depends on how the frost coverage is distributed? It is possible that a redistribution of frost-coverage without a net loss of ice might lead to a change in the detector signal. Why else would there be an influence by the ice morphology?

*The situation that you describe is interesting. It would violate assumption (i) in line 312 of the preprint, i.e., reflectance change reflects mass change, and thus one of the fundamental assumptions of chilled mirror hygrometry.*

*We argue that the large sonde-to-sonde variability in sensitivity seen in Fig. 11 is reflecting different states of ice morphology, which affect the response of $U_m$ to changes in the mass of the condensate. Therefore, effects of morphology are properly described by the morphological sensitivity A' (Eq. 2) and included in our analysis. In our understanding, large and/or 'patchy' ice crystals on the mirror have a low sensitivity compared to a collection of many small ice crystals on the mirror. In other words, as long as the morphology does not change within the condensate layer (water, ice I and II) during a single sounding, morphological differences between soundings are fully taken into account.*

*The remaining question is what happens when the morphology changes during a single flight segment. This could be either slow changes, such as ice sintering, or sudden changes, e.g. when an ice crystal breaks off. While sintering is likely too slow to matter, breakoffs, based on our experience, do occur, but rarely, and translate into a sudden jump in reflectance that is relatively easy to detect and should be flagged (e.g. until the next Golden Point occurs).*

*For these reasons, we believe describing the amount of condensate as ugcm-2 and the sensitivity in mV/ugcm-2 (or in mV/mol cm-2) to be reasonable.*

Line 315: It could probably be added that the instrument can fly in the wake of the parachute and balloon.

*Thank you for the comment. We have added this.*

Lines 422f: Please define what you mean by "fine structure". Is that structure on the scale of meters or many 10s to 100s of meters?

*Thank you for the comment. We mean on a scale < 100 m, i.e. the GCOS 'breakthrough' requirement. We have added this.*

Lines 424ff: This discussion compares instruments, which do not have quantified uncertainties (likely large and understood biases) with an instrument that has quantified uncertainties. This sentence should somehow reflect that.

*We do not think that this is really necessary, as Fig.2 shows clearly by how much FLASH and RS41 need to be corrected before they reach the same league as CFH, and we discuss this further in the text of Section 3.2.*

Line 434: Is that uncertainty taken from the data file (actually estimated) or taken from a publication (nominally)? The NDACC data files do provide these numbers.

*We reformulated the text and write now: "CFH has been shown to measure the frost point with less than 0.2 K error under equilibrium conditions in the laboratory (Fahey et al., 2014) and under good flight conditions (Vömel et al., 2016), which we assume to also apply also to the equilibrium situation of the Golden Points (see further Section 5.5 for a discussion of the uncertainty at the Golden Points)."*

Section 3.2: It is a good approach to correct a slower responding instrument such as the CFH using data from a faster responding instrument (FLASH) after it has been bias corrected. The minor challenge is that it is not absolutely clear where the FLASH biases come from, but as long as that instrument has sensitivity, this should work. For the RS41, this may be a little tricker to implement. The RS41 itself is a slow instrument in the region of interest and has been time lag corrected. That may introduce unreal structures in the vertical profile that are a result of the correction. Furthermore, the RS41 does lose sensitivity in the stratosphere. The manuscript gives the impression that the RS41 data could be used up to burst altitude. Please add a discussion up to what altitude this instrument could be used.

*We fully agree regarding the unknown origin of the bias of FLASH, but clarifying its origin would be beyond the scope of this paper. With regard to RS41, we have added a sentence mentioning that RS41 loses sensitivity below ~ 2% RH and thus (depending on location and season) may not be usable for altitudes much higher than the tropopause. On the other hand, further investigation is needed into how well the sensitivity determined with RS41 for the second ice film can be used deep in the stratosphere (as demonstrated for example in Fig 8 and new in Appendix E).*

Lines 482: This statement points at the true value of their work, how to minimize the uncertainties away from the equilibrium points using additional information coming from the detector signal.

*We appreciate the comment.*

Line 505: Why do the authors assume an area density to describe the amount of condensate? That implies a homogeneous condensate layer and wouldn't properly describe a patchy or coarse condensate layer, which is mentioned at other places in the manuscript. Describing the condensate layer by the number of molecules could be more appropriate.

*The quantity MH2O times the area covered by the condensate gives the amount of condensate, thus the number of molecules. However, the condensate \*density\* provides more direct information about the condensate coverage than the total amount. A mirror with a larger surface area would need more condensate, but the same MH2O to provide the reflectance behavior. We therefore don't see a need to change units.*

Line 508 and many other equations: The authors mix equations that are written on their own line and equations that flow within the text. It would be easier to follow if all equations were written on their own and given an equation number. Why do the authors introduce A' before A? Can they introduce another letter to avoid confusion between both?

*Thank you for the comment. The equation A' = – dUm/dMH2O defines the quantity A' and is so short that the manuscript would become much longer if all equations of this type were in display mode. Of course, this is a choice of style and nomenclature…*

Lines 511ff: The assumption that A' is constant over time is probably true over short time periods, but probably not over the time of an entire ascent, or even just stratospheric ascent. How will that affect their correction algorithm?

*See our response to your question regarding lines 262ff regarding changes of A' in the troposphere.*

*With regard to changes in the stratosphere, we address this in Section 5.5.1 (now Section 5.6.1). It is important to take into account the increase in thickness of the Prandtl boundary layer in the less dense stratospheric air. From the flights we have analysed, the assumption that A' is constant once an ice film is formed and is not subject to abrupt changes (e.g. tear off of pieces of the ice film) seems to hold.*

Section 4.1.2: The thickness of the diffusion layer depends on the speed of the air flowing across the mirror. Due to the pendulum motion of the payload, the air flow through the tube is likely the vary somewhat. During descent, the airflow is likely to be much different, changing the thickness of the diffusion layer significantly. Therefore, this parameter and others that include it cannot be taken as a constant. How much does it vary within a sounding?

*Indeed, we fully agree. The thickness of the diffusion layer varies with altitude and flow speed and we give a formula with a coarse estimation of it.*

*Within each segment (liquid film, first ice film, and second ice film until the tropopause, all during ascent) the thickness of the diffusion layer does not vary much (~ $1/\sqrt{\rho v}$). This may change when using the A' for the second ice film also deep in the stratosphere, where RS41 is no longer reliable (see Appendix E).*

Lines 565ff: The number of equilibrium layers depends largely on the controller. The performance of the controller depends on the last three items in this list. It is an active controller, and the performance is not independent of these.

*Thank you for the comment. We have changed 'performance' with 'setting' (or 'tuning') of the feedback controller.*

The variation of the airflow through the tube is a simple external disturbance, it is neither "good" nor "poor" for the balloon instruments.

*Agreed. Thank you for the comment. We have deleted this.*

The morphology is not just fine versus coarse frost, it is also the distribution of frost on the mirror (patchy versus homogeneous).

*Thank you for the comment. We have added this.*

Line 583: Where does the estimate of 250 um come from? It is not justified in section 4.1.2.

*It is typical for the upper troposphere, as we state in Section 4.1.2, using the equation provided there.*

Line 589: I would not call A constant, when the authors devote an entire section to discuss its variability. It's a parameter.

*Thank you for the comment. We agree. We now introduce A as a parameter.*

Lines 606ff: This is a key issue with the so called "golden points". I agree that the noise in the measurement system must be smoothed. A 21 s filter is about the time period of the controller response, i.e. a fair amount of the noise in the system is filtered out in this step. Filtering is important, but that makes "points" into "averages" and points no longer represent instantaneous measurements but rather time intervals. This is what happens in all frost-point or dew-point hygrometers.

*From a thermodynamic point of view, Golden Points are instantaneous points in time. Feature-preserving smoothing (Savitzki-Golay) is used to determine most precisely their position, as in peak finding algorithms. (We apologize for the 21 s, which should have been 15 s; we corrected this now.)*

Line 618: Where does this uncertainty (+/- 0.2 K) come from? In the caption of Figure 3, a value of +/- 0.3 K is stated.

*The +/- 0.2 K is from Fahey et al., 2014; Vömel et al., 2016 and Hall et al., 2016 in "good conditions", i.e. near equilibrium, as stated in Section 2.3 and now further described in Section 5.5 and Appendix A.*

*The thermistor error increases in the lower troposphere, where the resistance of the mirror thermistor is low for a constant ohmic noise (see also Hall et al., 2016). This is why the Golden Points uncertainty is 0.3 K for the liquid films, 0.2 K for the ice films.*

By their definition, the uncertainty at the golden points should be exactly 0 K. There is of course the calibration uncertainty; however, it is not relevant here. Since they average around the equilibrium point, the accuracy of that point has to be slightly degraded. Is that the source of their uncertainty? As pointed out earlier, the estimate of that uncertainty needs to be discussed in greater detail.

*0.2 K uncertainty are values stated in Fahey et al. 2014 and Vömel et al., 2016 in "good conditions", i.e. near equilibrium,, and is reasonable, as shown by adding in quadrature 0.11 K temperature uncertainty and < 0.17 K uncertainty at the Golden Points (for +-1 s timing uncertainty), which is a conservative estimate with CFH's PID controller on ice films.*

Line 624f: This has even been done away from the equilibrium points through averaging.

*Averaging sometimes may mimic well the effect of a nonequilibrium correction, but not always. Sections 5.3 and 5.4 are examples for this.*

Line 631: Vaisala is operationally correcting for time lag. Is there a reason not to use that?

*Yes. The Vaisala MW41 sounding processor corrects for time-lag, but from our experience, the correction is not adequately parametrized (Poltera, 2022, thesis). Vaisala have changed their time-lag correction on the MW51 sounding processor in 2022 (https://www.vaisala.com/en/sounding-data-continuity), however we do not have MW51 data in our data collection analyzed in this work, and thus we cannot state about the performance of the new Vaisala sounding processor.*

Do the authors know, what time constant Vaisala is using? If they are correcting the Vaisala RS41 data based on raw data, do they use the relative humidity at sensor temperature or that at ambient temperature? What smoothing is applied to the RS41 time lag corrected data? How does their time lag correction compare to that applied by Vaisala?

*We have estimated the RS41 time constant used by Vaisala MW41 by comparing raw and final data. About 10 s at -40 C and 100 s at -70 C sensor temperature, with exponential relationship, similarly to RS92 (Dirksen et al., 2014).*

*We apply the time-lag correction on the relative humidity at sensor temperature, not ambient temperature. Then smooth the result (boxcar of 25 s) and convert back to frost point. Our time-lag correction usually differs from that of Vaisala in the -30 C -60 C temperature regime, with our time constants being about 40% smaller.*

Line 648ff and Figure 3: The data in this figure were taken prior to the launch of the balloon with very different ventilation, with liquid water at condensate, and with constant humidity and constant temperature. This never happens in flight. This figure is good for showing the basic idea, but at the same time may hide the complications in flight, when ambient humidity and temperature are constantly changing. Can they show a similar example from a typical flight segment? Can the parameter A be derived as cleanly? What would be the uncertainty

and time dependence in flight? In the same context, it is probably not appropriate to talk of "monolayers", since there is no monocrystal of H2O on the mirror. Furthermore, how can the authors translate this parameter into a "absolute sensitivity" of the hygrometer? And how can they compare this "absolute sensitivity" to a precision during one lab campaign? Absolute sensitivity and precision are very different things.

*These are many good questions and we appreciate the reviewer's curiosity. Figure 3 was indeed included "for showing the basic idea". As you say, it is not easy to find a segment in flight of similar quality. But here is an example under good conditions:*

[Figure]

*Of course, the kinetics of a second ice film in flight is slower than the kinetics of a liquid water film on the ground  (only four Golden Points while Fig. 3 shows ten). Also, the range of $T_m$ - $T_{fp}^{RS41,corr}$ is much smaller, because the ventilation on the ground is poor and here we select a segment with excellent controller performance with deviations less than $\pm$ 0.2 K with a fairly compact scatter plot for A.*

*In contrast, the scatter plot for the determination of A in the Lindenberg sounding in Fig. 4 shows a bizarre in-flight excursion (for both FLASH and RS41). We looked into the reasons for this behavior. It is probably due to the fact that the payload exhibits enormous changes in ascent rate varying between 4.0 and 8.0 m/s with corresponding changes in ventilation, resulting in 30-% changes in $\Delta x_{diff}$ and A within just a few 100 meters. This is likely the reason for the looping features observed in the scatter plot.*

*Concerning the other questions.*

*Monolayer: we deleted this sentence as it might confuse readers.*

*Absolute sensitivity: we replace this difficult term by precision of CFH: "With detectable changes (at S/N = 2) in CFH reflectance of about 20 $\mu V/s$ during nighttime flights, this means (using Eq. 8a with B = 3 $x10^7$ K and $dU_m/dt$ = 20 $\mu V/s$) that CFH can achieve measurements with a precision of 0.1 K in frost point or 100 ppbv H2O in the lower stratosphere at 1 s temporal resolution (or 40 ppbv at 5 s resolution)."*

*Lab campaign: in principle, for a lab campaign such as AquaVit-1 the same relationship between $T_m$ - $T_{fp}^{RS41,corr}$ and the sensitivity of the CFH ice film exists and can be utilized.*

*Precision: yes, the uncertainty of the Golden Points method is now better described in Section 5.5.*

Line 687: Do the authors use the measured sensor temperature or, as written, an approximate ambient temperature plus 5 K?

*The measured sensor temperature. We have added this information (now in Appendix B).*

Section 4.2.3: This section indeed contains a new and interesting aspect. Here, they make use of information (the reflectance) that has previously been ignored.

*We appreciate the comment.*

Line 703: How many flight segments are there? From the previous descriptions I assume 3, i.e. liquid condensate, ice between -15 C and -53 C, and again ice below -53 C. However, that may not be correct. In particular, in the stratosphere, the RS41 loses all sensitivity, and this approach can no longer be used. I'm sure the authors considered that and filtered the data accordingly. However, I could not find that part explained.

*Three segments (liquid film, first ice film, second ice film stopping at the tropopause to avoid RS41 sensitivity loss issues), as stated at the beginning of Section 4.2.3 (now Appendix B).*

Line 704: Here, the manuscript mentions a 31-point second order Savitsky Golay filter. The caption to Figure 5 mentions a 15-point and 21-point (half width) filter. Are these indeed different? Does the filter width vary between soundings?

*31 points full width = 15 points half-width, which is equivalent to 15-s boxcar resolution (second-order Savitsky-Golay, see also Dirksen et al. (2014, their Eq. A4)).*

*Figure 5 is a special case with a filter-width tailored for this particular stratospheric ascent/descent comparison. Our standard processing would use 15-point half-width second order Savitsky-Golay to smooth the reflectance data. The uncertainty of smoothing of the reflectance data with different half widths is described now in Section 5.5.*

Line 805: The suggestion of using pre-launch data to estimate the instrument parameters is not appropriate. This requires that the sensor is ventilated, which it normally isn't. Furthermore, the instrument is not yet in full thermal equilibrium, which can create additional artifacts. Using in-cloud data also seems to be risky. CFH operations typically avoid liquid clouds, where 100% is a good assumption. In cirrus clouds, this assumption of ice saturation no longer holds.

*Thank you for the comment. You are right, this does not make sense. We have deleted this bullet point.*

Lines 860ff: The discussion about the temporal resolution is important and should be expanded. The method by the authors may indeed improve the temporal resolution over the smoothing that is done by other groups. Just using equilibrium points limits the resolution to the controller oscillations. Using what they call non-equilibrium correction may indeed improve that. It would be very beneficial, if they could define and quantify the temporal resolution for their method.

*Right. The temporal resolution is 25 s (or 125 m), with boxcar, except in few case studies. The reflectance data is smoothed with 15 s equivalent (i.e. 31 s full-width), 2nd order Savitzky-Golay, except in few case studies. This is stated now in the new Section 5.5 and also Section 6.2.*

Lines 882: This is an interesting discussion. Clearly, the frostpoint hygrometer shows some oscillations. As they point out, for example in the region between 11.1 km and 11.2 km, the mirror temperature appears too cold. They should also point out, that in this region, the reflectance seems to decrease (growing condensate), which is consistent. It also appears, that the rate of condensate growth is related to the level of cold bias, which is also consistent. They point this out 20 lines later, but this seems to be the main benefit of looking at the reflectance, which should be highlighted at the beginning of this section (or even much earlier in the paper).

*Thank you for the comment. We discuss this in Section 4.1.4.*

Section 5.3: This is an important point but should be shortened to bring out the true value of their correction approach.

*We agree and have shortened Section 5.3.*

Line 1050: Where does the number of 0.2 K come from? Does it refer to the WMO criteria?

*From Fahey et al., 2014; Hall et al., 2016; Vömel et al., 2016 for very stable frost control, i.e. near equilibrium, as stated in Section 2.3 and now also in Appendix A. The uncertainty in sampling Golden points is now described in the new Section 5.5.*

Lines 1053ff: Here they compare the benefits of their method to the raw frostpoint measurements. However, it would be more instructive, if they compared that to the NDACC data product. What is the benefit their method could bring to that data set?

*Deriving statistics of residual errors in the whole NDACC dataset is certainly an interesting undertaking, but goes beyond the scope of this paper. See discussion within Major Comment#2 above. Section 5.4 compares nonequilibrium correction to low-pass filtering, and the new Fig. 6C provides a comparison of the (qualitatively very different) uncertainties*

*of the Golden Points method and the uncertainty specified by NDACC for one sample sounding.*

Figure 7: Here they end the correction at the hygropause because of the higher reliability of the RS41 below. This point should be expanded, since it refers to the altitude up to which this correction may be used. Can the authors add the actual water vapor profile and indicate the tropopause altitude?

*The tropopause altitude is the end of the slabs, as the ability of RS41 rapidly decreases at higher altitudes (except for the Ny-Alesund 2020-0221 and Maïdo 2022-0122 cases and the new Fig. E1, which examine the applicability in the stratosphere).*

Section 5.5.1: This is an important section, since it indicates the strength of their approach for stratospheric measurements. It can be shortened and strengthened at the same time.

*Thank you for the comment. We have shortened the section and added another example in Appendix E.*

Line 1085: This is the first time that an altitude limit for the RS41 is mentioned. The limit of the RS41 and how they extend the approach using tropospheric data is important and should be highlighted earlier.

*Yes, this could have been clearer. We now point out in Section 3.2 that RS41 has detection limit of about 2 %RH and suffers from increased sonde-to-sonde variability in the stratosphere.*

Figure 8, Caption: Move all commentary into the main text. In particular, justify the RS41 limit in the main text in its separate discussion.

*Thank you for the comment. We believe this is a matter of style.*

Lines 1137ff: This entire paragraph is an interesting discussion, but not relevant here. What is missing is how the authors applied their correction in the absence of any RS41 or other reference data. That is explained in the following section (5.6) for Skydew. However, that section ends with the sentence that for the CFH it is better to use a second reference instrument. It is not explained how they did the correction shown in Figure 10.

*We have added an explanation (reference $H_2O$ value of 5.5 ppmv estimated from COBALD and Aura/MLS background values) that was missing from the original manuscript. We apologize for that.*

The non-equilibrium discussion of section 5.6 for Skydew disrupts the flow of the discussions for the CFH and does not contribute there. This could be addressed in two ways. Finish the entire discussion for the CFH (including section 6) and then elaborate on Skydew and its differences to CFH. Alternatively, expand section 5.6 to explain how this works for the CFH

and how the data shown in section 5.5.2 were created. In that case, it would be better to move section 5.6 before of section 5.5.

*Thank you for the comment. We believe it fits well as a final case of the overarching "Section 5: Nonequilibrium correction under various conditions"*

Lines 1180ff: Since the mirror temperature of the Skydew oscillates faster and with a stronger amplitude, have the authors considered that the basic equilibrium assumption may not be satisfied. Because of the fast and large oscillations, the entire system may not be in sufficient equilibrium and even measuring the temperature at much higher frequency may not solve that challenge. The ad-hoc time shift makes it work here, but this shift has been arbitrarily selected to make it work.

*As we mentioned above, there is no physical law stating how long an equilibrium needs to last before calling it an equilibrium. Therefore, there is no need for static or continuous equilibrium for chilled mirror hygrometry. Short moments in time with zero net flux suffice. The Skydew measurements suggest that reporting values at higher frequency and solving the Tm/Um subsecond timing issue could further increase the vertical resolution for Skydew measurements. However, you are right that the time shift was ad-hoc to make it work, and it is up to the Skydew PIs to investigate if a generally valid synchronization can be established.*

Lines 1195: Here the authors say that there is "good agreement", yet that there are "finite errors, whose quantification are beyond the scope of this paper". At the end of the paragraph, they state that for the CFH another sonde as reference is the best option. These statements are made without justifications and do require quantifications of errors. I don't believe they are beyond the scope of this paper; they should be at the core of this paper.

*Skydew' Golden Point uncertainties have been quantified by Sugidachi et al. (2025). The agreement is better than 0.15 K. We have adapted the text accordingly.*

Their example for the Skydew correction is in the upper troposphere, where the Vaisala radiosonde is probably still a very good reference not needing much correction. How does that work in the stratosphere 10 km higher at much colder mirror temperatures? Their section leaves the impression that it will always work without needing a second reference instrument.

*Right. We change the text to: "... due to the steep slopes dTm /dt at the Golden Points, even with synchronized Um and Tm, the noise in the measurements leaves SKYDEW's Golden Points with finite errors, that can be quantified from Fig. 10C with ± 1 s Golden Point timing uncertainties to be about 1 K in this case (see also Section 5.5)."  The Skydew correction in the upper troposphere may work in the stratosphere as well, especially under background conditions, but as you mention, the RS41 could not be used to validate the procedure 10 km higher.*

Line 1231: It is just a hypothesis that there are patches of coarse ice crystals on the mirror causing the large oscillations. There could also be a slew of other reasons causing this

behavior causing controller instabilities. I would suggest deleting that hypothesis. It would be interesting to know how often the large changes are related to sharp gradients.

*We specify this and write now: "Greater deviations can occur when the morphology of the ice on the mirror is poor (e.g., coarse ice crystals), resulting in low A' (Eq. 2) and/or if the H2O mixing ratio in the atmosphere changes significantly."*

Lines 1233ff: Can the authors compare that to the smoothing done in the NDACC data set? Such a statement would clearly point to an improvement compared to what has been done so far.

*We show such a comparison now in Fig. 6C and the new Section 5.5 provides the background for the differences between nonequilibrium correction and low-pass filtering.*

Line 1275: The authors should clarify, that RS41 data have several corrections applied. These may also contribute to the differences between the instruments. Faulty sensors is a poor choice of words here.

*We have removed the sentence and clarified these points.*

Lines 1278: Do the authors imply here, that only tropospheric data are shown in the statistics?

*Yes, because of the higher uncertainty in RS41 data in the lowermost stratosphere, which would have an impact on the nonequilibrium error statistics (i.e., the nonequilibrium error statistics could be misrepresented because of the lesser quality of RS41 past the tropopause). We have adapted the sentence and refer to Appendix Figure B1, which showcases the lower quality of RS41 measurements in the stratosphere.*

Line 1284 seems to imply that also data in the lower stratosphere are used. Figure 12 shows stratospheric frostpoint temperatures in regions where the RS41 is likely not to be sensitive. Please clarify the exact data range that was used.

*As mentioned in the text, all the data is from the troposphere (ground to the 'thermal hygropause', which is very similar to the WMO tropopause). For data from India and Nepal, frost point temperature < -80 C are found in the uppermost troposphere, with relative humidities far above the sensitivity limit of the RS41 (~2 %RH).*

Lines 1295ff: See earlier comment about the lack of equilibrium conditions in the mirror system of the Skydew. This applies for CFH as well.

*As mentioned above we do not share the opinion that there is a lack of equilibrium conditions in the mirror system of SKYDEW or CFH or that the Golden Points are not equilibrium points. Rather, we believe sub-second timing issues should be addressed by instrument developers, and > 1 Hz data rates might be useful as well.*

Line 1297: The noise in the resistance measurement can be evaluated prior to filling the cryogen. It is unlikely to contribute to the uncertainty.

*Yes, it should unlikely play a role. However, it could contribute to the random uncertainty in the lower troposphere, Hall et al. 2016 (their Table 2) and Vömel et al. 2016 (their Sect. 2.2) describe this uncertainty component and suggest it is not negligible (~0.1-0.2 K additional random uncertainty in dewpoint) in the lower troposphere.*

Figure 12 seems to indicate that the by far largest improvement of the corrections is for large outliers. This is an important contribution. Figure 12B is a little hard to interpret. It seems to imply that about 99% of all data have only a small improvement. But for the remaining 1% of all data, the improvement can be quite significant. How does that compare with earlier statements about the importance of the correction? Maybe I'm interpreting that figure incorrectly.

*Thank you for the comment. Figure 12B+C show that the method reduces significantly the largest errors (which are mostly negatively biased). We state in Section 6.2 (previously in Section 5.5) that 89 % of the measurement points (at 125 m vertical resolution) have an error due nonequilibrium error < 0.2 K and 98 % < 0.5 K , which is excellent performance for a chilled mirror hygrometer. Nevertheless, the remaining 11 % are important.*

Line 1311ff: Here they implicitly repeat the basic chilled mirror assumption. Averaged over many points, a working hygrometer measures the averaged dew-point or frost-point temperature. The method of smoothing has only little influence. However, here they could make the same comparison using the NDACC data and evaluate, which method is better using this metric.

*We meant "when averaged over a large number of soundings". We apologize for this mistake.*

Lines 1318ff: "The remaining …" This is highly speculative. I would suggest just deleting this sentence.

*We have removed the sentence.*

Line 1321: Stray light should not be an issue here, since the authors only used nighttime soundings. If they indeed included daytime soundings, it would be good to see the analysis separated by day and night. That would truly show the influence of stray light.

*We did not include daytime soundings in our analysis and removed the reference to stray light.*

Line 1327: Can the authors please quantify "occasional" in terms of percentage of soundings, not percentage of data points?

*We removed the "occasional". For the subset of points near the tropopause, 32 % of flights had a frost point error < 0.2 K. We state this now in Section 6.2.*

Line 1336: Here is another repetition of the assumption that the so called "golden points" are instantaneous measurements, where in fact, they rely on averaging, similar to other averaging methods.

*See discussion regarding the Major Comment #1 above*

Lines 1353ff: This is only stated here. Only one short profile section was used to make this argument (which could have been fortuitous), but no robust comparison with existing data was done. The paper would benefit greatly, if they supported this statement with a solid statistics using for example the NDACC data set.

*See discussion regarding the Major Comment #2 above*

Lines 1364: The uncertainty estimate of 4% is stated here but was not shown in the paper. What is the basis for this number? I must clarify that I agree that the uncertainty may well be reduced but understanding what the basis for their uncertainty estimate is will better support their claim. It may also point towards what are fundamental limitations and where could this technique be improved.

*We discuss this now in Section 5.5. The uncertainty assumes <0.2 K in frost point and <0.3 hPa RS41's pressure from GPS, and results in less than 4% uncertainty in mixing ratio (see also Hall et al., 2016)*

Line 1367: This is the first time they mention down sampling to 500m. That is extensive averaging, and the resulting uncertainty may be substantially lower than their estimate. How did they calculate their number?

*We discuss this now in Section 5.5.* It is an ad hoc estimate using the 0.4 K uncertainty for the nonequilibrium correction at 125 m vertical resolution. With 500 m = 4 x 125 m, the uncertainty of the 4 x 125 m averages is sqrt(1/4) x uncertainty of 125 m averages = 0.5 x 0.4 K = 0.2 K → 4% i n H2O mixing ratio.

Lines 1385ff: What do they mean by "a priori profiles". This sentence needs rephrasing, since I don't believe they want to say that "CFH a-priori profiles derived from the hygrometer's Golden Points alone are not useful".

*Thank you for this comment. We meant, an a priori profile derived from the linear interpolation of Golden Points are not useful in cases such as Fig. 6. There, the atmosphere had several structures that a Golden Points interpolation would have missed.*
*The paragraph has now been deleted in an effort to shorten text length as it had many repeats from previous sections.*

Text length:

In the following, I give an incomplete listing of text that can be combined and/or deleted. There are more and the authors should make an effort shortening their text and providing more focus.

*We thank the referee for this list of text that can be combined and/or deleted.*

Abstract: The entire abstract could be cut in half if some of the discussion therein would be limited to the main text and not repeated here.

*We appreciate the comment. This is a matter of style.*

I fully agree that water is a very important trace gas in the atmosphere. Yet, in the context of this lengthy manuscript submitted to a technology focused journal, I would suggest deleting section 1.1.

*We fully agree and have deleted Section 1.1*

Section 2.1.1 can be combined with section 2.2, which goes into further detail, and the combined section can be shortened significantly by referring to the appropriate original work.

*We have grouped non-consecutive 2.1.1 and 2.2 now into consecutive sections 2.2 and 2.3 for improved flow readability*

Section 2.1.3: The FLASH section appears to be out of place here and interrupts the flow of argument of the frostpoint hygrometers. Similarly, the FLASH section 4.3 interrupts the flow of the argument and contributes little overall. FLASH instruments are probably no longer available and may no longer play a role in observations of stratospheric water vapor. Since FLASH contributes little to the argument, it could be mentioned in a single sentence in the conclusion section of the paper or completely deleted. The argument of the paper on chilled mirror hygrometers would become stronger.

*We have moved the FLASH section to the Appendix.*

*We consider the FLASH discussion to be valuable for the argument of the paper on chilled mirror hygrometers, in particular, FLASH is used to validate the RS41-derived nonequilibrium correction, as shown in Figure 4, and thus strengthens the argument.*

Section 2.2.2 can probably be deleted and replaced by a reference to the relevant previous publications. The few points that are relevant here could be combined with section 2.1.1.

*We have moved Section 2.2.2 to Appendix A. The information there is still relevant to understand where the 0.2 K uncertainty at the Golden Points comes from.*

Lines 491: Another example of text that can be deleted, since it has been discussed and referenced earlier.

*We have deleted it.*

The introduction to section 4.1, lines 496 to 502 are a repeat from earlier.

*We have shortened this text.*

Line 510: Here they defined $\Delta U$ for liquid and ice using different scaling factors but make no use of that distinction later in the manuscript. In fact, other uses of $\Delta U$ are without the scaling factor. This line could just be deleted.

*We have removed $\Delta U$ and simplified the text.*

Lines 622f: This is one example of unnecessary cross-reference and preview, that can be deleted.

*We have adapted the text.*

The entire section 4.4 can be deleted. All points are discussed in detail later.

*We appreciate the comment, but do not share this opinion. The section fits well within the whole logical structure of Section 4: i) Equations, ii) Correction using auxiliary sensor, iii) Correction without auxiliary sensor,*

Lines 867ff: These are repetitions from earlier and can be deleted.

*We have shortened but not deleted the entirety of the text. We think it is a helpful illustrative argument that Um=Uset is not a synonym for equilibrium.*

Lines 992 to 1006 are repeats and can be deleted.

*We have shortened the text.*

Lines 1060ff: This is a repeat from earlier and can be deleted.

*At this point, we have kept the text even if there are some repetitions, as we believe it is useful for supporting Figure 7 coming right after.*

Line 1081: Delete "This is further …". This point is made in more detail two paragraphs later. There, the detailed discussion of the COBALD data is distracting and can be shortened.

Lines 1101: Since the ECC failed, this discussion adds nothing to the water vapor measurement. The ECC discussion can be deleted

*We have deleted "This is further …". We have grouped COBALD, ECC and RS41 discussion into one single paragraph and shortened the text.*

Lines 1124ff: The chemistry discussion is distracting and not relevant for the water vapor measurements. It can be deleted.

*We have deleted it.*

Line 1231: "… deviations of more than 5 K are possible …" says the exact same thing as the previous half sentence. This can be deleted.

*We have deleted it.*

Lines 1269fff: This has been described before and can be shortened/deleted here.

*We have shortened the text.*

Figure captions: In most figure captions, the text should be reduced to the minimum needed to explain the graphs. Any interpretation of these graphs should be moved to the main text and consolidated with the explanations there.

*We appreciate the comment. This is a matter of style.*

**Technical details:**

Line 16: Delete "true"

*We have deleted "true"*

Line 360: Delete "an instrument similar to". This was after all a chilled mirror hygrometer that was built specifically for that purpose.

*We have deleted "an instrument similar to"*

The introduction to section 3 could be shortened and combined with the previous discussions in sections 2.1.1 and 2.2.

*As mentioned above we take a slightly different view of the early hygrometer measurements by Brewer and Dobson. We appreciate the comment but left this intro unchanged.*

Lines 399fff: Can you provide an example for this misconception? Otherwise, don't use the term "common misconception".

*We have changed the sentence and replaced it with: "The effect of the feedback controller could lead to the misconception that Um = Uset is a sufficient condition for equilibrium, i.e., Tm = Tfp."*

Throughout the manuscript, please make sure to refer to figures either using Figure X or Fig. X, but not both.

*We appreciate the comment. We follow the general rule for scientific publications to use "Figure X" at the beginning of a sentence and else "Fig. X".*

In addition, make sure that the Figures are printed soon after their first reference (Figure 3 is referenced on page 23, but shown on page 30, Figure 4 is referenced on page 27 and printed on page 32, Figure 5 is referenced on page 33 and printed on page 35).

*We appreciate the comment and make an effort to achieve this.*

Line 652: Better use "average", "mean", or "median" value instead of "canonical".

*We have replaced "canonical" with "average" value.*

Throughout the manuscript the authors use the terms "accuracy", "precision", "error", "uncertainty" and "sensitivity" almost interchangeably. In the context of their work, they need to be more careful, in their wording, since they have different implications (systematic, random, combined uncertainties).

*We appreciate the comment and make an effort to choose the most appropriate terms.*

Line 785: "As in this sounding" refers to which sounding?

*The sounding on 13 March 2017 shown in Figs. 2 and 4. We have added this information.*

Line 943: Change "thickness" to "reflectance"

*Now removed. That and the previous sentence have now been deleted, as it confused readers.*

Line 1046: Change "whether" to "which"

*We truly mean "whether" here. There might be control systems other than PID that could be applied to chilled mirror hygrometers (i.e., minimizing |Um-Uset| is not the same as minimizing |dUm/dt|). We have a 'Golden Points' seeking algorithm in mind that does not rely on PID but we have not tested it yet. We have added this in the text (At the end of Section 5.4)*

---

## Author Comment (AC2)

**Review of**

**'The "Golden Points" and nonequilibrium correction of high-accuracy frost point hygrometers' by Poltera et al.**

**Author response to reviewer comments**

*We would like to thank the two anonymous reviewers and Takuji Sugidachi for their constructive and critical evaluations. Some of their comments were challenging and required considerable effort, but they also helped to improve the quality and readability of our manuscript. The reviewers' comments are listed below in black font, and our responses are in italicized blue font. We would like to thank our reviewers once again, as well as the editor for allowing us additional time for revision.*

**RC1 - Referee#1 Comment**

**'Comment on egusphere-2025-2003',  Anonymous Referee #1, 01 Jul 2025**

**Citation: https://doi.org/10.5194/egusphere-2025-2003-RC1**

Review report

The "Golden Points" and nonequilibrium correction of high-accuracy frost point hygrometers

By Yann Poltera, Beiping Luo, Frank G. Wienhold, and Thomas Peter

**Summary**

This paper proposes a very active use of the mirror reflectance data (Um) of the balloon-borne Cryogenic Frostpoint Hygrometer (CFH) measurements to obtain much more information than previously done, resulting in less uncertainty and higher temporal/vertical resolution of the measured water vapor concentrations. The central ideas include the so-called Golden Points where the time derivative of Um becomes zero, and moreover, the correction for the data points between these Golden Points, the latter of which is termed as nonequilibrium correction. The theoretical background and actual implementation procedures of this nonequilibrium correction are fully explained and discussed in Section 4; and, the theory with several key assumptions look very reasonable to me. Then, the correction is applied to a large number of (~70) CFH sounding data taken from various locations from the tropics, through Northern Hemisphere (NH) midlatitudes, to NH winter polar region, showing that the proposed method actually results in less uncertainty and higher temporal/vertical resolution of the CFH measurements. I believe that this paper manuscript would mark a very important step historically for balloon-borne chilled-mirror hygrometry.

*Thank you for this overall positive review.*

**Comment #1 'text length'**

While the manuscript is very well written after Section 3, I was a little bit 'uneasy' while reading through Sections 1 and 2. This is mainly because there is no explanation on the Golden Points nor on the nonequilibrium correction until Section 3. Instead, there are rather lengthy (in-depth) reviews on atmospheric water vapor, observational challenges, etc. (I also notice that there is duplicated information in several places.) I am afraid that this style may even discourage some readers to read it through. I personally think that the authors can go more quickly to the definition of the Golden Points and to the explanation of the nonequilibrium correction in the manuscript. At least, at the end of Section 1, the authors should explain the definition of the Golden Points and the overview of the nonequilibrium correction and describe what will be discussed in each of the following sections. This will make the introductory part even longer, thus the authors may consider shortening the review part and removing duplicated information. (Please let me note that the review part itself is very interesting, but probably is less relevant to the main points of this manuscript.)

*Thank you for your careful judgement. After reading our original manuscript again, we have to agree with you and apologize for some extensive review material. We have shortened Sections 1 and 2 by 44 % and start the introduction with a historical link to the Golden Points. And at the end of Section 1, we now provide an overview of the paper so that we hopefully do not deter any readers.*

**Specific comments and suggestions**

Please see below for specific comments and suggestions.

- Section 1: Please see the above comments for this section. (I do agree that this is a very good set of reviews on the matter, but perhaps less for this particular manuscript.)

*We have shortened Section 1 by about 46 %.*

- Section 2: I think that the authors can shorten this part as well, describing only the key technical aspects of CFH, RS41, and FLASH-B which are directly relevant to the discussion in and after Section 3. (The main topic of this paper manuscript is the application of Golden Points concept and nonequilibrium correction.)

*We have shortened Section 2 by about 40 %. Technical details CFH, RS41, and FLASH-B have been moved to the Appendix.*

- Paragraph at lines 171-177: SKYDEW may also be mentioned already here.

*We have added SKYDEW and corresponding reference (Sugidachi et al., 2025).*

- Line 213: Could you add a few more words to explain what does "auto-correlated error component" mean?

*We have added a few more words of explanation*

- Line 353: "resp." should be "or" (there are few other places where "resp." appears)

*We have changed this accordingly*

- Line 387: Sugidachi et al., 2025 is also a good example.

*We have added Sugidachi et al., 2025.*

- Line 434: "less than 0.2 K error" – please add an explanation why (although this may have already been explained implicitly somewhere).

*We reformulated the text and write now: "CFH has been shown to measure the frost point with less than 0.2 K error under equilibrium conditions in the laboratory (Fahey et al., 2014) and under good flight conditions (Vömel et al., 2016), which we assume to also apply also to the equilibrium situation of the Golden Points (see further Section 5.5 for a discussion of the uncertainty at the Golden Points)."*

Also, add "vapour" to water mixing ratio.

*We have added "vapor" (American english)*

- Paragraph starting from line 438: The technical information on FLASH-B may be moved to Section 2, so that we can concentrate on the Golden Points concept itself here.

*We have moved the technical information on FLASH-B to the Appendix*

- Paragraph starting from line 466: The technical information on RS41 may be moved to Section 2, so that we can concentrate on the Golden Points concept itself here.

*We have moved the technical information on RS41 to the Appendix.*

Also, for RS41 data provided from the ground-receiving system, a time-lag correction may have already been applied within the manufacturer software. Does the sentence mean that the authors make an additional time-lag correction? In other words, do you mean that the manufacturer's time-lag correction is insufficient? Please clarify this in the text.

*No, we apply our own time-lag correction from the raw humidity of the heated humidity sensor and humidity sensor temperature data in the Vaisala sounding files (raw humidity of the heated humidity sensor and humidity sensor temperature). We have now clarified this in Section 3.2. by referring to the Vaisala data (raw RH and T).*

- Line 520: Please add one sentence to justify this simplification.

*We have added the Re = rho_air /mu_air*V/L equation, stating that mu_air approx. 1.7x10-5 m2s-1 from the ground to the middle stratosphere and that L = 3.5 mm (Section 4.1.2.).*

- Lines 563-573: This part is related to the Golden Points. I think that this may be moved to Section 3.1, so that we can concentrate on the nonequilibrium equation here.

*That is indeed related to the Golden Points, but we believe it fits best here as the paragraph before describes different cases of dUm/dt,*

- Line 587: The term "standard diffusion coefficient" should appear here, not at line 590.

*We have changed this accordingly.*

- Section 4.2 (corrections to RS41 data) and Section 4.3 (corrections to FLASH-B data) may be moved to Section 2 or to Appendix/Supplement, because this manuscript mainly discusses the CFH data corrections (and the readers would be primarily interested in those, and probably not the details of the RS41 and FLASH-B corrections although they are important). Having rather long subsections here may disturb the logical flow.

*We have moved the description of the correction methodologies for RS41 and FLASH-B to the Appendix.*

- Lines 942-944: This last sentence, starting from "However" is not clear to me in terms of the logical relationship with the previous sentences. What is the point?

*Agreed. We have removed this and the previous sentence as it may confuse readers.*

- Lines 1004-1005. I am confused here. Is 10% RH "high"? Probably, additional explanation is needed why this number can be regarded as high.

*We have improved the text, now saying that RH is sufficiently high for the RS41 sensor.*

- Lines 1046-1047: "Thus, it is unclear whether PID control is the most appropriate approach to frost control of chilled mirror hygrometers.": Do you mean that there could be other, more sophisticated and/or appropriate control methods for upper-air chilled mirror hygrometry? Please add a few more words to explain this.

*Yes, we have a 'Golden Points' seeking algorithm in mind that does not rely on PID but we have not tested it yet. We have added this in the text (right before Section 5.5).*

- Section 5.5.1, in particular, the part "High humidity and low number density of aerosol particles point to the polar vortex as a possible region of origin.": Just out of curiosity, I looked at a reanalysis data set MERRA-2 (https://gmao.gsfc.nasa.gov/reanalysis/MERRA-2/docs/) whether this could be a case. NASA/GMAO has a website, https://fluid.nccs.nasa.gov/reanalysis/, where we can quickly see weather charts, for example, of potential vorticity at 50 hPa over the Arctic region: https://fluid.nccs.nasa.gov/reanalysis/classic_merra2/?stream=MERRA2&field=epv&level=50&fcst=2018 0419®ion=nps&tau=00&track=none

[Figure]

Thu 04/19/2018 00Z

The above figure shows that airmass with relatively high PV values (maybe in the form of differential advection) was actually coming to the western Europe region. The maximum PV stream (colored in yellow) over Europe is located somewhat to the east, but this may be related to difference in time (00 UTC for this figure), actual balloon trajectory, the choice of 50 hPa, etc. More investigation would be needed by downloading and analyzing full model-level reanalysis data (e.g., MERRA-2, ERA5, etc.) to investigate whether the hypothesis is true, although I am not suggesting to do so for this paper manuscript.

*We thank the reviewer for this analysis of the MERRA-2 reanalysis data set. We have performed a similar analysis, but decided not to include it in the paper.*

- Line 1185: "0.4 s" – perhaps add "for this particular case"

*We have added this*

---

## Author Comment (AC3)

**Review of**

**'The "Golden Points" and nonequilibrium correction of high-accuracy frost point hygrometers' by Poltera et al.**

**Author response to reviewer comments**

*We would like to thank the two anonymous reviewers and Takuji Sugidachi for their constructive and critical evaluations. Some of their comments were challenging and required considerable effort, but they also helped to improve the quality and readability of our manuscript. The reviewers' comments are listed below in black font, and our responses are in italicized blue font. We would like to thank our reviewers once again, as well as the editor for allowing us additional time for revision.*

**CC1 - Community Comment 1**

**'Comment on egusphere-2025-2003', Takuji Sugidachi, 06 Jul 2025**

**Citation: https://doi.org/10.5194/egusphere-2025-2003-CC1**

This article demonstrates that the use of "Golden Points" and the nonequilibrium correction for chilled-mirror hygrometers results in more reasonable frost point profiles with higher vertical resolution. I believe these approaches are based on the measurement principles of chilled-mirror hygrometers, and therefore provide a more appropriate method for mitigating the oscillations caused by PID control in the mirror temperature profile.

I have a specific comment regarding the section on the SKYDEW hygrometer. Section 5.6 describes the application of the Golden Point method to SKYDEW. To compensate the timing error associated with the Golden Point, the mirror temperature data are shifted by 0.4 s relative to the scattered light signal. This 0.4 s shift appears to be appropriate for the profile shown in Figure 10, as well as for many other cases where shifts in the range of 0 ~ 0.6 s are observed. I am aware that the timing of the condition $U_m/dt=0$ often lags behind the expected frost point. Do you have a theoretical explanation for why this shift is necessary? Or do you consider this timing error to be random?

Additionally, you mention that a small $|dU_m/dt|$ indicates a good measurement (e.g., lines 404 and 1372). However, for chilled-mirror hygrometers, the profiles with small $|dU_m/dt|$ typically indicates low sensitivity to changes in the condensate. In my view, an ideal measurement is when the profiles of mirror reflectance (or scattering) signal and mirror temperature exhibit large $|dU_m/dt|$ and small $|dT_m/dt|$, which makes more reliable and less uncertain determination of the point where $dU_m/dt=0$.

*Dear Takuji Sugidachi,*

*Thank you for your comment on our manuscript.*

*(i) Timing error on the SKYDEW hygrometer*

*We believe that the instrument developers best understand the source of the timing error, as it originates from sub-second internal processing and telemetry aspects.*

*Usually, the mirror temperature Tm and the reflectance Um are not sampled in parallel in a chilled-mirror hygrometer, but at different times (within 1 second). The microcontroller sometimes even performs multiple other actions between the sampling of Tm and Um. In order to reduce the timing error, these sub-second differences in sampling (and possibly averaging) time between Tm and Um should be taken into account and - if possible - be reduced as far as is feasible.*

*We do not consider the timing error to be random, but rather of systematic nature, i.e., constant over a given flight section. At least, this is how we have successfully treated the case study in Section 5.6 (Figure 10). However, here again, instrument developers are best suited to answer this question. The residual timing error, i.e., the error after correction of sub-second timing issues, might be considered of random nature.*

*(ii) Metric for good chilled mirror measurement*

*There are arguably different metrics for a high-quality frost point measurement.*

*One metric is the frequency of Golden Points (dUm/dt=0) occurrences. Large |dUm/dt| coupled with frequently occurring dUm/dt=0 and well-synchronized and accurate Tm data is a good quality measurement. The main advantages are the high frequency of Golden Points and the self-correction ability (which is made possible through the high frequency of Golden Points and large |dUm/dt| in-between Golden Points, see the case study in Section 5.7). A disadvantage is the post-processing requirement, as non-equilibrium errors between the Golden Points and sub-second timing issues degrade the quality and vertical resolution of the measurement in case they are not corrected (or smoothed out).*

*Another metric is the nonequilibrium error |Tm-Tf|. Small |dUm/dt|, together with a high sensitivity to changes in the condensate (A) and well-synchronized and accurate Tm data is also a good quality measurement (Eq. 8a). As you point out, the disadvantage is that small |dUm/dt| alone do not guarantee that the nonequilibrium error is small, as small |dUm/dt| might be caused by a low sensitivity to changes in the condensate, which is difficult to determine at first glance in the absence of additional sensors such as RS41 or FLASH on the same payload.*

*In our view, the frost point measurements achieve highest quality when all conditions are fulfilled: i) frequent dUm/dt=0 occurrences, coupled with ii) small |dUm/dt| and iii) a high sensitivity to condensate.*

*You describe the situation with large |dUm/dt| and small |dTm/dt|. This would indeed be an ideal measurement scenario for Golden Point (dUm/dt=0) sampling, especially if the Golden Points occur frequently. Large |dUm/dt| requires large non-equilibrium error |Tm - Tf| (see Eq. 8a). This means that for small |dTm/dt| and large |dUm/dt| to occur, |dTf/dt| has to be large. There are however many situations where |dTf/dt| is not large (i.e., where the atmospheric frost point is varying only slowly), such as during pre-launch, within clouds, within the well-mixed boundary layer, or in the stratosphere.*

*In addition, for the large |dUm/dt| and small |dTm/dt| measurement strategy to work consistently, the morphological sensitivity (A') and/or ventilation (v) would have to increase with altitude, which is challenging to implement (see Eq. 8a).*

*For these reasons, while we acknowledge that large |dUm/dt| and small |dTm/dt| would be an ideal measurement scenario, we don't think there is an instrument design that can consistently achieve this.*